# Variability of distributions of wave set-up heights along a shoreline with complicated geometry

Tarmo Soomere[1,2], Katri Pindsoo[1], Nadezhda Kudryavtseva[1], Maris Eelsalu[1]

[1]Laboratory of Wave Engineering, Department of Cybernetics, School of Science, Tallinn University of Technology, Akadeemia tee 21, Tallinn, 12618, Estonia
[2]Estonian Academy of Sciences, Kohtu 6, Tallinn, 10130, Estonia

*Correspondence to*: Tarmo Soomere (soomere@cs.ioc.ee)

**Abstract.** The phenomenon of wave set-up may substantially contribute to the formation of devastating coastal flooding in certain coastal areas. We study the appearance and properties of empirical probability density distributions of the occurrence of different set-up heights on an approximately 80 km section of coastline near Tallinn in the Gulf of Finland, the eastern Baltic Sea. The study area is often attacked by high waves propagating from various directions and the typical approach angle of high waves varies considerably along the shore. The distributions in question are approximated by an exponential distribution with a quadratic polynomial as the exponent. Even though different segments of the study area have substantially different wave regimes, the leading term of this polynomial is usually small (between –0.005 and 0.005) and varies insignificantly along the study area. Consequently, the distribution of wave set-up heights substantially deviates from a Rayleigh or Weibull distribution (that usually reflect the distribution of different wave heights). On about ¾ of the occasions it is fairly well approximated by a standard exponential distribution. In about 25% of the coastal segments it qualitatively matches a Wald (inverse Gaussian) distribution. This property signals that very high extreme set-up events may, in some locations, occur substantially more frequently than is expected from the probability of occurrence of severe seas.

## 1 Introduction

Global sea level rise in most of existing projections of climate change (Cazenave et al., 2014) is often associated with major consequences for the coastal zone (Hallegatte et al., 2013). The resulting economic damages to low-lying coastal areas (Darwin and Tol, 2001) may lead to a significant loss of worldwide welfare by the end of this century (Pycroft et al., 2016). Global sea level rise, however, contributes only a small fraction to the most devastating coastal floods. These events, in addition to being economically extremely damaging (Meyer et al., 2013), may also lead to massive loss of life and destruction of entire coastal communities (Dube et al., 2009).

A devastating flood is usually caused by the interplay of several drivers, with fundamentally different predictability, and physical, dynamic and statistical properties. A reasonable forecast of the joint impact of tides, low atmospheric pressure (inverted barometric effect), wind-driven surge and wave-induced effects requires a cluster of dedicated atmospheric, ocean circulation and wave models. The resulting high water levels may be additionally amplified by specific mechanisms and

events such as tide–surge interactions (Batstone et al., 2013; Olbert et al., 2013), meteorologically driven long waves (Pattiarachi and Wijeratne, 2014; Pellikka et al., 2014; Vilibic et al., 2014) or seiches (Vilibic, 2006; Kulikov and Medvedev, 2013). In addition, wave-driven effects at the waterline such as wave set-up and run-up (Stockdon et al., 2006) may greatly contribute to the damaging potential of extreme water levels. These phenomena are driven by momentum carried by waves,

and have different time scales and appearance. When a wave crest reaches the shore, the resulting temporary inland movement of the water, with a time scale comparable with the wave period, is termed run-up (see, e.g., Didenkulova, 2009 for an overview and references). In contrast, wave set-up is the increase in the mean water level due to the release of momentum of breaking waves.

Along with contemporary numerical simulations and a direct search for worst-case scenarios (e.g., Averkiev and Klevanny,
2010), the use of the probabilistic approach is another classic way to quantify the properties of extreme water levels and related risks. The relevant pool of literature contains substantial amounts of work on statistical parameters of water level variations (e.g., Serafin and Ruggiero, 2014; Fawcett and Walshaw, 2016), and extreme water levels and their return periods (e.g., Purvis et al., 2008; Haigh et al., 2010; Arns et al., 2013). Similar probabilistic analysis has been extensively applied to the average and extreme wave properties (e.g., Orimolade et al., 2016; Rueda et al., 2016), wave-driven effects at the
waterline (Holland and Holman, 1993; Stockdon et al., 2006), and properties of meteotsunamis (Geist et al., 2014, Bechle et al., 2015). On most occasions severe coastal flooding occurs under the joint impact of several drivers. This feature generates the necessity to consider multivariate distributions of their properties. Most often, the simultaneous occurrence of storm surges and large waves is addressed (e.g., Hawkes et al., 2002; Wadey et al., 2015; Rueda et al., 2016b). A few studies also include an analysis of joint distributions of significant wave heights, periods and directions (Masina et al., 2015).

Typical probability distributions of different constituents of extreme water levels may be fundamentally different. The distribution of observed and numerically simulated water levels is usually close to Gaussian (Bortot et al., 2000; Johansson et al., 2001; Mel and Lionello, 2014; Soomere et al., 2015). The total water level in semi-sheltered seas with extensive subtidal or weekly-scale variability may contain two components. In the Baltic Sea, one of these (that reflects the water volume of the entire sea) has a classic quasi-Gaussian distribution whereas the other component (that reflects the local storm
surge) has an exponential distribution and apparently mirrors a Poisson process (Soomere et al., 2015) similar to the non-tidal residual in the North Sea (Schmitt et al., 2018). The probabilities of occurrence of different single wave heights are at best approximated either by a Rayleigh (Longuet-Higgins, 1952), Weibull (Forristall, 1978) or Tayfun distribution (Socquet-Juglard et al., 2005). The probability distribution of run-up heights usually follows the relevant distribution for incident wave heights (Denissenko et al., 2011) but can be approximated by a Rayleigh distribution even if the approaching wave field does
not represent a Gaussian process (Denissenko et al., 2013). The empirical probabilities of average or significant wave heights in various offshore conditions usually resemble either a Rayleigh or a Weibull distribution (Muraleedharan et al., 2007; Feng et al., 2014) while Pareto-type distributions are more suitable for the analysis of meteotsunami heights (Bechle et al., 2015).

In this paper we focus on the appearance and properties of empirical distributions of wave-driven local water level set-up. This process, called set-up in the following, is a classic phenomenon on open ocean coasts. It may often provide as much as

1/3 of the total water level rise during a storm (Dean and Bender, 2006) and significantly contribute to extreme sea level events (Hoeke et al., 2013; Melet et al., 2016, 2018). The impact of this phenomenon *inter alia* contributes to the overall level of danger in the coastal zone because, for example, the baseline level of wave run-up (Leijala et al., 2018) includes the local elevation of water level owing to set-up. The physics of set-up has been known for half a century (Longuet-Higgins and Stewart, 1964). Adequate parameterizations of this phenomenon have been introduced more than a decade ago (Stockdon et al., 2006) and many models take it into account to a certain extent (SWAN, 2007; Roland et al., 2009; Alari and Kõuts, 2012; Moghimi et al., 2013).

The contribution from wave set-up still provides one of the largest challenges in the modelling of storm surges and flooding (Dukhovskoy and Morey, 2011; Melet et al., 2013). This reflects the intrinsically complicated nature of its formation. First of all, the set-up height strongly depends on the approach angle of waves at the breaker line. This angle is well-defined only if the coastline is almost straight, the nearshore is mostly homogeneous in the alongshore direction and the wave field is close to monochromatic (Larson et al., 2010; Viška and Soomere, 2013; Lopez-Ruiz et al., 2014; 2015). Generally, this angle is a complicated function of shoreline geometry, nearshore bathymetry, wave properties and instantaneous water level. Even if the basic statistical properties of wave fields (usually given in terms of significant wave height, mean or peak period, and mean propagation direction) are perfectly forecast or hindcast in a nearshore location, the evaluation of the further propagation of waves is a major challenge because, for example, refraction properties and the location of the breaking line change with the local water level.

Several studies have focused on the maxima of set-up heights over certain coastal areas (Soomere et al., 2013; O'Grady et al., 2015) or the maximum contribution from set-up to the local water level extremes (Pindsoo and Soomere, 2016). The problem of evaluation of maximum set-up heights has a relatively simple solution on comparatively straight open ocean coasts. The nearshore of such coasts is usually fairly homogeneous in the alongshore direction and the highest waves tend to approach the shore under relatively small angles. These features make it possible to use simplified schemes for the evaluation of the joint impact of refraction and shoaling in the nearshore (e.g., Larson et al., 2010). On many occasions it is acceptable to assume that waves propagate directly onshore (O'Grady et al., 2015) or to reduce the problem to an evaluation of the properties of the highest waves that approach the shore from a relatively narrow range of directions (Soomere et al., 2013). In areas with complicated geometry and especially in coastal segments where high waves may often approach at large angles it is necessary to take into account full refraction and shoaling in the nearshore (Viška and Soomere, 2013; Pindsoo and Soomere, 2015).

Even though high storm surges are often associated with severe seas, the formation of high set-up depends on many details of the storms and the impacted nearshore. It does not necessarily exhibit its maximum level in the coastal sections that are affected by the highest waves. The maximum storm surge and maximum set-up usually do not occur simultaneously (Pindsoo and Soomere, 2015). On the contrary, in coastal areas with complicated geometry each short segment may have its own 'perfect storm' that creates the all-highest sum of storm surge and set-up (Soomere et al., 2013). These observations call for further analysis of the properties of the set-up phenomenon.

As described above, research into the statistical properties of the main drivers of high local water levels and the reach of swash generated by large waves that attack the shore have revealed that the relevant distributions of the magnitude of these drivers are very different. They may include a Gaussian distribution for the water volume of the Baltic Sea (Soomere et al., 2015), an exponential distribution for storm surges (Schmitt et al., 2018), a quasi-Gaussian distribution for water levels at the

shores of the Baltic Sea (Johansson et al., 2011), a Weibull distribution for different significant wave heights (Feng et al., 2014), and a Weibull or Rayleigh distribution for wave run-up heights (Denissenko et al., 2013). The knowledge of the shape and parameters of such distributions is often crucial in various forecasts and management decisions.

In this paper we address the basic features of statistical distributions of set-up heights along an approximately 80 km coastal section in the vicinity of Tallinn Bay in the Gulf of Finland, the Baltic Sea. The shoreline of the study area has a complicated

geometry and contains segments with greatly different orientations. The goal is to identify the typical shapes of the distributions of the probability of occurrence of simulated wave set-up heights and to analyse the alongshore variability of these distributions.

The paper is organised as follows. Section 2 introduces the method of evaluation of the maximum set-up height for obliquely approaching waves. It also provides a short overview of the simplified method for rapid reconstruction of long-term wave

climate, the forcing data for the underlying wave model and the procedure of evaluation of properties of breaking waves based on the output of the wave model. Section 3 presents an analysis of spatial variations in the appearance of the empirical distribution of wave set-up heights in the study area. The core result is an estimate of the typical shape of empirical probability distributions of different set-up heights along the coast. A part of these distributions substantially deviate from the listed distributions and exhibit an unexpectedly large proportion of high set-up events compared to the classic Gaussian,

Rayleigh or Weibull statistics. The Kolmogorov-Smirnov test is applied to estimate the goodness of match of the empirical distribution of set-up heights with common theoretical distributions. Several implications of the results are discussed in Section 4.

## 2 Methods and data

### 2.1 Set-up height for obliquely incident waves

The classic concept of wave set-up (Longuet-Higgins and Stewart, 1962) relates the local increase in the water level with the release of the onshore component of radiation stress in the process of wave breaking. Based on this concept, it has been demonstrated that, in ideal conditions, the maximum set-up height $\eta_{\max}$ (with respect to the still water level) created by a train of monochromatic waves with a constant height propagating directly onshore along a planar impermeable beach is (McDougal and Hudspeth, 1983; Hsu et al., 2006)

$$\eta_{\max} = d_b \frac{40\gamma_b^2 - 3\gamma_b^4}{128} \approx \frac{5}{16}\gamma_b H_b, \tag{1}$$

where $\gamma_b = H_b/d_b$ is the breaking index that is assumed to be constant all over the surf zone, $d_b$ is the still water depth at the breaker line and $H_b$ is the wave height at the breaker line (Fig. 1). Expression (1) is used in many engineering applications (Dean and Dalrymple, 1991) and studies into the properties of set-up of waves that approach the shore at a relatively small angle (see Soomere et al., 2013 and references therein).

5   If waves approach under a non-negligible angle $\theta$ with respect to the shore-normal, the situation is much more complicated. Shi and Kirby (2008) argue that the water level set-down at the breaker line is invariant with respect to the approach angle of waves. The average deviation $\eta$ of the sea surface from the still water level within the surf zone of an impermeable planar beach is (Hsu et al., 2006; Shi and Kirby, 2008; the power of $\gamma_b$ in the first term at the right-hand side of their expression being corrected):

$$\eta = \frac{\gamma_b^2 \sin^2 \theta_b}{2h_b\left(8 + 3\gamma_b^2 - 2\gamma_b^2 \sin^2 \theta_b\right)}\left(d^2 - d_b^2\right) - \frac{3\gamma_b^2 - 2\gamma_b^2 \sin^2 \theta_b}{8 + 3\gamma_b^2 - 2\gamma_b^2 \sin^2 \theta_b}\left(d - d_b\right) - \frac{\gamma_b^2}{16} d_b . \tag{2}$$

The last term at the right-hand side of Eq. (2) represents the water level set-down $\eta_b$ at the breaker line and $\theta_b$ is the wave approach direction at breaking. Here $d = d(x)$ represents the water depth counted from the still water level at a particular distance $x$ from the shoreline and $\eta$ is a function of $x$. The maximum wave set-up $\eta_{\max}$ occurs somewhere inland where $d_{\max}$ is negative and the thickness of the water sheet $d^* = d + \eta_{\max} = 0$, and thus $\eta_{\max} = -d_{\max}$. For this location, Eq. (2)

15   reduces to:

$$\eta_{\max} + d_{\max} = \frac{\gamma_b^2 \sin^2 \theta_b}{2h_b\left(8 + 3\gamma_b^2 - 2\gamma_b^2 \sin^2 \theta_b\right)}\left(d_{\max}^2 - d_b^2\right) - \frac{3\gamma_b^2 - 2\gamma_b^2 \sin^2 \theta_b}{8 + 3\gamma_b^2 - 2\gamma_b^2 \sin^2 \theta_b}\left(d_{\max} - d_b\right) - \frac{\gamma_b^2}{16} d_b + d_{\max} = 0 . \tag{3}$$

For shore-normal waves $\theta_b = 0$ and Eq. (3) reduces to a linear equation:

$$-\frac{3\gamma_b^2}{8 + 3\gamma_b^2}\left(d_{\max} - d_b\right) - \frac{\gamma_b^2}{16} d_b + d_{\max} = 0 . \tag{4}$$

In this case the maximum set-up height $\eta_{\max}$ is defined by Eq. (1).

20   For obliquely approaching waves Eq. (3) is a quadratic equation with respect to $q = d_{\max}/d_b$:

$$\frac{\gamma_b^2 \sin^2 \theta_b}{2\left(8 + 3\gamma_b^2 - 2\gamma_b^2 \sin^2 \theta_b\right)}\left(q^2 - 1\right) - \frac{3\gamma_b^2 - 2\gamma_b^2 \sin^2 \theta_b}{8 + 3\gamma_b^2 - 2\gamma_b^2 \sin^2 \theta_b}\left(q - 1\right) - \frac{\gamma_b^2}{16} + q = 0 . \tag{5}$$

This equation can be rewritten as

$$8q^2 \gamma_b^2 \sin^2 \theta_b + 128q + \gamma_b^2\left(40 - 3\gamma_b^2\right) - \gamma_b^2 \sin^2 \theta_b\left(40 - 2\gamma_b^2\right) = 0 . \tag{6}$$

Equation (6) has two negative solutions for physically reasonable values of $\gamma_b$. The physically relevant solution to Eq. (6) must be bounded and should be almost equal to $q \approx -5\gamma_b^2/16$ for very small approach angles $\theta_b \approx 0$. Therefore, the expression

$$q_1 = \frac{-32 + \sqrt{1024 - 2\gamma_b^4 \sin^2 \theta_b \left[40 - 3\gamma_b^2 - \sin^2 \theta_b \left(40 - 2\gamma_b^2\right)\right]}}{4\gamma_b^2 \sin^2 \theta_b} \tag{7}$$

provides the desired solution. Equation (7) deviates from expression (30) of Hsu et al. (2006) by reasons discussed by Shi and Kirby (2008). The maximum set-up height for obliquely approaching waves is thus

$$\eta_{max} = -q_1 d_b = -\frac{q_1 H_b}{\gamma_b} . \tag{8}$$

This quantity is called simply set-up height below.

## 2.2 Wave time series in the nearshore of the study area

We evaluate the shape and parameters of the empirical probability distribution set-up heights along an approximately 80 km coastal segment of Tallinn Bay and Muuga Bay (Fig. 2). The study area is an example of a wave-dominated micro-tidal region. The shoreline is locally almost straight for scales up to a kilometre or two. Several relatively straight parts along the Suurupi Peninsula (grid points 1–10 in Fig. 2) and the area of Saviranna (grid points 137–143 in Fig. 2) are open to the north. However, at larger scales (from a few kilometres) the coast contains large peninsulas and bays deeply cut into the mainland.

The shores of these landforms are open to different directions and have greatly different wave regimes (Soomere, 2005). As the formation of set-up crucially depends on the wave height and direction (or approach angle), this type of coastal landscape makes it possible to analyse the wave set-up distribution for coastal sections with radically different wave climates, and also the associated magnitudes of set-up (Soomere et al., 2013).

The fetch length in the Gulf of Finland is >200 km for westerly and easterly winds but <100 km for all other wind directions.

The highest significant wave height (5.2 m) in the Gulf of Finland has been recorded twice in a location just a few tens of km to the north of the study area (Tuomi et al., 2011). The strong winds in this region blow predominantly from the south-west and north-north-west. Easterly storms are less frequent but may generate waves as high as those generated by westerly storms (Soomere et al., 2008). Strong storms with winds from the north-north-west may generate significant wave heights >4 m in the interior of Tallinn Bay (Soomere, 2005). The varying mutual orientation of high winds, propagation direction of

waves and single shoreline segments makes it possible to identify potential alongshore variations in the distributions of set-up heights.

We employ time series of wave properties (significant wave height, wave period and propagation direction) reconstructed using the wave model WAM cycle 4 and one-point high-quality wind information from the vicinity of the study area. The

wave model is implemented in a triple nested version with the resolution of the innermost grid about 470 m (Soomere, 2005). The regular rectangular grid of the coarse model covered the whole Baltic Sea with a step of $3 \times 6'$ (3′ along latitudes and 6′ along longitudes, that is, about three nautical miles). We applied 24 evenly spaced directions and 25 frequencies ranging from 0.042 to 0.41 Hz with an increment of 1.1.

Experience with this model in the Baltic Sea and Finnish archipelago indicates that it is important to adequately represent the wave growth in low wind and short fetch conditions (Tuomi et al., 2011; 2012). To meet this requirement, in calculations with the wind speed below 10 m/s we applied an increased frequency range of waves up to 2.08 Hz in order to correctly represent wave growth in low wind conditions after calm situations (Soomere, 2005). Full wave spectrum from the coarse model was used as boundary conditions for the first nesting with a similar regular grid with a step $1 \times 2'$ (about 1 nautical

mile). This grid covered the interior of the Gulf of Finland to the east of 23°18′E. The bathymetry of these two models is based on data from (Seifert et al., 2001) with a resolution $1 \times 2'$. Full wave spectrum from the model for the Gulf of Finland was used as boundary conditions for the second nesting. The study area (Fig. 2) was covered with a regular grid with a step of $1/4 \times 1/2'$ between 24°28′ and 25°16′E and to the south of 59°41′N. The bathymetry for the second nesting bathymetry is constructed based on maps issued by the Estonian Maritime Board with typical resolution of about 200 m. The wave height

is presented in terms of significant wave height $H$ . This quantity, often denoted as $H_S$ and below called wave height, is defined as $H = \sqrt{m_0}$ , where $m_0$ is the zero-order moment of the one-dimensional wave spectrum, and is estimated with a resolution of 1 cm.

    The nearshore of the study area is divided into 174 coastal segments with a length of about 500 m (Fig. 2). Each segment corresponds to a nearshore wave model grid cell. Ignoring the presence of sea ice may lead to a certain overestimation of the

overall wave energy in the region but apparently does not significantly distort the shape of the probability distribution of different wave heights (Fig. 3). It is therefore likely that the shape of distributions of set-up heights and the variation in these distributions along the shoreline are also reconstructed adequately.

    We employ a simplified method for rapid reconstruction of long-term wave climate. The computations are speeded up by replacing exact calculations of wave generation, interactions and propagation by an analysis of precomputed maps of wave

properties for different wind speed, direction, and duration. This simplification avoids reconstruction of all the wave time series and relies on a favourable feature of the local wave regime, namely that wave fields rapidly become saturated and have relatively short memory in the study area (Soomere, 2005). Consequently, a reasonable reproduction of wave statistics is possible by the assumption that an instant wave field in Tallinn Bay is a function of a short time period of wind dynamics. This assumption justifies splitting the calculations of time series of wave properties into independent sections with duration

of 3–12 hours. The details of the model set-up, bathymetry used, and the implementation and validation of the outcome have been repeatedly discussed in the literature (Soomere, 2005; Soomere et al., 2013).

    The described approach makes it possible to circumvent one of the major issues of replication of Baltic Sea wave fields, namely, the frequent inconsistency of different modelled wind data sets (Nikolkina et al., 2014). Similar to, for example,

wave-driven sediment transport, wave set-up is intrinsically sensitive with respect to the wave propagation direction. As the nearshore wave directions in areas with complex geometry and bathymetry may be greatly impacted by local features, it is crucial to properly reconstruct the offshore wave directions. This is only possible if the wave model has correct information about wind directions. This is an issue in the Gulf of Finland where atmospheric models often fail to reproduce wind

directions (Keevallik and Soomere, 2010). To overcome this issue, we use wind data from an offshore location in the central part of this gulf. The wind recordings at Kalbådagrund (59°59′ N, 25°36′ E, a caisson lighthouse located on an offshore shoal) are known to represent marine wind properties well (Soomere et al., 2008). Even though this site is located at a distance of about 60 km from the study area, it is expected to correctly record wind properties in the offshore that govern the generation of surface waves in the open sea.

Wind properties at Kalbådagrund were recorded starting from 1981 once every 3 hours for more than two decades, but since then they have been filed at a higher time resolution. To ensure that the forcing data is homogeneous, we downsampled the newer higher-resolution recordings by selecting the data entries once in 3 hours. The entire simulation interval 1981–2016 contained 103 498 wind measurement instants with a time step of 3 h. In about 9000 cases (less than 10% of the entire set) either wind speed or direction was missing. These time instants were excluded from the further analysis. As some of these

instants involved quite strong winds, our analysis may underestimate the highest wave set-up events for some segments of the shore. However, as we are interested in the statistical properties of most frequently occurring set-up heights and in the alongshore variations of these properties, it is likely that omitting these data does not substantially impact the results.

To roughly estimate the adequacy of the described method for speeding up the estimates of the properties of wave climate via rapid reconstruction of time series of approximate wave properties, its outcome is compared with the results of wave

measurements made by Marine Systems Institute, Tallinn University of Technology, at Tallinnamadal (Fig. 2, 59°42.723′ N, 24°43.890′ E). The data come from a pressure sensor. The accuracy of estimates of significant wave height is ±0.1 m. The description of the location and parameters of the sensor are presented in http://efficiensea.org/files/mainoutputs/wp4/efficiensea_wp4_27.pdf. The data set is available at https://www.emodnet-physics.eu/map/platinfo/piroosplot.aspx?platformid=8974.

As the measurements are available starting from 2012, the comparison is performed for the time interval of 2012–August 2016. The measured wave properties are compared with the modelled properties at the closest grid point of the sea area represented at 470 m resolution at 59°41′ N, 24°45′ E. The buoy is located about 3 km from the border of this area and the distance between the buoy and the centre of the closest grid cell is 3.34 km. The comparison (Fig. 3) only includes the instants when both measured (green) and modelled (red) wave parameters were available. We use for comparison the

significant wave height

The basic properties of wave heights such as the maximum (measured 5.58 m, modelled 5.77 m), mean (measured 0.643 m, modelled 0.697 m) and median (measured 0.40 m, modelled 0.54 m) are represented reasonably. The bias of the model (about 0.05 m) is at the same level as the typical bias for modelled wave properties in the Baltic Sea in the most recent

simulations (Björkqvist et al., 2018). As our study basically relies on statistical properties of wave fields (the probability of occurrence of seas with different significant wave height, period, and direction), the analysis below, strictly speaking, does not require an exact reconstruction of the sequence of wave events. In this context, the root mean square difference of the modelled and measured time series wave heights (0.5 m) is reasonable. This value is about twice as large as in (Björkqvist et

al., 2018) for the Gulf of Finland (0.20–0.31 m) or northern Baltic Proper (0.26 m) and comparable to the level of this quantity for the Sea of Bothnia (0.31–0.56 m, Björkqvist et al., 2018).

The highest waves in the coastal areas over the entire simulation interval are plausible (Soomere et al., 2013, Figure 3). The maximum wave height in 1981–2012 was 5.2 m in a section that was fully open to one of the directions of the strongest winds from north-north-west. Such wave conditions have been measured twice since 2001 in the interior of the Gulf of

Finland (Soomere, 2005; Pettersson et al., 2013).

The appearance of the relevant empirical probability distributions of the occurrence of different wave heights is similar for both data sets (Fig. 3). The location and height of the peaks of these distributions (that represent the properties of most frequently occurring waves) have a reasonable match. The model overestimates to some extent the frequency of waves with heights of 0.8–1.5 m and underestimates the frequency of highest waves in the area (>2 m). The overall appearance of the

distribution for modelled wave heights resembles a Weibull distribution. The empirical distribution for measured wave heights higher than 0.4 m better matches an exponential distribution and exhibits much larger variability in the frequency of very high (>3 m) waves.

## 2.3 Nearshore refraction and shoaling

The nearshore grid cells selected for the analysis (Fig. 2) are located in water depth ≥4 m in order to avoid problems with

reconstruction of wave heights under possible intense wave breaking in these cells in the strongest storms. Some of the cells are located in much deeper water, at a depth of 20–27 m. The nearshore of the study area contains various underwater features and bottom inhomogeneities. This means that shoaling and refraction may considerably impact the wave fields even along the relatively short paths (normally ≤1 km in our model setup) from the model grid cells to the breaker line. The predominant wind directions in strong storms are from the south-west, north-north-west and west (Soomere et al., 2008).

Consequently, high waves often approach some of the selected grid cells at large angles with respect to the shore-normal. Therefore, it is not acceptable to assume that the incidence angles are small. As a result, oversimplified approaches to replicate the changes in wave properties in the immediate nearshore (Lopez-Ruiz et al., 2014; 2015) and even advanced approximations of refraction and shoaling (Hansen and Larson, 2010) may fail.

For this reason we calculate the joint impact of shoaling and refraction of approaching waves in the framework of linear

wave theory. Following (Soomere et al., 2013), we assume that the numerically evaluated wave field for each time instant is monochromatic. The wave height is characterised by the numerically simulated (significant) wave height $H$, peak period $T_p$ and mean propagation direction $\theta$ (clockwise with respect to the direction to the North). These properties are evaluated

at the centre of each selected grid cell. The significant wave height at this location is denoted as $H_0$. The approach direction $\theta_0$ at this location with respect to the onshore-directed normal to the shoreline is calculated from $\theta$ based on an approximation of the relevant (about 500 m long) coastal segment by a straight line that follows the average orientation of the shoreline in this segment. Similarly, it is assumed that the nearshore seabed from the centre of each grid cell to the waterline is a plane with isobaths strictly parallel to this straight line. Finally, we assume that breaking waves are long waves. Then the wave height $H_b$ at the breaking line can be found as the smaller real solution of the following algebraic equation of 6th order (Viška and Soomere, 2013; Soomere et al., 2013):

$$\frac{H_b^5 g}{H_0^4 \gamma_b}\left(1 - \frac{gH_b}{\gamma_b}\frac{\sin^2 \theta_0}{c_{f0}^2}\right) = c_{g0}^2\left(1 - \sin^2 \theta_0\right). \tag{9}$$

Here $c_g$ is the group speed, $c_f$ is the phase speed and the subscripts "0" and "b" indicate the relevant value at the centre of the particular wave model grid cell and at the breaker line, respectively. The phase and group speed at the wave model grid cell are estimated based on the standard expressions of linear theory using the wave number $k$ evaluated from the full dispersion relation for linear monochromatic waves $2\pi/T_p = \sqrt{gk \tanh kd_0}$ with the period equal to the peak period $T_p$ and the water depth $d_0$ equal to the model depth for the particular grid cell. We assume that waves at the breaker line (where the water depth is $d_b$) are long waves. The set of assumptions is completed with the common notion that the breaking index is $\gamma_b = H_b/d_b = 0.8$ (Dean and Dalrymple, 1991). Part of the introduced assumptions, such as a plane seabed and dry coast without any vegetation, monochromatic wave fields and a constant value of the breaking index for wind-seas as well as the ignoring of the wave period (or steepness) in the calculations are not fully realistic. The potential impact of these approximations is discussed in Section 4.

The procedure of evaluation of set-up heights is thus as follows. We start from the numerically simulated significant wave height $H_0$, peak period $T_p$, and mean propagation direction $\theta$ with respect to the direction to the North. Next we calculate the phase ($c_{fb}$) and group speed ($c_{gb}$) of such waves at the model grid cell and find the propagation direction with respect to the shore-normal $\theta_0$. Equation (9) is employed subsequently to evaluate the changes to the wave height owing to refraction and shoaling on its way from the model grid point to the breaker line. In essence, it links the model output $H_0$ (given in terms of significant wave height) with the height $H_b$ of breaking waves. The phase and group speed at the breaker line are estimated from the dispersion relation for long waves: $c_{gb} = c_{fb} = \sqrt{gd_b}$. The wave approach direction $\theta_b$ at the breaker line with respect to the shore-normal is calculated from Snell's law $\sin \theta / c_f = const$. Thereafter we employ Eqs. (7) and (8) to find the set-up height for the particular time instant.

Several earlier studies of extreme set-up heights (Soomere et al., 2013; Pindsoo and Soomere, 2015) followed this procedure but took into account only waves that propagated under high angles (not larger than ±15°) with respect to the shore-normal and ignored the correction expressed in Eqs. (7, 8) for waves that approached under a nonzero angle. This approach is denoted S2013 below. It is adequate on the open ocean coasts where waves usually approach the shore under relatively small angles but it may fail in semi-sheltered basins with short fetch.

## 3 Results

### 3.1 Maximum set-up heights

The phenomenon of wave set-up is only significant if large waves propagate towards the shore. This is usually the case on open ocean coasts where swells almost always create set-up. The situation may be different in sheltered sea areas with complicated geometry where intense swells may be infrequent and the majority of the wind wave energy may propagate in an offshore direction, this being common in the study area. The wind regime of the study area is a superposition of four wind systems (Soomere et al., 2008). The most frequent wind direction is from south-west (that is, from the mainland to the sea). The proportion of wave fields that propagate onshore is 40–70% along the entire study shoreline (Fig. 4). The statistical properties of set-up heights discussed below thus represent 40 000–70 000 examples of wave fields in each coastal segment. The only exception is grid cell 107 (Figs. 2, 4) between Viimsi Peninsula and the island of Aegna that is sheltered for almost all wind directions.

We start from a comparison of maximum set-up heights evaluated using the above-described approach and the method employed in S2013. The two sets of estimates differ insignificantly (by less than 0.1 m) in about 80% of the coastal segments (Fig. 5). The alongshore variations in the maxima of set-up heights evaluated from Eq. (8) are considerably smaller than those estimated using the approach of S2013. The largest examples of set-up heights reach 1 m and the majority of maximum set-up heights for single coastal sections are 0.6–0.8 m in both sets of estimates.

The largest differences between the two sets become evident in segments that are sheltered from predominant storm directions, most notably in deeply cut bays. Estimates based on Eqs. (7, 8) are often remarkably (by up to 50%) higher in these sections than those derived using S2013. This feature signals that the highest waves approach the shore at a relatively large angle in such sections. This property shows the importance of the generation of remarkable set-up heights by obliquely approaching high waves. Therefore, ignoring waves that approach under large angles may substantially underestimate the maximum set-up height in some coastal segments.

In other words, the impact of refraction often overrides the effect of geometric blocking of waves by changing the orientation of the coastline. Refraction thus often redirects wave energy so that even beaches that are seemingly well sheltered geometrically may at times receive remarkable amounts of wave energy (*cf* Caliskan and Valle-Levinson, 2008). The differences in the maxima of set-up heights evaluated using the two approaches for such coastal sections are often 0.2–0.3 m and reach up to 0.5 m. Such a strong impact of refraction is thought to be responsible for a local increase in wave heights in

the Baltic Sea (Soomere, 2003) and also in extreme ocean conditions (Babanin et al., 2011). The processes that are not resolved by phase-averaged wave models such as reflection and diffraction may add even more wave energy to seemingly sheltered coastal segments.

On the contrary, S2013 overestimates the maximum set-up height in a few locations at headlands that are fully open to the

Gulf of Finland (Fig. 5). A likely reason for such a sporadic overestimation is the sensitivity of the formation of set-up with respect to the approach angle of waves. The magnitude of set-up rapidly decreases with an increase in the approach angle. This decrease is ignored in S2013. As a result, the height of set-up created by waves that approached at angles of 10–15 degrees with respect to shore-normal was overestimated. This feature also demonstrates the importance of the correct evaluation of refraction and shoaling.

Some differences between the results presented in this paper and those described in Soomere et al. (2013) and Pindsoo and Soomere (2015) stem from the different time intervals used in the calculations. Simulations for 1981–2012 indicate that the maximum set-up heights in coastal areas open to the east were mostly from the 1980s (Soomere et al., 2013) even though the maximum wave heights occurred starting from the mid-1990s. This feature may be related to a change in the directional structure of strong winds with easterly storms being relatively weak for about two decades. There is increasing evidence,

however, that this process has reversed and strong easterly storms have returned to the area. Evidence of this change is the event with significant wave height 5.2 m that was recorded in the Gulf of Finland during an extreme easterly storm on 29–30 November 2012 (Pettersson et al., 2013). Pindsoo and Soomere (2015) observed that many new all-time highest set-up events apparently occurred in coastal segments open to the east since 2012. This process has led to the generation of maximum simulated waves at a number of locations on the eastern Viimsi Peninsula near Leppneeme (grid cells 115–117 in

Fig. 2) in 2010 (Fig. 6). These aspects will be addressed in more detail elsewhere.

## 3.2 Frequency of occurrence of set-up heights

As the modelled wave heights are evaluated with an accuracy of 1 cm, using even finer resolution for the construction of empirical probability distributions is not justified. The total number of entries in time series of positive set-up heights is 40 000–70 000. Therefore, using the resolution of 1 cm ensures that at least a few examples of set-up heights will belong to

the relevant "size classes" of set-up heights down to frequencies about $10^{-2}$%. This range is apparently large enough to estimate the main properties of the distributions in question.

The shape of the empirical distributions of the occurrence of set-up heights varies extensively in the study area (Fig. 7). This shape matches an exponential distribution in the majority (about 75%) of the model coastal segments[1]. Such a distribution is

---

[1] An early discussion version of this paper (available at https://www.earth-syst-dynam-discuss.net/esd-2016-76) contained a bug in the script for the calculation of set-up heights and for the subsequent evaluation of the parameters of their probability density function. This bug led to an erroneous conclusion about the frequency of occurrence of various kinds of distributions of set-up heights in different coastal sections as well as to severe overestimation of the frequency of matches of these distributions with an inverse Gaussian distribution.

represented by a straight line in the semilogarithmic (log-linear) coordinates used in Figs. 7, 8. It apparently reflects a background Poisson process that also describes storm surges in the study area (Soomere et al., 2015). This type of distribution appears for coastal segments that are open to the common strong wave directions. For example, grid point 23 (Fig. 7a) is open to the northwest and north, grid point 96 (Fig. 7c) is located near the western shore of Viimsi Peninsula and is open to the west and northwest, and grid point 145 (Fig. 7f) is widely open to the directions from the northwest to the northeast.

The distribution is convex upwards at a few locations that are sheltered from most of predominant approach directions of strong waves, including north-north-west. This shape is evident in the most sheltered location of eastern Kopli Bay (grid point 43, Fig. 7b) and to a lesser extent in coastal sections sheltered by the island of Aegna (grid point 106, Fig. 6d). The relevant empirical distributions of set-up heights can be reasonably approximated by a two-parameter Weibull or Gaussian distribution. They both have a convex upwards shape in log-linear coordinates.

A subset (about ¼) of the presented distributions exhibit a different, clearly concave upwards shape in log-linear coordinates. This feature is evident in coastal sections that are sheltered from a few predominant wave directions. Strong winds blow in this region usually from three directions: west, north-north-west, and east (Soomere et al., 2008). The segments that exhibit a concave upwards shape of the distribution (e.g., grid point 129, Fig. 6e) are mostly sheltered against waves that approach from the west or north-north-west but are open to the east. However, segments that are widely open to all strong wave directions (e.g., grid point 1, Fig. 7) may also exhibit a concave upwards shape of the empirical distribution of set-up heights. This concave upwards appearance clearly differs from the shape of the usual distributions of the magnitude of wave phenomena (Fig. 8) such as the classic (Rayleigh) distribution of single wave heights (Longuet-Higgins, 1952), the Tayfun distribution of the heights of largest waves, the Weibull family of distributions for the occurrence of various wave conditions, or the Rayleigh distribution for run-up of (narrow-banded) Gaussian wave fields (Didenkulova et al., 2008). Therefore, none of the above-mentioned distributions can be used for the universal approximation of the probabilities of different set-up heights.

To further explore the shape of the distributions of set-up heights and their possible variations along the shoreline, we assume that these distributions belong to the family of general exponential distributions. The overall appearance of empirical distributions in log-linear coordinates (Fig. 7) suggests that their shape can be, as a first approximation, matched with a quadratic polynomial $az^2 + bz + c$, where $z$ is the set-up height. In other words, the empirical probability density $P(z)$ is approximated by the following function:

$$P(z) = \exp(az^2 + bz + c) \qquad (10)$$

The fitting is performed in log-linear coordinates used in Figs. 7, 8. In the case $a = 0$, distribution (10) reduces to a classic exponential distribution $\sim A\exp(bz)$. The values $a < 0$ correspond to convex upwards distributions that eventually can be approximated by a Weibull or normal distribution whereas $a > 0$ indicates that the distribution is locally concave upwards.

A more important difference between distributions with $a = 0$ and $a \neq 0$ is in the nature of their tails. If $a = 0$, the probability of large set-up heights decreases as $\sim A \exp(bz)$ when $z \gg 1$ while in the case $a < 0$ this probability decreases much faster, as $\sim A \exp(az^2)$. It is often said that the former distribution has a heavy tail (and a comparatively large probability of very large set-up heights) whereas the latter distribution has a light tail (and a lower probability of very large

set-up heights). The case $a > 0$ is only possible locally for a certain range of set-up heights and serves as an indication that some larger set-up height are more probable that expected from distributions with $a \leq 0$.

Such a fitting procedure is not straightforward for several reasons. Firstly, the number of nonzero points of the distributions in Fig. 7 is highly variable along the study area similar to the variation in the typical magnitude of the set-up. Secondly, the relevant empirical distributions have gaps for some value(s) of the set-up height. A natural reason for this feature is that we

are looking at very low probabilities (down to 0.001%, that is, a few occasions) of occurrence of relatively high set-up events in 1981–2016. Thirdly, a few locations have several outliers. There are remarkably high set-up events that do not follow the general appearance of the empirical distribution of set-up heights for the particular location (Fig. 7b, d, f). Such events apparently reflect severe storms in which the wind pattern was favourable for the development of very large waves that approached a certain coastal segment at a small angle. The presence of similar outliers is characteristic, for example, for time

series of sea level in Estonian waters (Suursaar and Sooäär, 2007) and is associated with situations when strong storms blow from a specific direction.

To estimate the impact of these aspects on the results, we performed three versions of the fitting procedure. Firstly, we used all data points in the relevant distributions starting from the height of 0.01 m to evaluate the coefficients $a$, $b$ and $c$. Secondly, we used for the same purpose only set-up heights from 0.01 m to 0.4 m (Fig. 7). This approach was not applicable

in some locations where set-up heights did not reach 0.4 m. Thirdly, we evaluated these coefficients starting from the height of 0.01 m to the first gap in the empirical distribution (the lowest set-up height that did not occur in 1981–2016). Doing so made it possible to check whether the shape of the distribution is governed by the majority of events or if it is dominated by the presence of a few very large set-up heights (Fig. 8).

The particular values of the coefficients $a$, $b$ and $c$ depend to some extent on the chosen version (Fig. 7). The shape of the

approximate distribution is invariant with respect to the particular choice. All distributions also match reasonably well data points corresponding to the largest set-up heights. The differences between the resulting theoretical distributions are mostly insignificant. The relevant estimates are located almost in the middle of the 95% confidence intervals of each other (Fig. 7).

The coefficients $a$ at the leading term of the approximating polynomial (Fig. 9, 10) are mostly very small, in the range of (–0.005, 0.005). Their 95% confidence intervals normally include the zero value. This feature indicates that on most occasions

the parameter $a$ can be set to zero and the distribution of set-up heights can be reasonably approximated with an exponential distribution at a 95% level of statistical significance. On such occasions, the entire process can be adequately approximated by a Poisson process and the parameter $b$ characterises the vulnerability of the particular coastal segment with respect to the set-up phenomenon similarly to the analysis of storm-driven high water levels (Soomere et al., 2015).

A few outliers of the parameter $a$ in relatively sheltered coastal segments were negative and reached values down to –0.08 (Fig. 9). These values correspond to distributions with convex upwards shape in semilogarithmic coordinates and are thus qualitatively similar to the family of Gaussian or Weibull distributions.

### 3.3 Fitting the empirical distribution with theoretical distributions

Importantly, in about a quarter of the coastal segments in the study area, the parameter $a$ is positive and its 95% confidence intervals do not include the zero value. In other words, in each of such locations, the leading term $a$ of the quadratic polynomial $az^2 + bz + c$, estimated from the time series of wave properties for this location, is positive at a 95% significance level. Note that this estimate is valid for single points only and is not applicable for the set of such points. A positive leading term corresponds to the concave-up appearance of the relevant distributions of set-up heights for a certain range of these

heights. This means that large set-up events may be systematically much higher and/or occur much more frequently than one could expect from the classic Gaussian, Weibull or Poisson-type statistics. The described features indicate that the empirical distribution of set-up heights can be, at least locally, approximated using an inverse Gaussian (Wald) distribution with a probability density function (Folks and Chhikara, 1978)

$$P = \sqrt{\frac{\lambda}{2\pi z^3}} \exp\left[-\frac{\lambda(z-\mu)^2}{2\mu^2 z}\right].$$

(11)

For a certain set of parameters $\lambda$ (the shape parameter) and $\mu$ (the mean), a part of the graph of this function has a concave upward shape in semilogarithmic coordinates (Fig. 8). It thus fairly well approximates the empirical distributions of wave set-up at the relevant locations. For large values of $z$ this function behaves similarly to the probability density function of an exponential distribution and thus, in the above-mentioned notion, has a "heavy" tail and signals that large set-up heights are more frequent than for a Gaussian or Weibull process.

All coefficients of the quadratic approximation of the exponent vary insignificantly along the study area. This is remarkable because the shape of the relevant Weibull distribution (and thus the shape parameter of this distribution) for different wave conditions varies considerably in the study area (Soomere, 2005). The variations in the leading coefficient $a$ are uncorrelated with the values of maximum set-up heights in the study area. It is thus likely that the locations where an inverse Gaussian distribution governs the properties of set-up heights appear because of a specific match of the directional structure of winds

and the orientation of the coastline. This feature also signals that the basic features of the distribution are only weakly (if at all) connected with the properties of local wave climate. This conjecture is supported by comparatively small variations in the values of other parameters in the polynomial approximation (Fig. 9b,c). The values of $c$ are all positive and mostly in the range of 2.5–4 (Fig. 9). The values of parameter $b$ are, as expected, almost everywhere negative, concentrated around –0.2, typically varying between –0.1 and –0.4. A few locations with positive values of this parameter correspond to large negative

values of $a$.

To shed more light on which theoretical distribution is most suitable for the description of the probabilities of occurrence of wave set-up heights, we fitted the distributions using the *R* packages "fitdistrplus" for fitting any distribution to the data (Delignette-Muller and Dutang, 2005; v. 1.0-14), "actuar" (v. 2.3-3) for fitting with inverse Gaussian distribution, and "goft" (v. 1.3.4) to make an initial guess of the parameters of the inverse Gaussian distribution under *R* version 3.6.1. As the

probability density function of an inverse Gaussian distribution is everywhere positive, this fit can only be applied to positive values of wave set-up, that is, for the cases when waves are propagating towards the shore.

As an example of the appearance of the fit by different theoretical distributions, we start from the analysis of the data in the westernmost point of the study area presented in Fig. 8. The underlying data set contains 3724 (~6%) negative values out of 61720 entries. The fit was performed for the set-up values in the range from 0.01 to 0.4 m (Fig. 11). Attempts to fit the data

in question with an exponential, Gaussian and Weibull distributions (Fig. 11) demonstrate that these distributions do not replicate the location and height of the maximum and inaccurately follow the empirical distribution for larger set-up values. A variation in the higher and lower cutoff values leads to a certain change in the parameters of the fitted distribution change, but the inverse Gaussian fit remains appropriate for all cut-off values.

The goodness of fit of the cumulative distribution function (CDF) with different theoretical distributions is much better (Fig.

12). The empirical data do not follow a Gaussian distribution, whereas both Weibull and exponential distributions provide an acceptable match to the observed distributions. The inverse Gaussian shows the best agreement among the tested CDFs. As expected, the differences between all fitted distributions are relatively small for larger set-up heights but become more evident for the most frequently occurring set-up heights (Fig. 12).

We applied a Kolmogorov-Smirnov test to clarify which theoretical distribution describes the data in question at best. The

lower is the D-value, the smaller is the difference between the distributions. The corresponding D-values from this test (Kolmogorov-Smirnov statistic) are 0.039 for the inverse Gaussian distribution, 0.067 for the Weibull distribution, 0.120 for the exponential distribution, and 0.158 for the Gaussian distribution. Therefore, an inverse Gaussian distribution at best matches the empirical distribution of set-up heights at this location. However, the hypothesis that the data and random points from the fitted inverse Gaussian distribution are from the same distribution was rejected, even at an 80% confidence level.

However, the probability that the empirical distribution represents a data set with other tested theoretical distributions was much smaller. Therefore, even though there is no rigorous evidence that the empirical data follow an inverse Gaussian distribution, it provides the closest fit to the data compared with the other three tested distributions.

This proportion of the "remoteness" of all four distributions from the empirical probability distribution of set-up heights persists for the entire study area and different cut-off values (Fig. 13). The cut-off values were tested in the range from 40%

to 95% of the maximum set-up height in a particular coastal section. In terms of D-values of the Kolmogorov-Smirnov test, an inverse Gaussian distribution shows a better approximation to the empirical distribution of set-up heights than all other distributions. The Weibull distribution overall provides a slightly worse fit except for several (<7%) coastline locations (Fig. 13). An exponential distribution, even though it seems to follow the empirical distribution in most occasions closely, has a

smaller probability to describe set-up heights adequately. This feature apparently reflects the fact that an exponential distributions is a particular (one-parameter) case of the family of (two-parameter) Weibull distributions.

**4 Discussion and conclusions**

The analysis reveals that numerical estimates of maxima of wave set-up heights are relatively sensitive with respect to how
the impact of radiation stress and the transformation of wave properties in the nearshore are evaluated. The magnitude of the related effects substantially depends on the bathymetry. Refraction can easily override the purely geometric effects of shoreline orientation changes and redirect substantial levels of wave energy into seemingly sheltered shore sections. This means that high-resolution information about wind (including wind directions) and bathymetry, together with advanced methods for the evaluation of propagation and impact of radiation stress in the nearshore in operational and hindcast models
of coastal flooding, are required.

The core message from the analysis is that the empirical probability distribution of different set-up heights can usually be fairly well approximated by a standard exponential distribution $\exp(-\lambda z)$. When the exponent function of the general exponential distribution is approximated using a quadratic function, the coefficient of its leading term does not differ from zero at a 95% significance level for more than ¾ of the coastal segments of the study area. As the study area contains a
variety of sections with different orientations and with radically different wave properties, it is likely that the qualitative shape of the distribution only weakly depends on the properties of the local wave climate.

Another important message is that the basic shape of this distribution function is concave upwards in a log-linear plot for a substantial number of coastal segments. The local shape of the relevant empirical distributions of wave set-up heights can be adequately approximated with a family of inverse Gaussian (Wald) distributions. Similarly to the exponential distribution,
the probability density function of an inverse Gaussian distribution decays as $\exp(-\lambda z)$ for $z \gg 1$. Even though the absolute values of the coefficients of the leading term of such a quadratic approximation are relatively small, the goodness of fit with other classic distributions, such as Rayleigh or Weibull distributions (Fig. 8) that decay as $\exp(-\lambda z^2)$ for $z \gg 1$, is clearly worse. As the coefficient of the linear term of this quadratic approximation is relatively small (Fig. 9b), the use of a Lévy distribution might also be appropriate.
This result is intriguing because sensible approximations with inverse Gaussian (Wald) distributions are scarce in descriptions of geophysical phenomena. Perhaps the most well-known example of the use of a Wald distribution is to describe the time a Brownian motion (with positive drift) takes to reach a fixed positive level. Other examples include statistical properties of soil phosphorus (Manunta et al., 2002), long-distance seed dispersal by wind (Katul et al., 2005), and some models of failure (Park and Padgett, 2005).
The appearance of the distribution of modelled wave heights in the offshore (Fig. 3) is convex upwards in the range of relatively frequent wave heights of 0.5–1.7 m. It would thus be natural to expect that this property also becomes evident in

setup heights. The distribution of measured wave heights largely follows an exponential distribution for 1.2–3.2 m high waves and is only slightly convex upwards for 0.5–1.5 m high waves. This difference in the distributions for modelled and measured wave heights suggests that the presence of convex upwards distributions of set-up heights in nature may be even more pronounced than is demonstrated in Figs. 7 and 9. This difference also signals that the approximations employed to

evaluate wave properties are not responsible for the presence of convex upwards distributions of set-up heights. For example, a natural conjecture from Fig. 3 is that ignoring ice cover and the use of discontinuous wind data have at least partially supported the convex upwards shape of the distribution of wave heights.

Some of the introduced assumptions such as the ideal plane and rigid seabed, the presence of a dry coast without any vegetation, and ignoring the wave period and the particular value of the coastal slope (and thus wave reflection) in the

calculations are not fully realistic. In other words, it is assumed that all coastal segments are (i) favourable for the formation of high set-up and (ii) approximately homogeneous alongshore. These assumptions are only valid for selected segments of the study area. They all generally lead to an overestimation of set-up heights (Dean and Bender, 2006). As they impact set-up heights in the same manner, independently of wave properties, it is likely that they mainly stretch the resulting distributions of set-up heights towards larger values but do not modify their basic shapes. It might be expected that the

impact of other simplifications such as the assumption of monochromatic wave fields, using a constant value of the breaking index and employing long wave approximation for breaking waves, generally emphasizes the role of approach directions. Therefore, it is likely that the set of assumptions used makes the established features more noticeable than they would be for real wave fields.

Finally, we note that the presented results do not require any modification of the classic estimates of extreme values of set-up

heights and their return periods based on, for example, the block maximum method. Namely, the limiting distributions of independent block maxima follow either a Gumbel, Weibull or Fréchet distribution notwithstanding the distribution of the underlying values (Coles, 2004). This general theorem is obviously also valid for any time series that follows an inverse Gaussian distribution. A subtle implication from the qualitative match of statistics of set-up heights with an inverse Gaussian distribution is that set-up events with heights close to extreme heights may be much more frequent than their estimates based

on classic Gaussian or Weibull statistics and also clearly more frequent than similar estimates for Poisson processes. This increase in the probability of large wave set-up events is balanced by a similar decrease in the relative number of events with an average magnitude compared to normally or Weibull distributed events. The described features basically indicate that the frequency and role of close to extreme set-up events (and their contribution to damages and economic losses) may be underestimated based on observations of similar events of average height. In particular, severe set-up events may occur

substantially more frequently that could be expected from the probability of the occurrence of severe seas.

**Code availability**: From the authors on request (Matlab and *R* scripts).

**Data availability**: Kalbådagrund wind data are available from the Finnish Meteorological Institute.

Wave measurements: https://www.emodnet-physics.eu/map/platinfo/piroosplot.aspx?platformid=8974

**Sample availability:** not applicable.

**Authors team:** Tarmo Soomere, Katri Pindsoo, Nadezhda Kudryavtseva, and Maris Eelsalu,

**Author contribution.** T. Soomere designed the study, derived the equations and approximations used in the paper, produced Figure 5, compiled the introduction and discussion, checked the consistency of the results, and polished the text. K. Pindsoo developed the scripts, ran the simulations, produced most of graphics, and drafted the body parts of the manuscript. N. Kudryavtseva performed testing of fits of empirical data with different theoretical distributions, wrote the relevant parts of the text and prepared Figures 11–13. M. Eelsalu recalculated the graphics, performed a comparison of the properties of the reconstructed waves with measured waves near the study area, evaluated the distributions of set-up heights for time series of wave properties measured in the northern Baltic Proper, drafted the relevant parts of the manuscript, and produced updated maps.

**Competing interests:** The authors declare that they have no conflict of interest.

**Disclaimer:** All authors have approved the submitted version.

**Acknowledgements**

The research was financed by the institutional support of the Estonian Ministry of Education and Research (IUT33-3), the project "Sebastian Checkpoints – Lot 3 Baltic" of the call MARE/2014/09 and by the Estonian Research Council (ETAg) to two networks: the ERA-NET Rus+ network EXOSYSTEM and FLAG-ERA network FuturICT2.0. The authors are greatly thankful to the Finnish Meteorological Institute for making the Kalbådagrund wind data public, the Marine Systems Institute, Tallinn University of Technology, for providing wave data at Tallinnamadal, and to three anonymous referees for valuable suggestions towards improvement of the manuscript.

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

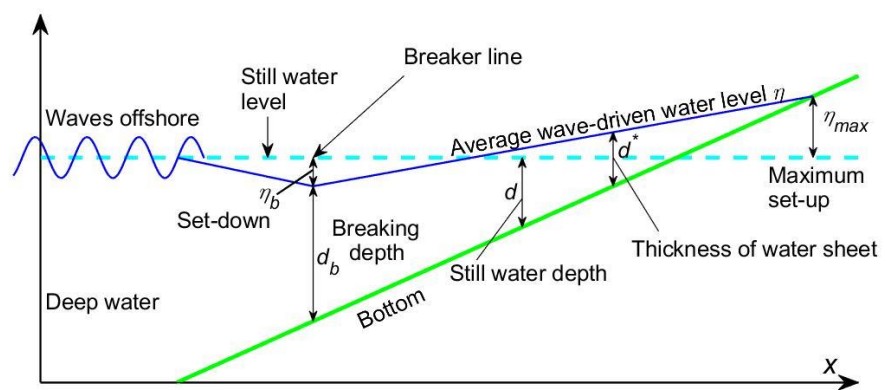

**Figure 1: Scheme of the cross-section of a coastal area with wave set-up. The sign of $\eta$ is positive if the average wave-driven water level exceeds the still water level and negative in the opposite case. The sign of $d$ is positive in the area covered by still water and negative in the otherwise dry section of the coast. The quantity $d*$ is non-negative.**

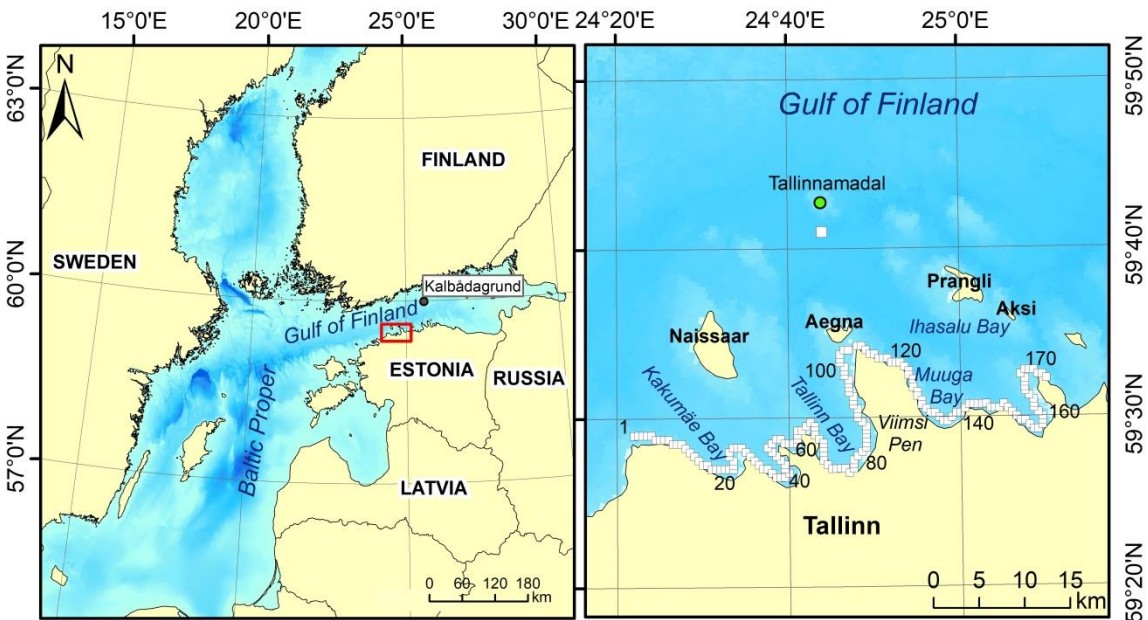

**Figure 2: Study area (red box in the left panel) in the vicinity of Tallinn Bay. Small circles along the shoreline in the right panel indicate the nearshore grid cells of the wave model WAM with a resolution of about 470 m. The grid cells are numbered consecutively from the west to the east. The green circle shows the location of the wave buoy at Tallinnamadal and the white square to the south of it the closest grid point of the wave model used for comparison of modelled and measured wave data.**

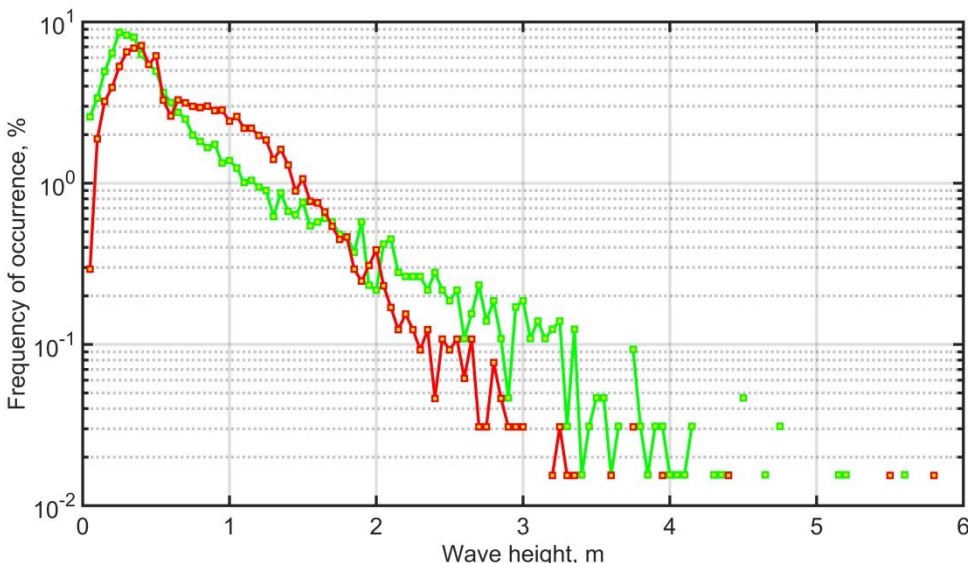

**Figure 3: Empirical probability distributions of measured wave heights at Tallinnamadal (green) and modelled wave heights in the closest grid cell (red) with a resolution of 1 cm in 2012–2016. The missing lines between the data points indicate gaps in the sets of frequencies.**

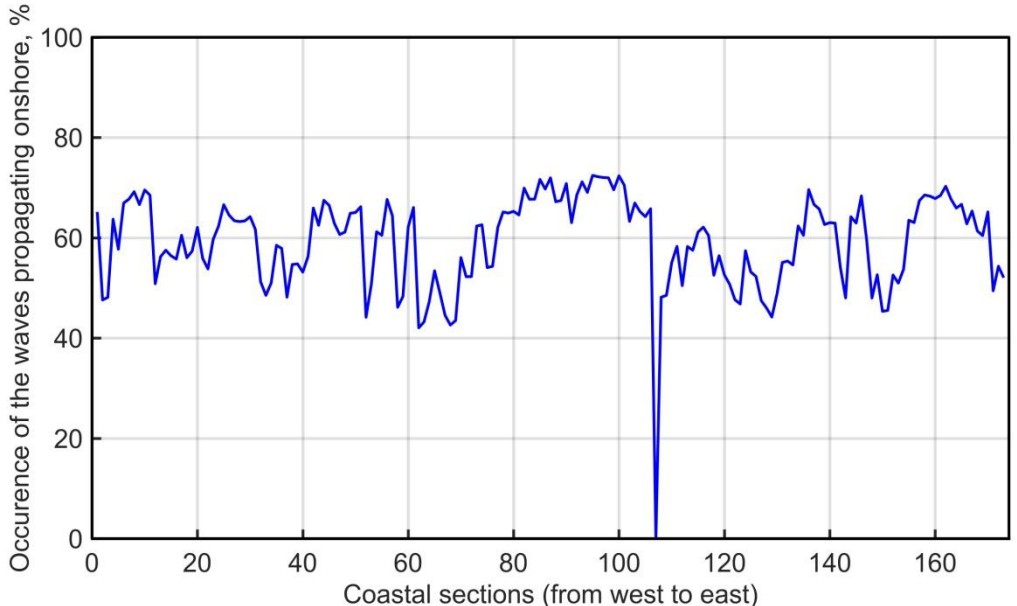

5    **Figure 4: The percentage of occurrence of waves that propagate onshore and produce elevated wave set-up events in the study area.**

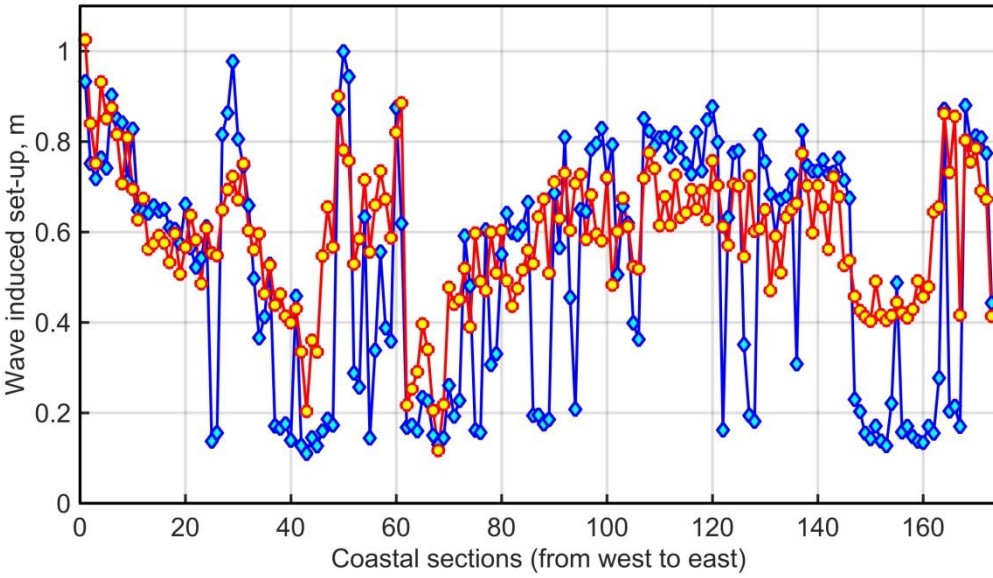

**Figure 5: Maximum set-up heights evaluated using all onshore-propagating wave fields and Eqs. (1–8) (red circles) and similar heights evaluated using only those waves that approach the shore at an angle less than ±15° from shore-normal (blue diamonds).**

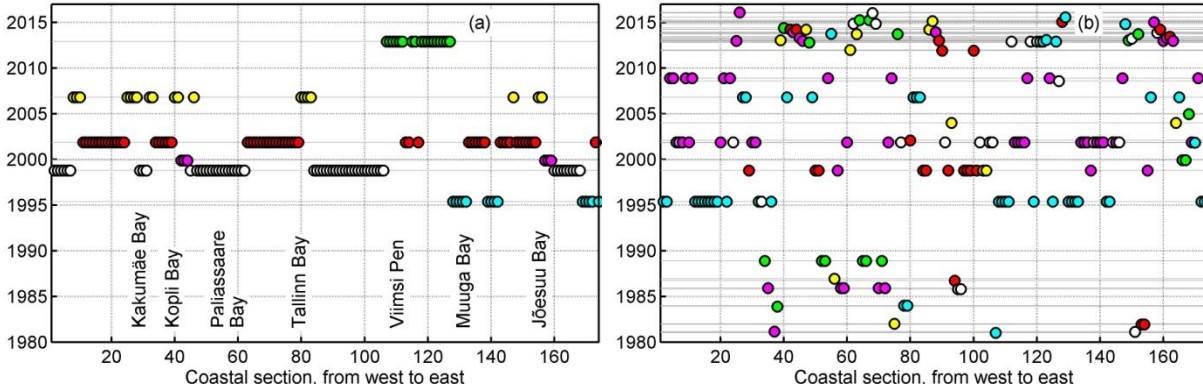

**Figure 6: (a) Six storms that caused the highest waves in different coastal sections of the study area in January 1981–May 2016. Notice the cluster of green circles along the eastern coast of Viimsi Peninsula in an autumn storm of 2013. (b) 58 storms that caused the highest wave set-up in these sections in January 1981–May 2016. The set-up heights are evaluated similarly to the procedure in (Pindsoo and Soomere, 2015) using only waves approaching at an angle ±15° with respect to shore-normal. Colours vary cyclically in time and correspond to different storms in single years.**

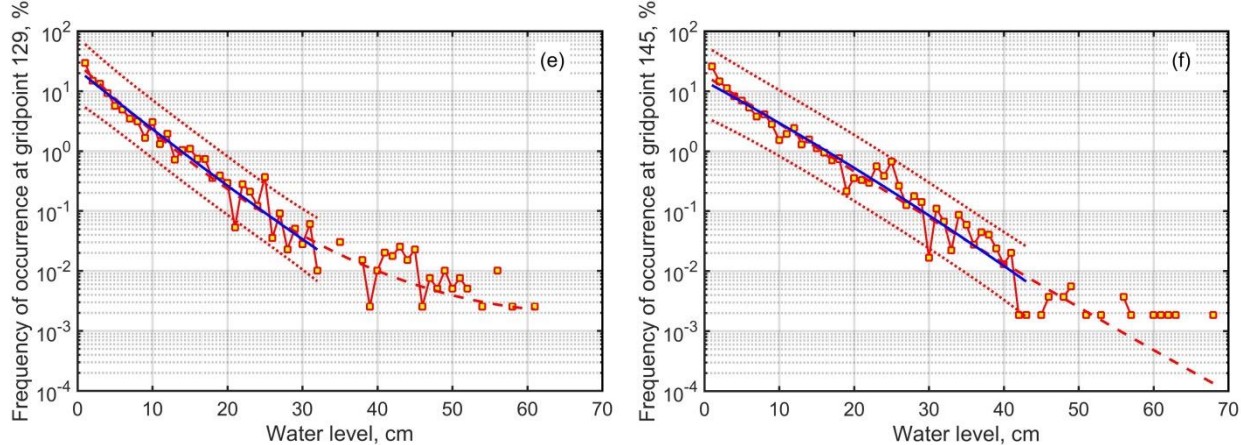

**Figure 7: Simulated distributions of various set-up heights (red squares) at various locations in the Tallinn Bay area open to different directions. Blue line: interpolation with a quadratic function from the set-up height of 0.01 m to the first gap in the empirical distribution; red dotted lines: its 95% confidence intervals; red dashed line: similar interpolation using all data points. The interpolating lines evaluated using only the data points from 0.01 m to 0.4 m are fully masked by blue lines. The resolution of all distributions is 1 cm.**

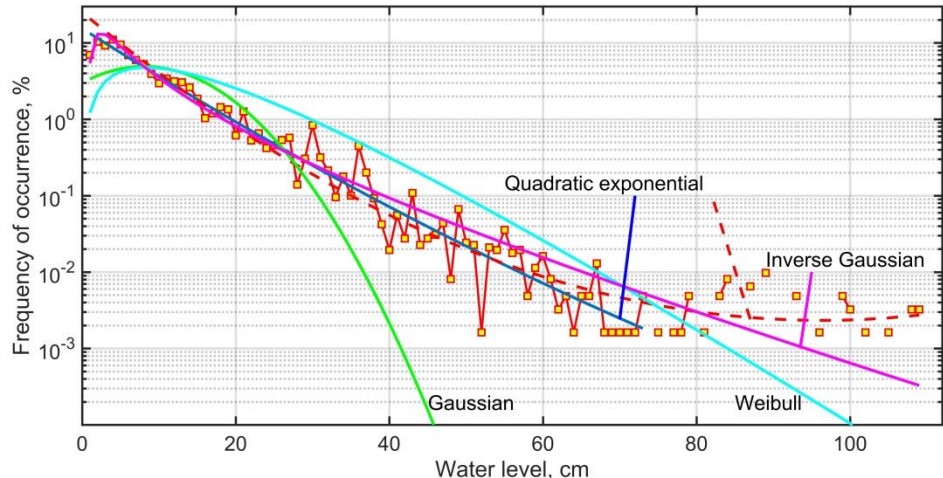

**Figure 8: Simulated distributions of various set-up heights (red squares) with a resolution of 1 cm in the westernmost coastal segment of the study area (grid point 1) and the relevant Gaussian, Weibull and inverse Gaussian (Wald) distributions evaluated using the method of moments. Blue line: interpolation of the empirical distribution in semilogarithmic coordinates with a quadratic function (equivalently, the formal local exponential distribution with a general quadratic exponent) from the set-up height of 0.01 m to the first gap in the empirical distribution; red dashed line: similar interpolation using all data points.**

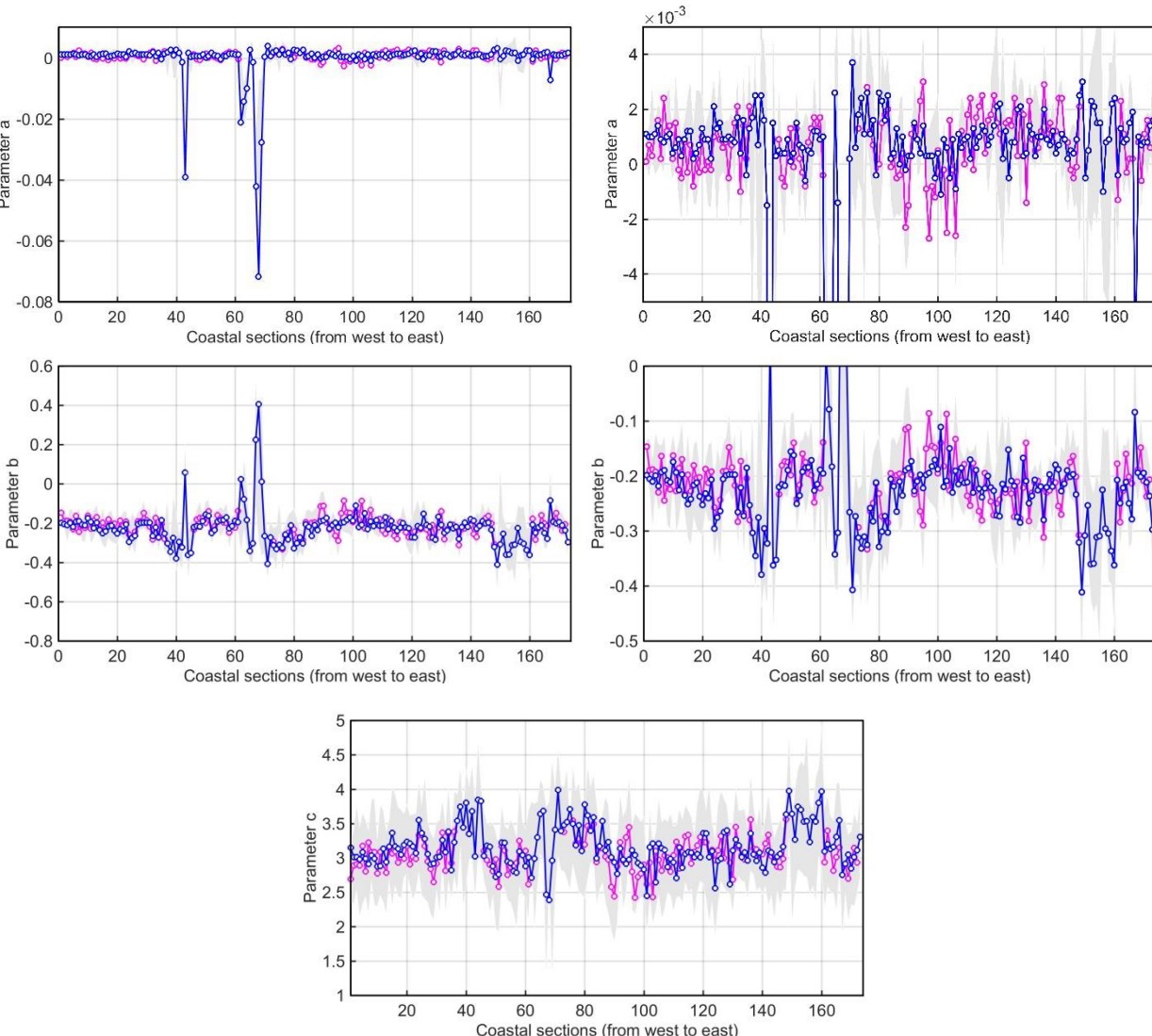

**Figure 9: Alongshore variation of the coefficients** *a, b, c* **of the quadratic approximation** $az^2 + bz + c$ **of the exponent of empirical distributions of set-up heights in the Tallinn Bay area. For the parameters** *a* **and** *b,* **more detailed alongshore variation is presented in graphs with vertically stretched scales. Blue line: the respective parameter calculated for the range of set-up heights from 0.01 m to the first gap in the empirical distribution; the grey area marks the 95% confidence interval of this value, the pink line describes the values of the relevant parameter for the range of set-up heights from 0.01 m to 0.04 m.**

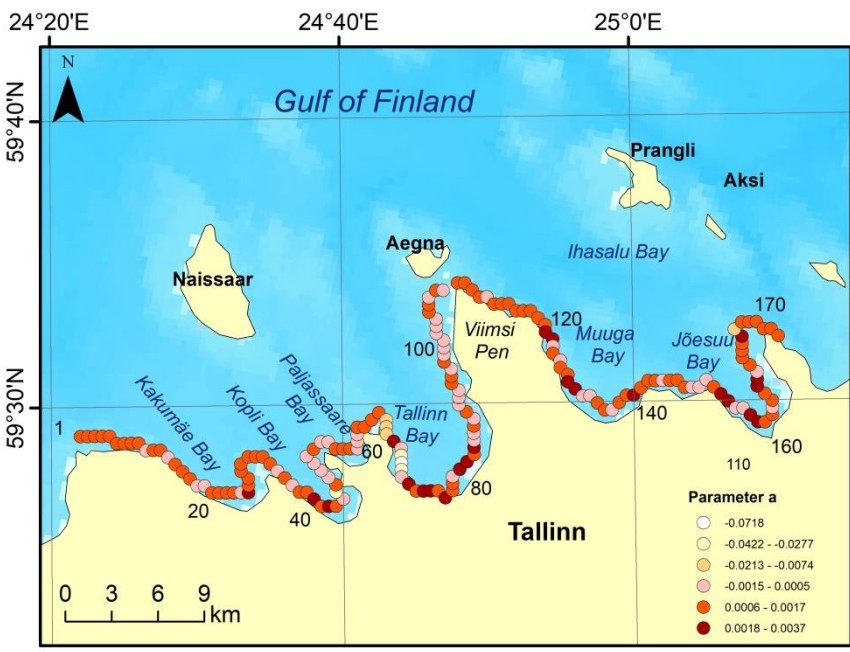

**Figure 10: Alongshore variation in the coefficient of the leading term (colour code) in the approximation of the exponent of empirical distribution of set-up heights at single locations. The grid cells are numbered consecutively from the west to the east.**

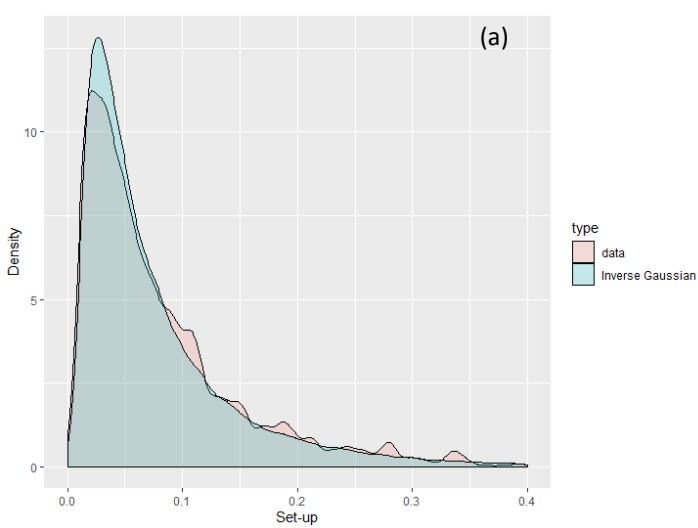

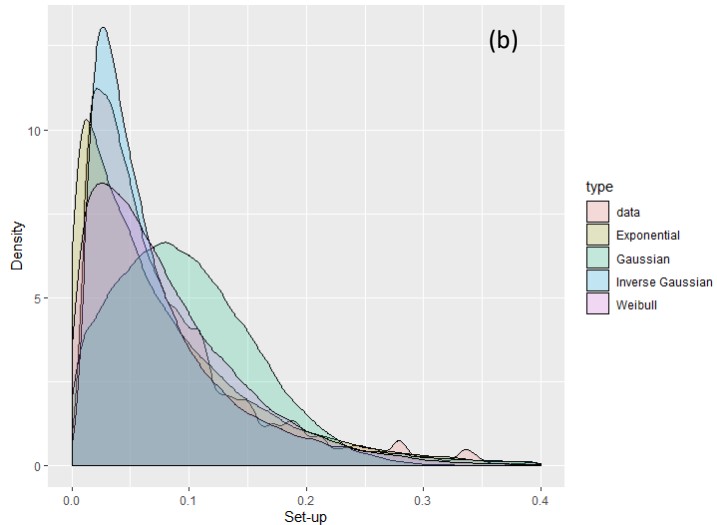

**Figure 11. (a) The best fit of the data set presented in Fig. 8 with an inverse Gaussian distribution with the mean** $\mu = 0.0804 \pm 0.0004$ **and shape** $\lambda = 0.0793 \pm 0.0005$ **and the best fit of the same data set with an exponential (rate 12.44±0.06), Gaussian (mean 0.0804±0.0003, std 0.0700± 0.0002) and Weibull (shape parameter 1.247±0.004, scale parameter 0.0867±0.0003) distribution.**

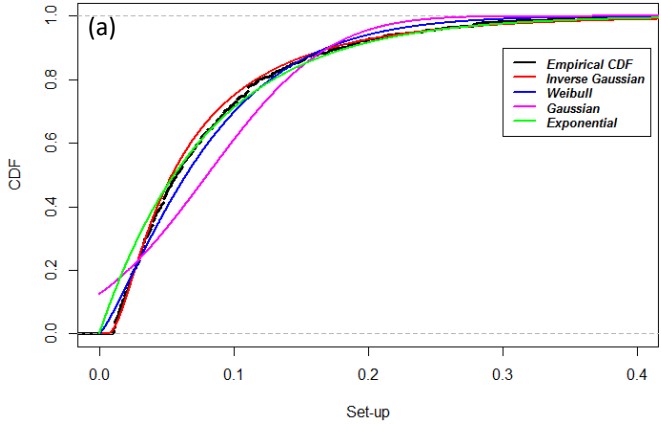

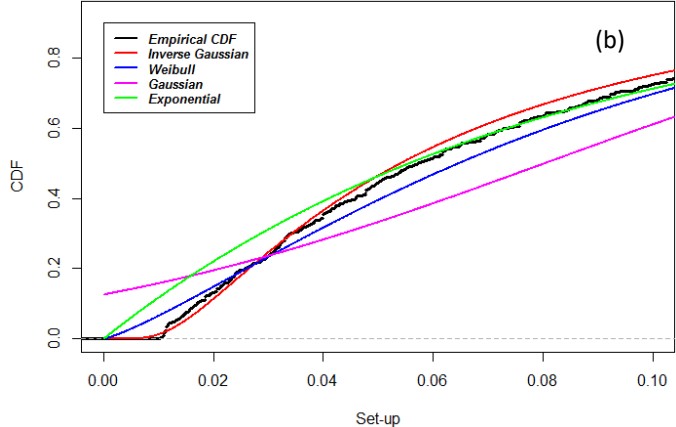

**Figure 12: (a) Match of the cumulative distribution function of the empirical distribution of occurrence of different set-up heights for the data presented in Fig. 8 with inverse Gaussian, Weibull, Gaussian and exponential distributions. (b) The same match for set-up heights <0.11 m.**

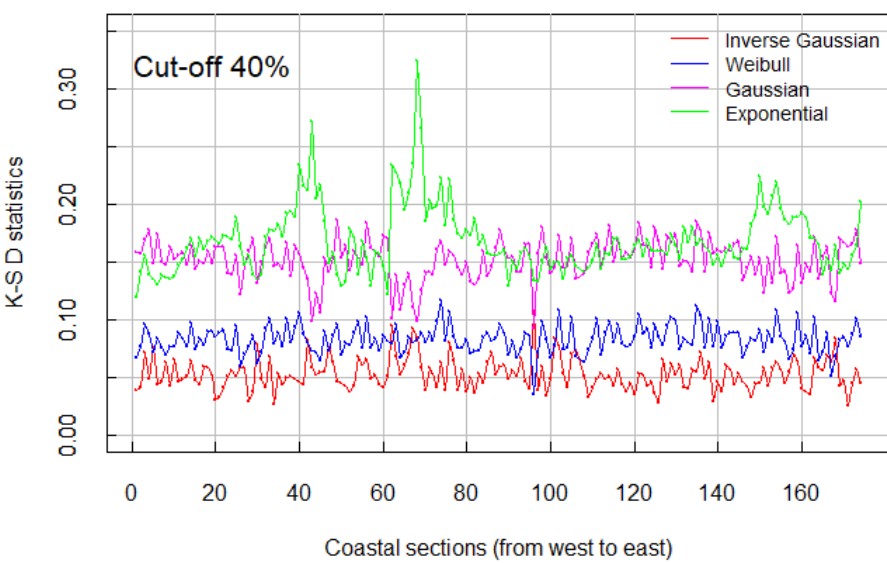

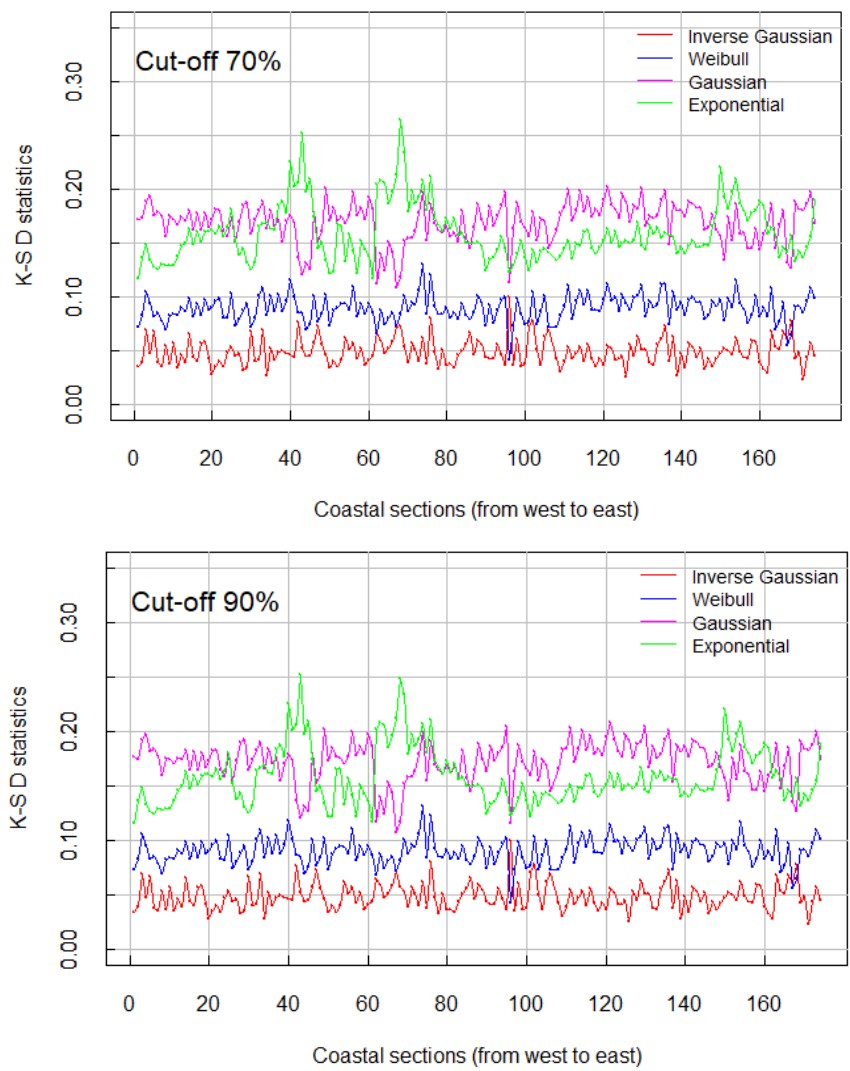

**Figure 13: Alongshore variation in the D-value of Kolmogorov-Smirnov test for the goodness of fit of the empirical probability distribution of set-up heights with an inverse Gaussian, Weibull, exponential, and Gaussian distribution for different cut-off values (40%, 70% or 90% of the maximum set-up height in a particular coastal section).**