# Peer review of "Variability of distributions of wave set-up heights along a shoreline with complicated geometry"

_Ocean Science, 2019_

## Referee Comment (RC1) · Anonymous Referee #1 · 3 Jun 2019

[a4paper,11pt]article [english]babel

Title: Variability of distributions of wave set-up heights along a shoreline with complicated geometry Author(s): Tarmo Soomere and Katri Pindsoo MS No.: os-2019-25 MS Type: Research article
**General comments**

The study is aimed to investigate the alongshore variability of the empirical statistical distribution of maximum wave set-up occurrence in a morphologically complicated situation. The area in exam is embedded in the Gulf of Finland, in the eastern Baltic Sea. The selected shoreline has been divided in very small segments, each containing the coastal grid output of a triply nested climatological run of the WAve Model. The maximum set-up has been calculated algebraically from the properties of the wave field at the breaker line, the water depth and the orientation of the shoreline in each segment for each member of the climatology. At each segment, the frequency of occurrence is then plotted against the simulated maximum wave set-up and a guadratic-exponent (three parameter) law is fitted to the data. In 3/4 of the segments the higher-order coefficient is found equal to zero at the 95% confidence level. In all the other cases, the leading quadratic coefficient is not null at the 95% confidence level, so a Wald (invert Gaussian) distribution is assumed. The method used for the evaluation of the wave set-up is fairly standard and consistent, the statistical analysis is rather on the gualitative side, but the results show a sort of internal coherence. There are in my opinion several problems in the study that should be addressed in order to improve the quality of the work. Some of the problems, listed as points in the specific comments, would probably require some further analysis on the data, some work on the figures and a general review of the text. Many weaker points are listed in the technical corrections.

**Specific comments**

1. In my opinion the main problem is in the results: it was found that 3/4 of the coastal points have an exponential distribution, while all the others have a completely different distribution. But there is no clear indication in the study about the reason behind it. There are some educated hypothesis, but no direct link is provided which can relate the type of distribution with some physical quantity like the angle of approach to the coastline, the wave climate or the bathymetry. This means there is no way to generalise the results outside the area of interest. As a result the work may have a distinct 'geographical' interest but is affected by a lack of 'physical' significance. Further effort should be devoted to understand the reason.

2. The value of the leading coefficient in the quadratic expression in the exponent of the distribution is found remarkably close to zero in all cases, but it is not zero at 95% confidence level in a large fraction of cases. 95% is pretty high but it is not a matter of faith. My impression is that the results would change significantly if different levels were chosen, i.e. 98% or 90%. The dependence of the results (in terms of the number of cases having exponential or else distribution) on an arbitrary choice would show a weakness of the method, indicating a lack of robustness in the statistical analysis.

3. By the way, I would have expected as a first tentative analysis the standard extreme value approach, using a fitting of the empirical CDF by means of plotting position functions. It is less subjective than the method used in the present study and has the added value to introduce a return period, which would be welcome in this case. With 3 parameters at disposal and plotting over a log-log scale, it takes a lot to discard a Weibull distribution. If all cases could be described with a similar distribution then the observation at 1. would be irrelevant.

4. Some of the panels in figures 6,7 show that the higher values of the set-up have the same probability. This is very odd, and led me thinking if there might be some problem with the independence of the data. It looks like the entire block of data would belong to the same storm. In the description of the methodology it should be described in detail
how the problem of the serial correlation of the data has been taken care.

5. The wind data gaps are a big problem if they are systematic in the upper percentiles. It should be taken care of in some way, and discussed in the conclusion.

6. Some figures are very hard to read, in particular figure 4. What is the rationale about the choice of the cases illustrated in figure 6,7?

7. There are some technical points which should be better explained, in particular the presence of 'data gaps in the distribution (the lowest set-up height that did not occur in 1981-2016)' at page 11. Is the statistical analysis in the range [0.01-0.4] really necessary? How the angle of incidence with the normal to the coastline was evaluated? How the phase and group velocities were estimated?

8. The language should be improved.

**Technical corrections**

There are many slightly inaccurate statements in the text which should be adjusted:

1 p.1 line 29: actual tides are not perfectly regular in many coastal areas (astronomical tides are).

1 p.2 line 16: 'neither completely independent nor completely dependent' does not give a lot of information.

1 p.2 line 19: significant wave heights. And it must be defined somewhere, because there is no definition in the manuscript. Add a symbol like  $H_{m0}$  if spectral. What is  $H_0$ ? Use it in all the manuscript consistently.

1 p.3 line 16: Normally instruments and model refers to statistical properties of wave fields: significant wave height, peak period, mean period and mean direction.

2.1 p.4 line 25: reference not found.

2.1 p.5 eqn (2): Here averaged eta is a function, it is customary to indicate the arguments in parentheses.
2.1 p.5 lines 5-6: the meaning of 'formal' is unclear, the choice of 'formal' and 'actual' is not particularly fortunate.

in Figure 1: introduce the axis - it is not obvious the sign of h and eta, introduce d and  $d^*$ .

2.2 p.6 line 8: Wave directions? wind?

2.2 p.6 line 9-10: Suggestion: it is possible to analyse the set-up for different values of forcing and wave propagation geometry.

2.2 p.6 line 13: Highest significant wave height is sufficient.

2.2 p.6 line 14-16: In meteorology it is customarily is to indicate wind coming from west as westerly, wind going toward west as westward. Eastern storm is unclear.

2.2 p.6 line 28: It is not actually the model implementation and it does not increase the efficiency of the model. It is a simplified method of reproducing the wave climate avoiding to processing all the time series.

2.2 p.7 line 6: That is an understatement. The wave simulation depends on wind, if the wind is not adequate the simulation is just noise.

2.2 p.7 line 6: 'In particular.' actually that is a completely different matter.

2.2 p.7 line 7: Someone might argue that wave directions and propagations in shallow waters and complex morphology depend more on bathymetry than on wind direction.

2.2 p.7 line 19: This is a huge problem for the statistical analysis. In my opinion every other choice (interpolation, replacement with model data, looking for other sources of data) would be better than simply not considering the data corresponding to gaps. see point 5.

2.3 p.7 line 21: Suggestion: in water depth >4 m

2.3 p.7 line 25: See note 2.2 p.6 line 14-16

2.3 p.7 line 28: Suggestion: oversimplified

2.3 p.7 line 31: How could significant wave height be monochromatic? 'As usual' is not enough to justify the assumption.

2.3 p.8 line 1: The mean wave direction provided by the model is not referred to the normal to the shoreline. This derivation must have been a successive operation which

OSD
should be described appropriately.

2.3 p.8 line 7: If  $H_b$  is a water level and  $H_0$  is a significant wave height they are different quantities having the dimension of length.

2.3 p.8 line 7: Missing reference

2.3 p.8 line 8: It should be explained how the phase speed and the group velocity were evaluated.

2.3 p.8 line 14-15: How the assumptions might affect the results? to be discussed in the conclusion.

2.3 p.8 line 14-15: But it is used in the successive section, isn't it?

3.1 p.9 line 5-15: See previous observation. The 'simpler method' is used throughout the manuscript: it should really be described better. Fig.4 is very hard to read but it gives the impression that the results are very different. The text is rather confusing and seemingly incoherent. One is tempted to understand that the set-up is greater for greater angles of incidence than for normal waves. It may be worth to observe that numerical statistical models like WAM are not able to deal with diffraction and reflection of waves.

3.2 p.10 line 7-10: It is not clear how to verify the statement, the analysis is rather qualitative and the figures describe only a very small part of the set of 174 segments. Suggestion: replication  $\rightarrow$  simulation

3.2 p.10 line 18: The discussion seems to exclude the possibility that somewhere in the whole region considered there could be a poisson process, which is contradicted by the results in p. 11 line 10-15.

3.2 p.10 line 24: Suggestion: approximation  $\rightarrow$  fitting procedure

3.2 p.10 line 28: 'Unexpected' does not explain the reason of the high values. On what basis the high values are assumed outliers? Looking at fig. 7, if the range of the setup is considered only in the range [0.01,0.4] maybe the distribution could have been 'forced' to be exponential. This part of the text is not sufficiently clear.

OSD

---

## Referee Comment (RC2) · Anonymous Referee #2 · 5 Jul 2019

The authors discuss the often less analysed, especially in climate science, but important topic of wave set-up heights and occurrence of these heights along a 60 km long part of the Estonian coastline at the Gulf of Finnland. Whereas the overall topic raised by the authors is of interest, the way it is presented make it hard for the reader to understand the overall purpose of this study. In the present form, I can not recommend to accept this manuscript. However, after addressing some major concerns by the authors I would suggest to accept the publication after major revisions.

General comments

The main criticism is that in the current form it is not clear what is the scientific question

they wanted to tackle in the first place and what is the main conclusion at the end of this work. Is it to establish some kind of climatology of wave set-up height in that area? Or is it to improve the general understanding of properties of the wave set-up? Can one translate the findings to other coastal regions, at least can one assume some? What is the reason to find an approximation of the empirical distribution? Also, the authors should put more emphasis in the manuscript, as one of the most cited and may be important references Soomere et al. 2013 is missing in the reference list.

Specific comments

If it is understood correctly they have used a single wind observation in favour of simulated wind information as forcing for the wave model. They argued that the simulated wind of atmospheric models are some times inaccurate and have some problems in representing the real wind fields. While this is certainly true, the analysis would benefit to present a short validation of the simulated wave field against some observation in the study area to show that a single forcing site is capable to represent the coastal wave climate reasonable.

Which Equation is exactly used for estimating the wave set-up height. They stated Eq (8) is used, but they also introduce the effect of shoaling and refraction is this afterwards used or neglected. Also, two Equations have the same reference number (9).

Some of the figures should also be refined. Figure 2 shows the model domain and the coastal segments with coloured dots that represent the value of a parameter of an estimated approximation. At this point this is too much information for the reader, one does not know what parameter "a" stands for. For instance, a colour coding, or a simple labelling of the number of the coastal segment would be of much more help later on, as in the other figures only the index number of the coastal section is displayed on the x-axis.

It is not clear what is the difference between the left and right side of figure 5, besides more storms are involved on one side. What is the colour coding for?

---

## Referee Comment (RC3) · Anonymous Referee #3 · 16 Jul 2019

Review comments for the manuscript "Variability of distributions of wave set-up heights along the shoreline with complicated geometry" in Ocean Science Discussions (os-2019-25)

The authors present wave set-up calculations based on pre-computed wave model results for a shoreline with complex geometry located in the Gulf of Finland. The wave set-up distributions are based on long time series (1981-2016) and they are calculated for separate segments along the coast. The wave set-up is approximated by accounting also for varying incident angles, and the approximation is compared to a more simple approach. The main finding is that while the wave set-up distribution is well

represented by a standard exponential distribution in most segments, in about 25% the distribution does not follow any standard distribution (Gaussian, Weibull, Rayleigh). At these locations the distribution is fairly well approximated by an inverse Gaussian distribution, meaning that the probabilities of the highest wave set-ups are higher with respect to the mean value compared to e.g. a Rayleigh distribution.

The paper is mostly well written and within the scope of Ocean Science. The results are also interesting. I can recommend that this paper is published in Ocean Science after major revisions. I have three main concerns. The first is that this paper is apparently a resubmission, but the results and conclusions have changed since the first paper, and I don't know what to make of this. The second concern is regarding the pre-calculated wave data. The third is how the distributions are presented and, to a degree, interpreted.

Main comment #1: The manuscript seems to be a resubmission of this paper: https://www.earth-syst-dynam-discuss.net/esd-2016-76/. Still, the previous version of the paper presents different results, suggesting that the relevant distribution is inverse Gaussian at all locations instead of just 25%. The papers seem to be based on the same data, so I don't know what has changed. The first version of the paper claims:

"The distribution of wave set-up heights matches a Wald (inverse Gaussian) distribution along the entire study area. Even though different sections of the study area are open to different directions and host substantially different wave regimes, the leading term of the exponent in the associated inverse Gaussian distribution varies insignificantly along the study area and generally is close to $-1$."

The current version claims:

"Even though different segments of the study area host substantially different wave regimes, the leading term of this polynomial is usually small (between $-0.005$ and $0.005$) and varies insignificantly along the study area. Consequently, the distribution of wave set-up heights substantially deviates from a Rayleigh or Weibull distribution (that

15 usually reflect the distribution of different wave heights). In about $\frac{3}{4}$ of occasions it is fairly well approximated by a standard exponential distribution. In about 25% of coastal segments it matches a Wald (inverse Gaussian) distribution. "

The authors should make it clear why their results have changed from the previous paper.

Main comment #2: It is not clear from the paper how the wave data has been obtained or how accurate it is. The main assumption is that the wave data in the nearshore areas reach a steady state quickly, which is why pre-computed maps can be used. I assume the maps are chosen based on the wind direction and the wind speed. Still, has the runs themselves been modelled using wind data from a numerical weather prediction model? If not, how have the Baltic Sea wide simulations been produced that provide the boundary data? Data from atmospheric models have been used by several studies in the Baltic Sea (including nearshore areas) and it is the industry standard. There is no reason to not produce the maps using proper wind data.

Are you using WAM 4.5.1 or a later version? Cycle 4.5.1 had a bug in the depth-induced wave breaking source term that lead to unphysical high values at certain points. It has been fixed in cycle 4.5.4. This must be ruled out as an explanation for your surprising results.

Using Kalbådagrund data to force nearshore areas will overestimate the wind speed and probably result in an inaccurate wind direction. The authors cite some of their previous work, but I think some more details are needed also in this paper. The validity of the modelling approach comes into question exactly because the authors present surprising results. Running a high-resolution model for 35 years is clearly a huge task. Creating the maps with a proper forcing is not. I think the Estonian Weather Service runs HIRLAM, which has been tried and tested in the GoF. A minimum requirement would be that the authors should identify the times that are responsible for creating the "surprising" high wave set-ups and validate the instances by running the "full model".

This will satisfy that, to the degree we can trust wave models, the surprising results are real.

Main comment #3: Usually the distributions are viewed and fitted by looking at the cumulative function, not the probability density function. The cumulative function is more stable, and I don't see any reason not to use it here. The comments regarding "gaps" in the distribution are also confusing. What is the physical meaning of these gaps? The "first gap" is controlled by the accuracy of the wave data (10 cm, 1 cm etc), so why is this a meaningful point? There is also no reason to expect that experimental/modelled data should be without these gaps, so the existence of one is not a defining point in the distribution.

I am also a bit worried about the highest points being correlated. Is this the case? Then one event that is overestimated might "destroy" the entire distribution. Are the highest values recorded during the time ice was present? This needs to be evaluated carefully. If you have found that the distribution of the wave set-up is really completely different for two neighrouring grid points about 500 m apart, then also give an example of the detailed shoreline structure that leads to such a dramatic variation so that the reader can be convinced what is going on and find some physical basis for this change in behaviour. Providing the structure of the shoreline that leads to this inverse-Gaussian behaviour would make this study highly useful for other coastal areas.

It is unclear how the second order polynomial fitting is done. It seems like you are fitting a polynomial to the data in the log-linear plot (is the variable p the probability density?). Please write the formula for the exponential distribution and how the polynomial fits into that, and for what fit this generalized exponential function becomes the "normal" exponential function. This will make it so much easier to follow the discussion and agree with the conclusions.

I will save minor comments for the revised manuscript.

---

## Author Comment (AC1) · 15 Aug 2019

We are thankful for the very detailed and professional insight of the Referee into our manuscript and the comments on the results that we consider interesting for the marine and coastal science community. The suggestions of the Referee are very much appreciated and following them will indeed improve the manuscript.

1. We agree that our results may reflect certain specific features of the study area and may not be directly extendable to other types of coast. One of the major conjectures is that the (empirical) probability density function of wave set-up heights may have an unusual shape in some coastal sections. This feature is obviously unlikely on relatively

straight and basically homogeneous shores where the properties of the local wave climate, refraction and shoaling change slowly along the shoreline, and therefore the properties of wave set-up are also mostly uniform in the alongshore direction. In the light of this comment we feel that an inclusion of some examples of the distribution in question, in typical Baltic Sea conditions (e.g., on the Baltic proper shores of Latvia and Lithuania) would clarify which kind of distribution of wave set-up heights is usual on the coasts of this water body. It is also a good idea to single out and describe in detail the features of coastal segments in which a Wald distribution of wave set-up emerges.

2. The core message of the manuscript is that wave set-up heights may follow a qualitatively different distribution from the "standard" ones that describe properties of other drivers of high local water levels and the reach of large waves (Gaussian for the water volume of the Baltic Sea, exponential for storm surges, Weibull for different significant wave heights, Weibull or Rayleigh for wave run-up). The wider problem here is that a comprehensive description of water levels would need the inclusion of one more type of distribution in the relevant analysis. We agree that the chosen level of statistical significance (that the leading coefficient of the quadratic approximation to the exponent is nonzero) is overexploited and does not guarantee that the distributions in question substantially deviate from exponential or Gaussian. We are happy to add estimates of the frequency of emergence of a Wald distribution that rely on different levels of statistical significance.

3. We intentionally focused on the analysis of the shape of probability density functions of set-up heights. A cumulative distribution function is, in essence, an integral over the probability density function (pdf) and thus potentially suppresses possible irregularities of the pdf. Our conclusions are based on the shape of this pdf for relatively frequently occurring set-up heights. The analysis discards the very large occurrences of this height. This approach is intentional because very large values are scarce (and thus the shape of the pdf has large uncertainty for these values) and because these values may follow another (generalized extreme value) distribution. We will expand the material to

include some examples of empirical cumulative distribution functions but we think that the analysis of extreme set-up heights is the subject of another study.

4. The highest values of set-up heights that have the same (very low) probability have occurred only once during the considered time period. As the entire simulation contains 103 498 single instants of wave properties, one occasion corresponds, theoretically, to $10^{-3}$%. As we exclude the cases with zero set-up heights (e.g., waves propagating offshore), the number of instants of wave properties varies between 40 000 and 70 000 for different coastal segments.

Most of the data points with the smallest frequency of occurrence thus correspond to one, two or three occasions of the relevant set-up height classes with a step of 1 cm. As strong wave storms usually last less than 5–6 hours in the Baltic Sea, different data points at a 0.001% or comparable level mostly represent different storms. The level of serial correlation of single wave set-up heights is implicitly minimized by using a non-traditional approach for the reconstruction of wave properties that is based on the sequence of wind properties once every 3 hours and contains a minimum amount of "memory" of wave fields. In the context of our analysis, the possibility of serial correlation should have no impact on the results as we focus on probability density functions and do not carry out any analysis in which serial correlation may have a role (e.g., sequences of events, block maxima or similar).

5. Wind data gaps are always a problem in reconstructions of marine hydrodynamic fields. As our conclusions rely on the shape of the probability density function for relatively frequently occurring set-up heights (and we even exclude the most infrequent, equivalently the highest, set-up events), it is likely that the impact of gaps in the recordings of the strongest winds have very little impact on our conclusions. We will explain this aspect in the revised manuscript.

6. Thank you for the comments. We shall try to reshape Fig. 4 for better readability. Figure 6 is meant to demonstrate how different are the shapes of various distributions

and where the basic difference between Gaussian, Weibull and Wald distributions is. The underlying set-up data are just for illustration (but still represent the most frequent case of exponentially distributed set-up heights). The panels of Fig. 7 are presented for coastal segments with different orientation. As numbers of coastal segments are hardly visible in Fig. 2, we will add a scheme with the location of these segments.

7. We meant gaps in the calculated empirical distributions, not in the time series of wind or wave properties. It is natural that some specific set-up heights in such empirical distributions (in our case with a resolution of 1 cm) simply do not occur. This happens for very large values of set-up heights that are populated by a few events. We interpret the presence of such a gap as showing that the number of occasions for the relevant set-up height (and for the higher values of set-up) is too small for the use in the estimates of the shape of the entire distribution.

On the one hand, it is generally necessary to use as wide a range of data points as possible of the probability density function in order to adequately evaluate its shape. On the other hand, the use of data points that correspond to very large and infrequently occurring events is questionable because these data points may have relatively large uncertainty (as they reflect only a few events) and, more importantly, they may follow another distribution (e.g., an extreme value distribution). For this reason we limited the values of set-up heights to 40 cm. As implicitly demonstrated in Fig. 6, higher than 40 cm set-up events do not occur at all in some coastal segments. If they occur, the number of such events is usually a few dozen, and only in a few segments exceeds 100. Therefore, such events form less than 0.1% of all set up events. In this context we would like to emphasize once more that our aim is to understand the basic properties (such as the shape) of the probability density function of set-up heights.

As discussed on p. 7–8, we employed the numerically evaluated wave properties from the standard WAM model. The angle of incidence of waves is evaluated based on the mean approach direction of waves at the centre of the model grid cell and a piecewise linear approximation of the shoreline. Doing so led to some problems at the ends of

some peninsulas that were poorly represented even at this resolution (470 m). We will describe this procedure in detail in the revised manuscript. The phase and group velocities were calculated from the standard finite-depth dispersion relation based on the peak period and water depth.

We also appreciate the list of technical corrections that we will address in full in the revised version. Also, we shall definitely consult with a native speaker, expert in the field, to bring the use of English to a much better level.
* * *

---

## Author Comment (AC2) · 15 Aug 2019

We very much appreciate the efforts of Referee #2 identifying weak points of the manuscript and for recommendations to improve its quality.

We agree that our formulation of the core question (or hypothesis) of the study and the main conclusion are not perfect. The basic aim is to find out which type of theoretical distribution describes the probability of occurrence of different wave set-up heights. As set-up may contribute up to 1/3 of the nearshore water level rise during strong storms, this question is of clear interest not only from the theoretical viewpoint but also for practical coastal management.

[Figure]

There has been extensive research into the statistical properties of other drivers of high local water levels and the reach of large water waves. The relevant distributions are very different: Gaussian for the water volume of the Baltic Sea, exponential for storm surges, quasi-Gaussian for water levels on the shores of the Baltic Sea, Weibull for different significant wave heights, and Weibull or Rayleigh for wave run-up. The knowledge of the shape and parameters of such a distribution is often crucial in various forecasts and management decisions. The process of wave set-up should be, to our understanding, no exception. Thus, it is necessary to determine the shape of its typical probability density function. This is, in essence, the precondition for building an adequate climatology of set-up heights.

When we started the analysis of set-up heights, the expectation was that the result would be one of the listed distributions. Surprisingly, a Wald distribution popped up in a non-negligible manner. This 'surprise' is apparently not properly represented in the introduction, and we will adjust the introduction, discussion and conclusions accordingly.

The problem of the translation of our results to other locations was also highlighted by Referee #1. We think that part of our results (namely, that set-up heights on the majority of occasions follow an exponential distribution) has universal relevance and would have justified an international publication on its own. The other major conjecture is that the (empirical) probability distribution function of wave set-up heights may have an unusual shape in some coastal sections. This feature might be specific to coastlines with complicated shape and strongly anisotropic wind and wave regimes. However, we think that additional efforts are necessary to find out why, when and where this type of probability distribution emerges. These questions are definitely important as a proper answer to them may lead to more adequate estimates of marine-induced risks in affected coastal segments but, to our understanding, are out of the scope of the current study.

We apologize for the missing bibliographic information in the reference list. It will definitely be added into the revised version.

As for specific comments, we agree that the use of one-point wind information and a simplified method of reconstruction of the wave properties in the study area is a fragile aspect of this manuscript. Even though we fully agree that an adequate reproduction of wave time series would add much confidence in our results, a detailed simulation of wave time series over several decades with a resolution suitable for the study area is an enormous task. However, there are numerous indications that even extremely simple parametric models forced by one-point winds lead to very reasonable reconstruction of time series of wave properties for several locations of the Estonian nearshore (Suursaar, 2010, 2013, 2015; Suursaar and Kullas, 2009). Our study basically relies on statistical properties of wave fields (the probability of occurrence of seas with different wave height, period, and direction) and, strictly speaking, does not require a reconstruction of time series. It is thus very likely that our approach, in particular the use of highly reliable wind information and relatively high-resolution wave model, properly represents statistical properties of wave fields. Thus, we will add a relevant comparison.

The mathematics of wave set-up is a complicated matter for waves that approach the shore under large incident angles and we probably compressed this material too much. In fact we use a sequence of equations. We start from Eq. (9) on page 8 to evaluate the changes to the wave properties owing to refraction and shoaling as waves propagate from the model grid point to the breaker line. This equation gives us properties of the wave field at the breaker line. Thereafter we employ Eq. (7) to find the set-up height. The derivation of this equation is provided because of intense discussion of the physics of breaking of obliquely approaching waves by Hsu et al. (2006) and Shi and Kirby (2008). We apologise for the presence of two equations (9). The second one on page 11 is not really used and is only presented to illustrate the functional form of the probability density function of a Wald distribution, and will be renumbered.

We are also thankful to the Referee for highlighting quality issues with Fig. 2 (which we shall redraw). The left panel of Fig. 5 shows the time when the all-time highest waves

occurred in the study area in 1981–2016 whereas the right panel of this figure shows the timing of the all-time highest set-up events. This figure is meant to explain that high waves in the nearshore of an area with complicated geometry do not necessarily lead to high wave set-up.

Hsu, T.-W., Hsu, J. R.-C., Weng, W.-K., Wang, S.-K., and Ou, S.-H.: Wave setup and setdown generated by obliquely incident waves, Coast. Eng., 53, 865–877, 2006.

Shi, F.Y., and Kirby, J.T.: Discussion of "Wave setup and setdown generated by obliquely incident waves" by T.-W. Hsu et al., Coastal Engineering, 53, 865–877, 2006, Coast. Eng., 55(12), 1247–1249, 2008, doi:10.1016/j.coastaleng.2008.08.001.

Suursaar, Ü.: Analysis of wave time series in the Estonian coastal sea in 2003–2014. Estonian J. Earth Sci., 64(4), 289–304, 2015, doi:10.3176/earth.2015.35.

Suursaar, Ü.: Locally calibrated wave hindcasts in the Estonian coastal sea in 1966–2011. Estonian J. Earth Sci., 62(1), 42–56, 2013, doi:10.3176/earth.2013.05.

Suursaar, Ü.: Waves, currents and sea level variations along the Letipea – Sillamae coastal section of the southern Gulf of Finland. Oceanologia, 52(3), 391–416, 2010, doi: 10.5697/oc.52-3.391.

Suursaar, Ü., and Kullas, T.: Decadal variations in wave heights off Cape Kelba, Saaremaa Island, and their relationships with changes in wind climate. Oceanologia, 51(1), 39–61, 2009, doi:10.5697/oc.51-1.039

---

## Author Comment (AC3) · 15 Aug 2019

We very much appreciate the comments of the Referee and are grateful for the overall positive attitude to the manuscript.

Main comment #1: The manuscript is a substantially revised version of our paper titled "Inverse Gaussian distribution of wave set-up heights along a shoreline with complicated geometry", originally submitted to Earth Systems Dynamics and available at https://www.earth-syst-dynam-discuss.net/esd-2016-76/.

In our cover letter we informed the editor of Ocean Science about the history of the

manuscript.

The Editor of Earth System Dynamics, Dr Anna Rutgersson mentioned in her final comment that "the presentation of the methods and results was not very clear" but encouraged some rewriting and a resubmission (which we understood as a hint that the paper could be better placed elsewhere).

The rejection from Earth Systems Dynamics was, in some ways, fortuitous. Namely, during the rewriting and additional check of the results we discovered a bug in the script for the calculation of set-up heights and for the subsequent evaluation of the parameters of their probability density function.

Removing this bug lead to the conclusion that typically (in about 75% of coastal segments of the study area) set-up heights follow a classic exponential distribution but still in about 1/4 of the cases they follow a Wald distribution. In this sense the conclusions have changed significantly and it might be sensible to retract the early version from the website of Earth Systems Dynamics. We are willing to do so if applicable and acceptable according to the relevant policy of Earth Systems Dynamics. Alternatively, we can discuss this issue in the current paper, with our apologies to the readers of the discussion version in Earth Systems Dynamics.

We also note that the manuscript under consideration has been submitted to Coastal Engineering but was rapidly returned because the editors felt that the results did not have clear direct applications for engineering.

Main comment #2: We admit that a weak point of this manuscript is the way we produce the wave data. A detailed simulation of the Baltic Sea wave time series over several decades with a resolution suitable for the study area (grid size less than about 500 m) is an enormous task. It is technically and computationally possible but, to our understanding, not strictly necessary in the context of our manuscript. The reason is that the presented results only depend on statistical properties of wave fields. The joint probability of occurrence of seas with different wave height, period, and direction
in the nearshore is already sufficient for estimates of probability density functions of wave set-up (provided the change in wave properties owing to refraction and shoaling is adequately represented). We understand that our approach does not guarantee that these properties of the wave climate are exactly represented; however, a reconstruction of time series of wave properties is not necessary to support the results presented in our manuscript. Even though our approach for wave modelling seems oversimplified, we still think that it makes it possible to reveal, perhaps in a relatively contrast manner compared to the real situation (because sea ice is ignored), the basic properties of the "climate" of wave set-up.

The representativeness of modelled wind data is still problematic in areas such as the Gulf of Finland. As it is an elongated water body, replication of wind directions is crucial. Research in the 2000s has indicated that atmospheric models nicely reproduced wind speed but the wind directions for a certain range of directions had systematic deviations from the measured directions (Ansper and Fortelius 2003). It is of course a question of judgment which data better represent reality.

Another, more serious problem is that coastal measurement stations on the southern coast of the Gulf of Finland (that is, in the study area), on many occasions, completely failed to represent offshore wind properties (Keevallik, 2003; Soomere and Keevallik, 2003). Even though the atmospheric models have become much better over the last 15 years and new physics, data sources and assimilation procedures have been added, reconstructions of older wind fields still rely on the same ground truth. It is therefore our view that the use of even high-quality modelled wind data for the research in this manuscript would basically replace one source of uncertainty with another.

The wave patterns have been calculated using several realisations of early versions of cycle 4 of the WAM model. They may have older versions of wave physics (in particular, they tended to overestimate wave heights in shallow areas) but, to our knowledge, they did not contain any major bugs. All nearshore grid cells are located at depths >4 m and most of them even in deeper than 8 m sea. Given the short wavelengths of severe

seas in the study area (usually <6 s), major problems with incorrect representation of wave properties in shallow areas are unlikely (albeit not totally excluded). To be on safe side in this respect, we will indicate the model water depth along with the presence of a Wald distribution.

As mentioned above, we fully agree that the use of alternative wind data would considerably strengthen the message of the manuscript and that the identification of the origin of outliers of set-up heights is necessary. However, we would like to stress that the shape of the probability density function is evaluated using relatively frequently occurring set-up heights. Moreover, our analysis explicitly discards all very large examples of this height. We only stress in the text that if the set-up heights really follow a Wald distribution, unexpectedly large values of set-up would be more probable than in the case of an exponential or Gaussian distribution.

Main comment #3: It was our intention to understand which type of theoretical distribution should be used for the description of set-up heights. For this reason we specifically analysed the shape of the probability density function as a clue to the answer. The problem is wider. Namely, other drivers of high local water levels and wave attack have very different distributions. Water volume of the entire Baltic Sea follows a Gaussian distribution, storm surges are Poisson-like processes reflected by an exponential distribution, different cases of significant wave height are likely to be Weibull (or Tayfun) distributed, and a Weibull or Rayleigh distribution usually works for wave run-up.

In this context it would be nice to demonstrate that set-up heights follow one of these. This is, however, not always the case. To our understanding, the most convincing way of proving that is to directly estimate the shape of the governing part of the relevant distribution. As this point was also raised by other Referees, we will include some information about the behaviour of the cumulative distribution functions.

The gaps in the calculated empirical distributions certainly depend on how the "classes" of the set-up height are defined. In a physical interpretation, the presence of such a

gap shows that the number of occasions for the relevant set-up height (and for the even higher values of set-up) is too small for the use of this value in the estimates of the shape of the entire distribution. In other words, we think that the distribution we are looking at has large uncertainty for set-up values that are higher than the first gap. Moreover, very large set-up heights may be parts of a population that follows some extreme value distribution. Visually, these very large values would all support the match with a Wald distribution. To avoid "false positive" detections of this distribution it is better to look at the distribution of relatively frequently occurring set-up heights. This is especially important in the log-linear representation where data points based on a few outliers may easily override the behaviour of data points that reflect thousands of cases.

The issue of potential serial correlation has also been touched on by Referee #2. Serial correlation is not a problem in our study. In particular, the use of 3-hourly wind data and the assumption that the memory of wave fields is short already removes most of the correlations between the highest set-up events. The typical duration of severe wave conditions in the study area is just a few hours. To maximally remove the impact of serial correlations, we limit the consideration of the shape of the probability density function to the relatively frequently occurring set-up heights.

We are thankful for the suggestion to describe in more detail the link between the geometry of the particular coastal segment with the presence of a Wald distribution. We shall definitely do so. We also apologize for the quite compressed manner of presentation of the background mathematics and the details of fitting, and will describe the procedure in more detail.

Ansper, I., and Fortelius, C.: Verification of HIRLAM marine wind forecasts in the Baltic, Publicationes Instituti Geographici Universitas Tartuensis 93, 195–205, 2003 [In Estonian].

Keevallik, S.: Possibilities of reconstruction of the wind regime over Tallinn Bay, Proc.

Estonian Acad. Sci. Eng., 9, 209–219, 2003.

Soomere, T., and Keevallik, S.: Directional and extreme wind properties in the Gulf of Finland, Proc. Estonian Acad. Sci. Eng., 9(2), 73–90, 2003.

---

## Author Response (AR1)

Dear Sir/Madam:

We sincerely thank the Referees for their detailed comments that greatly helped us to improve the manuscript. We have given careful consideration to all remarks and have expanded some sections of the paper. Also, we have carefully edited the entire text and removed several minor issues and typos.

5 The comments of Referees are represented using normal font below and our response and a description of corrections using italics.

Sincerely

Tarmo Soomere, Katri Pindsoo and Maris Eelsalu
03 November 2019

**Anonymous Referee #1** ()
The study is aimed to investigate the alongshore variability of the empirical statistical distribution of maximum wave set-up occurrence in a morphologically complicated situation. The area in exam is embedded in the Gulf of Finland, in the eastern Baltic Sea. The selected shoreline has been divided in very small segments, each
15 containing the coastal grid output of a triply nested climatological run of the WAve Model. The maximum set-up has been calculated algebraically from the properties of the wave field at the breaker line, the water depth and the orientation of the shoreline in each segment for each member of the climatology. At each segment, the frequency of occurrence is then plotted against the simulated maximum wave set-up and a quadratic-exponent (three-parameter) law is fitted to the data. In 3/4 of the segments the higher-order coefficient is found equal to zero at
20 the 95% confidence level. In all the other cases, the leading quadratic coefficient is not null at the 95% confidence level, so a Wald (invert Gaussian) distribution is assumed. The method used for the evaluation of the wave set-up is fairly standard and consistent, the statistical analysis is rather on the qualitative side, but the results show a sort of internal coherence.
There are in my opinion several problems in the study that should be addressed in order to improve the quality of
25 the work. Some of the problems, listed as points in the specific comments, would probably require some further analysis on the data, some work on the figures and a general review of the text. Many weaker points are listed in the technical corrections.
*Thank you for highlighting the weak points and, in particular, for suggestions to improve the manuscript.*

30 **Specific comments**
1. In my opinion the main problem is in the results: it was found that 3/4 of the coastal points have an exponential distribution, while all the others have a completely different distribution. But there is no clear indication in the study about the reason behind it. There are some educated hypothesis, but no direct link is provided which can relate the type of distribution with some physical quantity like the angle of approach to the coastline, the wave
35 climate or the bathymetry. This means there is no way to generalise the results outside the area of interest. As a result the work may have a distinct 'geographical' interest but is affected by a lack of 'physical' significance. Further effort should be devoted to understand the reason.
*We agree with this comment but still think that it is particularly important to identify and highlight the situations where the distribution of a certain phenomenon (wave set-up heights in our context) may follow a different*
40 *distribution than expected, even when the reasons behind the emergence of such a distribution are not fully clear. To our knowledge, there exist no systematic studies of the distribution of set-up heights. In this context, the core result is that this height in the majority of coastal sections follows an exponential distribution. To our understanding, this is a novel result that expands our knowledge of nearshore processes. This is already interesting as the heights of single waves and the time series of significant wave heights usually follow a Rayleigh*

*or Weibull distribution. The applicability of an exponential distribution for the description of set-up heights seems to be a universal feature as it also becomes evident when we employed time series of offshore measured wave properties for such calculations. The emergence of another distribution in some coastal segments is, to our understanding, a far more interesting feature, for which we have no good explanation as yet. However, as it emerges on many occasions, we think it is important to at least mention its presence and the possible implications of its presence. We have reformulated some sections of the manuscript so that this line of thinking becomes clearer.*

2. The value of the leading coefficient in the quadratic expression in the exponent of the distribution is found remarkably close to zero in all cases, but it is not zero at 95% confidence level in a large fraction of cases. 95% is pretty high but it is not a matter of faith. My impression is that the results would change significantly if different levels were chosen, i.e. 98% or 90%. The dependence of the results (in terms of the number of cases having exponential or else distribution) on an arbitrary choice would show a weakness of the method, indicating a lack of robustness in the statistical analysis.

*Yes, the main conclusion of the study is that with quite a high confidence, one can use an exponential distribution to approximate the distribution of set-up heights. The use of the 95% confidence level follows a common line of thinking that this threshold is a good indicator for saying that certain things have some property that is statistically significant. The use of 90% level of statistical significance would lead to an about twofold increase in the number of sections with nonzero coefficient at the quadratic term (in terms of statistical significance at this level) and the use of 98-99% level would render almost all but 6–7 segments following an exponential distribution of set-up heights. This variation is, however, an intrinsic feature of the application of this type of statistical analysis but would still lead to the conclusion that in a few locations the distribution of set-up heights seems to follow an inverse Gaussian distribution.*

3. By the way, I would have expected as a first tentative analysis the standard extreme value approach, using a fitting of the empirical CDF by means of plotting position functions. It is less subjective than the method used in the present study and has the added value to introduce a return period, which would be welcome in this case. With 3 parameters at disposal and plotting over a log-log scale, it takes a lot to discard a Weibull distribution. If all cases could be described with a similar distribution then the observation at 1. would be irrelevant.

*We agree that this analysis would be interesting and important; however, our goal here is not to look at extreme values. The main point of Section 3 about maximum set-up heights is to compare the results obtained using the simplified approach of Soomere et al. (2013), with those calculated using the approach in this paper. We intentionally explain the differences in terms of maximum set-up heights because the differences in mean set-up heights are much smaller.*

*The underlying theorems (e.g., Coles, S.: An introduction to statistical modelling of extreme values. Springer, 3rd printing, Springer, London, 208 pp., 2004) state that the distribution of extreme values (e.g., the block maxima) is independent of the particular distribution of the underlying time series (e.g., of set-up heights) under a quite general set of conditions. Therefore, establishing the (parameters of the relevant generalised) extreme value distribution provides no information about the underlying distributions (which we try to quantify). This aspect is explained in the Discussion section. However, we shall definitely perform this kind of exercise in the future.*

4. Some of the panels in figures 6,7 show that the higher values of the set-up have the same probability. This is very odd, and led me thinking if there might be some problem with the independence of the data. It looks like the entire block of data would belong to the same storm. In the description of the methodology it should be described in detail how the problem of the serial correlation of the data has been taken care.

*The reason for repeating values of probabilities is that the highest values of set-up heights that have the same (very low) probability have occurred only once or twice during the considered time period. As the underlying wind data set has a time step of 3 hours and usually extreme wave storms last no more than 5–6 hours in the Gulf of Finland, it is unlikely that any set of very large set-up values would belong to the same storm. As the entire simulation contains 103 498 single instants of wave properties, one occasion corresponds, theoretically, to $10^{-3}$%. As we exclude the cases with zero set-up heights (e.g., waves propagating offshore), the number of instants of wave properties varies between 40 000 and 70 000 for different coastal segments. Consequently, one occasion in such segments corresponds to $2–3x10^{-3}$%.*

*The level of serial correlation of single wave set-up heights is implicitly minimized by using an approach for rapid reconstruction of the wave climate. It is based on the sequence of wind properties once every 3 hours and contains a minimum amount of "memory" of wave fields. Moreover, in the context of our analysis, the possibility of serial correlation should have no impact on the results as we focus on probability density functions and do not carry out any analysis in which serial correlation may have a role (e.g., sequences of events, block maxima or similar).*

5. The wind data gaps are a big problem if they are systematic in the upper percentiles. It should be taken care of in some way, and discussed in the conclusion.

*We agree that gaps in wind data are always a problem in reconstructions of marine hydrodynamic fields. As our conclusions rely on the shape of the probability density function for relatively frequently occurring set-up heights (and we even exclude the most infrequent, equivalently the highest, set-up events from some parts of the analysis), it is likely that the impact of gaps in the recordings of the strongest winds have very little impact on our conclusions. We explain this aspect in the revised manuscript.*

6. Some figures are very hard to read, in particular figure 4.

*It is true that we tried to pack too much information into a small space but we did not find a better way to present this figure (now Fig. 5).*

What is the rationale about the choice of the cases illustrated in figure 6,7?

*Figure 6 (now 8) is meant to demonstrate how different the shapes of various distributions are, and where the basic difference between Gaussian, Weibull and Wald distributions is. The underlying set-up data are just for illustration (but still represent the most frequent case of exponentially distributed set-up heights).*

*The panels of Fig. 7 are presented for coastal segments with different orientation. As numbers of coastal segments are hardly visible in Fig. 2, we added a scheme with the location of these segments.*

*We reshaped and expanded the presentation of the information on Figs. 6, 7 (now 7, 8) at the beginning of Section 3.2 and edited it for clarity so that Figures 6 and 7 (now 7 and 8) are interchanged in the revised manuscript. Also, it is now made clear that the main outcome of the analysis is that generally an exponential distribution matches the empirical distributions of set-up heights.*

7. There are some technical points which should be better explained, in particular the presence of 'data gaps in the distribution (the lowest set-up height that did not occur in 1981-2016)' at page 11.

*We meant gaps in the calculated empirical distributions, not in the time series of wind or wave properties. It is natural that some specific set-up heights in such empirical distributions (in our case with a resolution of 1 cm) simply do not occur. This happens for very large values of set-up heights that are populated by a few events. We interpret the presence of such a gap as an indication that the number of occasions for the relevant set-up height (and for the higher values of set-up) is too small for use in the estimates of the shape of the entire distribution.*

Is the statistical analysis in the range [0.01-0.4] really necessary?

*Yes, because our aim is to understand the basic properties (such as the shape) of the probability density function of set-up heights. To avoid possible distortions by the presence of a few very large set-up heights (that may have a completely different distribution as discussed above), we perform in parallel the analysis of the shape of the distribution based only on well-defined values of this distribution. The possible difference in the results is implicitly demonstrated on Fig. 7 (Fig. 6 in the original manuscript) where the inclusion of large set-up values "bends" the approximation towards a concave upwards shape.*

*Higher than 40 cm set-up events do not occur in some coastal segments. If they occur, the number of such events is usually a few dozen, and only in a few segments exceeds 100. Therefore, such events form less than 0.1% of all set up events but their presence may considerably modify the shape of the approximation to the empirical distribution of set-up heights.*

How the angle of incidence with the normal to the coastline was evaluated?
*We employed the numerically evaluated wave properties from the standard WAM model. The angle of incidence of waves is evaluated based on the mean approach direction of waves at the centre of the model grid cell and a piecewise linear approximation of the shoreline. This is now described in more detail in the revised manuscript*

How the phase and group velocities were estimated?
*The phase and group speed at the wave model grid cell were estimated based on the standard expressions of linear theory using the wave number $k$ evaluated from the dispersion relation for linear monochromatic waves $2\pi/T_p = \sqrt{gk \tanh kh_0}$ with the period equal to the peak period $T_p$ and the water depth $h_0$ equal to the model depth for the particular cell. The phase and group speed at the breaker line were estimated from the dispersion relation for long waves: $c_{gb} = c_{fb} = \sqrt{gh_b}$ . This is now described in the manuscript.*

8. The language should be improved.
*We have asked a colleague-native speaker to edit the manuscript.*

**Technical corrections**

There are many slightly inaccurate statements in the text which should be adjusted:
1 p.1 line 29: actual tides are not perfectly regular in many coastal areas (astronomical tides are).
*Thank you; our remark was not necessary, and we deleted it and adjusted the sentence as follows: „A reasonable forecast of the joint impact of tides, …"*

1 p.2 line 16: 'neither completely independent nor completely dependent' does not give a lot of information.
*This formulation has been replaced by the following: „On most occasions severe coastal flooding occurs under the joint impact of several drivers"*

1 p.2 line 19: significant wave heights. And it must be defined somewhere, because there is no definition in the manuscript. Add a symbol like Hm0 if spectral. What is H0? Use it in all the manuscript consistently.
*Thank you; we have followed this recommendation and explained after the definition of significant wave height that the rest of the manuscript uses simply „wave height", having in mind significant wave height.*

1 p.3 line 16: Normally instruments and model refers to statistical properties of wave fields: significant wave height, peak period, mean period and mean direction.
*Yes, we adjusted the expression accordingly*

2.1 p.4 line 25: reference not found.
*Thank you, it is added.*

2.1 p.5 eqn (2): Here averaged eta is a function, it is customary to indicate the arguments in parentheses.

*Yes, η is, technically, a function of four explicit arguments (h, h_b, theta_b, gamma_b) and one implicit argument (a spatial coordinate, say, x) via h(x). As its particular values are not used in the further derivation and η itself is eliminated from the final expression, it seems to us that showing explicitly these arguments would make the equation overly complicated to read, so we only added remark that it depends on the location x along the coastal profile. Also, as the overbar is really not necessary, we omitted it from the formulas.*

2.1 p.5 lines 5-6: the meaning of 'formal' is unclear, the choice of 'formal' and 'actual' is not particularly fortunate.

*Yes, indeed; we have reformulated these expressions.*

In Figure 1: introduce the axis - it is not obvious the sign of h and eta, introduce d and d*.

*Thank you. We have replaced the entire figure, inserted both axes, and made clear how the variables are defined. The signs of the variables are explained in the figure caption. The use of H for wave heights and h for depths (thickness of water layer) may be confusing indeed, so we have changed notations so that „d" is showing the depth.*

2.2 p.6 line 8: Wave directions? wind?

*An extra „the" distorted the point.*

2.2 p.6 line 9-10: Suggestion: it is possible to analyse the set-up for different values of forcing and wave propagation geometry.

*We plan to extend the analysis for the entire Baltic Sea shore with a somewhat lower resolution (perhaps 1-2 nautical miles) to determine how many of the results are locally confined.*

2.2 p.6 line 13: Highest significant wave height is sufficient.

*Thank you.*

2.2 p.6 line 14-16: In meteorology it is customarily is to indicate wind coming from west as westerly, wind going toward west as westward. Eastern storm is unclear.

*Thank you, it is now formulated in an unambiguous manner*

2.2 p.6 line 28: It is not actually the model implementation and it does not increase the efficiency of the model. It is a simplified method of reproducing the wave climate avoiding to processing all the time series.

*Thank you; this is exactly what we are doing, and now we say so.*

2.2 p.7 line 6: That is an understatement. The wave simulation depends on wind, if the wind is not adequate the simulation is just noise.

*Yes, of course, we deleted it.*

2.2 p.7 line 6: 'In particular..' actually that is a completely different matter.

*Deleted.*

2.2 p.7 line 7: Someone might argue that wave directions and propagations in shallow waters and complex morphology depend more on bathymetry than on wind direction.

*Yes, we added a short comment on this. In fact, we would like to have as accurate as possible information about both wave directions and heights offshore (at the selected wave model grid cells). We assume that the further propagation of the waves over ~500 m long stretch to the breaker line depends almost entirely on the local bathymetry and geometry, and calculate it as exactly as we can (and ignore wind input over this stretch).*

2.2 p.7 line 19: This is a huge problem for the statistical analysis. In my opinion every other choice (interpolation, replacement with model data, looking for other sources of data) would be better than simply not considering the data corresponding to gaps. See point 5.

*We agree that this is crucial in all applications where, e.g., the time series, maximum events or worst-case scenarios are addressed. We focus exclusively on the shape of empirical distributions of set-up height. We deliberately avoid consideration of the „tails" of these distributions that are based on a relatively small number of time instants and could be affected by missing data. Moreover, we actually look at how the shape of these distributions changes along the study area. This shape is mostly governed by small and „intermediate" values of wave and set-up heights that are all represented by a large number of time instants. In this sense it is likely that our results are quite insensitive with respect to a relatively large number of missing entries in the wind data set. This conjecture is confirmed by a comparison of modelled and measured wave statistics for 2012–2016 at the end of Section 2.2.*

2.3 p.7 line 21: Suggestion: in water depth >4 m
*Thank you, it makes the expression more transparent*

2.3 p.7 line 25: See note 2.2 p.6 line 14-16
*Thank you, it is now formulated in an unambiguous manner*

2.3 p.7 line 28: Suggestion: oversimplified
*Thank you; this is better indeed.*

2.3 p.7 line 31: How could significant wave height be monochromatic? 'As usual' is not enough to justify the assumption.
*We speak about „numerically evaluated wave field for each time instant [that is] is monochromatic". We reshaped the formulation. Also, we make clear now that the approach & assumptions largely follow those made in (Soomere et al., 2013).*

2.3 p.8 line 1: The mean wave direction provided by the model is not referred to the normal to the shoreline. This derivation must have been a successive operation which should be described appropriately.
*Thank you for highlighting this gap. We have amended the description: „.The approach direction $\theta_0$ with respect to the onshore-directed normal to the shoreline is calculated from $\theta$ based on an approximation of the relevant (about 500 m long) coastal segment by a straight line that follows the average orientation of the shoreline in this segment."*

2.3 p.8 line 7: If Hb is a water level and H0 is a significant wave height they are different quantities having the dimension of length.
*Hb is the wave height at the breaking line. Water level was denoted using lowercase h, so we think the expressions are fine. Still, we have denoted water level by „d" in the revised manuscript.*

2.3 p.8 line 7: Missing reference
*Our apologies; it is added to the reference list.*

2.3 p.8 line 8: It should be explained how the phase speed and the group velocity were evaluated.
*Everything was done using the expressions from standard linear wave theory. The steps are now described in more detail.*

2.3 p.8 line 14-15: How the assumptions might affect the results? to be discussed in the conclusion.
*It is now discussed in the concluding section.*

2.3 p.8 line 14-15: But it is used in the successive section, isn't it?

*As the assumption of ignoring waves that approach the shore at large angles was used in several earlier papers, we think it is sensible to discuss in the body of the paper how strongly and where it may affect the results in terms of maximum set-up heights.*

*Also, we removed the remark on p.8, lines 15-16 of the original manuscript about the isolation of the worst*
5 *possible scenarios as this aspect was not addressed in this manuscript.*

3.1 p.9 line 5-15: See previous observation. The 'simpler method' is used throughout
the manuscript: it should really be described better.

*We replaced the „simpler method" by „S2013" and defined it as „the approach [used in] (Soomere et al., 2013)"*

10 Fig.4 is very hard to read but it gives the impression that the results are very different.

*The results are somewhat noisy indeed; however, this unfortunately often happens when (wave) maxima are compared. One point of Fig. 4 is to show that the approach used in earlier studies led to fairly large uncertainties (and possibly incorrect values for some coastal areas) and that the use of the entire range of approaching waves and more adequate approximations (here Eqs. 1–8) will make a clear difference for such areas. Another point is*
15 *that in some 80% of the segments in question, the maximum values are almost insensitive with respect to the ignoring of waves that approach the shore at angles larger than 10-15 deg.*

The text is rather confusing and seemingly incoherent. One is tempted to understand that the set-up is greater for greater angles of incidence than for normal waves.

*We substantially reshaped the text so that is hopefully clearer and more consistent now. We also tried to make*
20 *clear that if the highest set-up heights are created by waves with greater angle of incidence, then this feature reflects the absence of high waves that approach the shore at small angles.*

It may be worth to observe that numerical statistical models like WAM are not able to deal with diffraction and reflection of waves.

*Thank you, the original text mixed up reflection and refraction.*

25 3.2 p.10 line 7-10: It is not clear how to verify the statement, the analysis is rather qualitative and the figures describe only a very small part of the set of 174 segments.

Suggestion: replication → simulation

*Our original formulation was deceptive indeed and has been corrected; see also reply to the next observation.*

3.2 p.10 line 18: The discussion seems to exclude the possibility that somewhere in the whole region considered
30 there could be a Poisson process, which is contradicted by the results in p. 11 line 10-15.

*Thank you; one of our main points is that for some ¾ of cases an exponential distribution (equivalently, a background Poisson process) serves as an acceptable fit. We added the sentences: „A feasible option for matching the empirical distribution in the majority of the coastal segments is an exponential distribution that reflects a background Poisson process. This distribution is represented by a straight line in log-linear*
35 *coordinates used in Fig. 6, 7 and also describes, e.g., storm surges in the study area (Soomere et al., 2015)." to make that clear, and deleted some less relevant remarks and repetitions on lines 16-19 of the original manuscript.*

3.2 p.10 line 24: Suggestion: approximation → fitting procedure
*Thank you, we have changed the wording accordingly*

40 3.2 p.10 line 28: 'Unexpected' does not explain the reason of the high values. On what basis the high values are assumed outliers? Looking at fig. 7, if the range of the setup is considered only in the range [0.01,0.4] maybe the distribution could have been 'forced' to be exponential. This part of the text is not sufficiently clear.

*The text was indeed too compact. We have added an explanation as to what is meant and why we think it is reasonable to exclude such values from the fitting procedure: „Thirdly, a few locations have several outliers – a small number of remarkably high set-up events that obviously do not follow the general appearance of the empirical distribution of set-up heights for the particular location (Fig. 6b, d, f). Such events apparently reflect severe storms in which the wind pattern has been favourable for the development of very large waves whereas such waves approach a certain coastal segment under a small angle. The presence of similar outliers is characteristic, for example, for time series of sea level in Estonian waters (Suursaar and Sooäär, 2007) and is associated with situations when strong storms blow from a specific direction. Similarly to outliers in sea level time series, these events likely follow an extreme value distribution (Coles, 2004) and thus should be left out from the analysis of the shape of the distribution of set-up heights."*

**Anonymous Referee #2 ()
The authors discuss the often less analysed, especially in climate science, but important topic of wave set-up heights and occurrence of these heights along a 60 km long part of the Estonian coastline at the Gulf of Finland. Whereas the overall topic raised by the authors is of interest, the way it is presented make it hard for the reader to understand the overall purpose of this study. In the present form, I can not recommend to accept this manuscript. However, after addressing some major concerns by the authors I would suggest to accept the publication after major revisions.
*Thank you for the recommendations towards improving the manuscript. We have carefully worked through and implemented all these recommendations in the revised manuscript.*

**General comments**
The main criticism is that in the current form it is not clear what is the scientific question they wanted to tackle in the first place and what is the main conclusion at the end of this work. Is it to establish some kind of climatology of wave set-up height in that area? Or is it to improve the general understanding of properties of the wave set-up?
*Thank you for this. We agree that the formulation of the core question (or hypothesis) of the study and the main conclusion were not perfect. The basic aim is to find out which type of the theoretical distribution describes the probability of occurrence of different wave set-up heights. As set-up may contribute up to 1/3 of the nearshore water level rise during strong storms, this question is of clear interest not only from the theoretical viewpoint but also for practical coastal management.*
*There is massive research into statistical properties of other drivers of high local water levels and reach of large water waves. The relevant distributions are very different: a Gaussian distribution for the water volume of the Baltic Sea, exponential for storm surges, quasi-Gaussian for water levels at the shores of the Baltic Sea, Weibull for different significant wave heights, and Weibull or Rayleigh for wave run-up. The process of wave set-up should be, to our understanding, no exception. Thus, it is just necessary to find out the shape of its typical probability density function. This is, in essence, the precondition for building an adequate climatology of set-up heights. We describe these arguments in more detail in the revised manuscript.*
*When we started the analysis of set-up heights, the expectation was that the result would be one of the listed distributions. Surprisingly, a Wald distribution popped up in a non-negligible manner. We have adjusted the Introduction, Discussion and Conclusions to make these aspects more clear.*

Can one translate the findings to other coastal regions, at least can one assume some?
*This question was also asked by Referee #1. It is possible to some extent. The applicability of an exponential distribution for the description of set-up heights seems to be a universal feature as it becomes evident also when we employed time series of offshore measured wave properties for such calculations. The presence of quasi-Gaussian of Weibull statistics for sheltered coastal areas is apparently a local feature. The emergence of another*

*distribution in some coastal segments is, to our understanding, a far more interesting feature, for which we have no decent explanation yet. However, as it emerges on many occasions, we think it is important to at least mention its presence and possible implications of its presence.*

What is the reason to find an approximation of the empirical distribution?

5 *The knowledge of the shape and parameters of such a distribution is often crucial in various forecasts and management decisions.*

Also, the authors should put more emphasis in the manuscript, as one of the most cited and may be important references Soomere et al. 2013 is missing in the reference list.

*We apologise for this shortcoming that is of course improved now. Also, we have revised the entire manuscript*
10 *for clarity and correct use of English.*

**Specific comments**

If it is understood correctly they have used a single wind observation in favour of simulated wind information as forcing for the wave model. They argued that the simulated wind of atmospheric models are some times
15 inaccurate and have some problems in representing the real wind fields. While this is certainly true, the analysis would benefit to present a short validation of the simulated wave field against some observation in the study area to show that a single forcing site is capable to represent the coastal wave climate reasonable.

*We agree with this criticism. To show that the modelled data set reasonably represents the local wave climate, we included into the paper a comparison of buoy measurements in the vicinity of the study area with modelled wave*
20 *data close to the buoy (end of Section 2.2 plus one new figure). The buoy data are available from 2012 and the comparison is performed for the time interval of 2012–August 2016. The basic properties of wave heights are represented reasonably. The bias is 0.05 m (which we consider as almost perfect, given the distance of the wind measurement site from the study area). The rms difference is 0.5 m. That is about twice as large as for the most recent direct simulations and on the level of this quantity for some other sub-basins of the Baltic Sea; however, it*
25 *is reasonable for our purposes as we do not need the exact sequence of wave events. The scatter diagram (not shown in the manuscript) is almost perfectly symmetric. The appearance of the relevant empirical probability distributions of the occurrence of different wave height is similar for both data sets.*

Which Equation is exactly used for estimating the wave set-up height. They stated Eq (8) is used, but they also introduce the effect of shoaling and refraction is this afterwards used or neglected.
30 *The mathematics of wave set-up is a complicated matter for waves that approach the shore at large incident angles and we probably compressed this material too much. In fact we use a sequence of equations. We start from Eq. (9) on page 8 to evaluate the changes to the wave properties owing to refraction and shoaling on their way from the model grid point to the breaker line. This equation gives us properties of this wave field at the breaker line. Thereafter we employ Eq. (7) to find the set-up height. The derivation of this equation is provided*
35 *because of intense discussion of the physics of breaking of obliquely approaching waves in Hsu et al. (2006) and Shi and Kirby (2008). We inserted a short version of this scheme into the text at the end of Section 3.2.*

Also, two Equations have the same reference number (9).

*We apologise for the presence of two equations (9). The second one on page 11 is not really used and is only presented to illustrate the functional form of the probability density function of a Wald distribution. It is*
40 *renumbered in the revised manuscript.*

Some of the figures should also be refined. Figure 2 shows the model domain and the coastal segments with coloured dots that represent the value of a parameter of an estimated approximation. At this point this is too much information for the reader, one does not know what parameter "*a*" stands for. For instance, a colour coding, or a

simple labelling of the number of the coastal segment would be of much more help later on, as in the other figures only the index number of the coastal section is displayed on the x-axis.
*Thank you for this observation. We have removed this colour coding from Fig. 2 and created a new figure 8 to highlight the spatial distribution of the values of this coefficient.*

5  It is not clear what is the difference between the left and right side of figure 5, besides more storms are involved on one side. What is the colour coding for?
*The left panel of Fig. 5 shows the time when the all-time highest waves occurred in the study area in 1981–2016, whereas the right panel of this figure shows the timing of the all-time highest set-up events. This figure is meant to explain that high waves in the nearshore of an area with complicated geometry does not necessarily mean high*
10  *wave set-up. The colour coding is necessary to separate storms in single years.*

**Anonymous Referee #3** ()
The authors present wave set-up calculations based on pre-computed wave model results for a shoreline with complex geometry located in the Gulf of Finland. The wave set-up distributions are based on long time series
15  (1981-2016) and they are calculated for separate segments along the coast. The wave set-up is approximated by accounting also for varying incident angles, and the approximation is compared to a more simple approach. The main finding is that while the wave set-up distribution is well represented by a standard exponential distribution in most segments, in about 25% the distribution does not follow any standard distribution (Gaussian, Weibull, Rayleigh). At these locations the distribution is fairly well approximated by an inverse Gaussian distribution,
20  meaning that the probabilities of the highest wave set-ups are higher with respect to the mean value compared to e.g. a Rayleigh distribution.
The paper is mostly well written and within the scope of Ocean Science. The results are also interesting. I can recommend that this paper is published in Ocean Science after major revisions. I have three main concerns. The first is that this paper is apparently a resubmission, but the results and conclusions have changed since the first
25  paper, and I don't know what to make of this. The second concern is regarding the precalculated wave data. The third is how the distributions are presented and, to a degree, interpreted.
*Many thanks for this. We hope to meet all the concerns in the revised manuscript.*

Main comment #1: The manuscript seems to be a resubmission of this paper: https://www.earth-syst-dynam-discuss.net/esd-2016-76/. Still, the previous version of the paper presents different results, suggesting that the
30  relevant distribution is inverse Gaussian at all locations instead of just 25%. The papers seem to be based on the same data, so I don't know what has changed. The first version of the paper claims:
"The distribution of wave set-up heights matches a Wald (inverse Gaussian) distribution along the entire study area. Even though different sections of the study area are open to different directions and host substantially different wave regimes, the leading term of the exponent in the associated inverse Gaussian distribution varies
35  insignificantly along the study area and generally is close to –1."
The current version claims:
"Even though different segments of the study area host substantially different wave regimes, the leading term of this polynomial is usually small (between –0.005 and 0.005) and varies insignificantly along the study area. Consequently, the distribution of wave set-up heights substantially deviates from a Rayleigh or Weibull
40  distribution (that usually reflect the distribution of different wave heights). In about ¾ of occasions it is fairly well approximated by a standard exponential distribution. In about 25% of coastal segments it matches a Wald (inverse Gaussian) distribution. "
The authors should make it clear why their results have changed from the previous paper.

*The manuscript is a substantially revised version of our paper titled "Inverse Gaussian distribution of wave set-up heights along a shoreline with complicated geometry", originally submitted to Earth Systems Dynamics and available at https://www.earth-syst-dynam-discuss.net/esd-2016-76/.*
*In our cover letter we informed the editor of Ocean Science about the history of the manuscript.*
5  *The Editor of Earth System Dynamics, Dr Anna Rutgersson mentioned in her final comment that „the presentation of the methods and results was not very clear" but encouraged some rewriting and a resubmission (which we understood as a hint that the paper could be better placed elsewhere).*
*The rejection from Earth Systems Dynamics was, in some way, fortuitous. Namely, during the rewriting and additional check of the results we discovered a bug in the script for the calculation of set-up heights and for the*
10  *subsequent evaluation of the parameters of their probability density function.*
*Removing this bug led to the conclusion that typically (in about 75% of coastal segments of the study area) set-up heights follow a classic exponential distribution but in about ¼ of the cases they follow a Wald distribution. In this sense the conclusions have changed significantly and it might be sensible to retract the early version from the website of Earth Systems Dynamics. We are willing to do so if it is acceptable according to the relevant policy of*
15  *Earth Systems Dynamics. Alternatively, we can discuss this issue in the current paper, with our apologies to the readers of the discussion version in Earth Systems Dynamics. In particular, we have inserted a relevant footnote into the manuscript (at the beginning of Section 3.2), and are willing to follow the recommendations of the editors.*

Main comment #2: It is not clear from the paper how the wave data has been obtained or how accurate it is. The
20  main assumption is that the wave data in the nearshore areas reach a steady state quickly, which is why pre-computed maps can be used. I assume the maps are chosen based on the wind direction and the wind speed. Still, has the runs themselves been modelled using wind data from a numerical weather prediction model? If not, how have the Baltic Sea wide simulations been produced that provide the boundary data? Data from atmospheric models have been used by several studies in the Baltic Sea (including nearshore areas) and it is the industry
25  standard. There is no reason to not produce the maps using proper wind data.
*The details of the scheme for rapid reconstruction of wave properties are extensively described in Soomere (2005). This publication is open to everybody and we think that it is not appropriate to repeat the details in this manuscript. We understand that the scheme is not perfect. In particular, it assumes that the wind field is homogeneous all over the Baltic Sea. This assumption is the weakest aspect of the use of one-point wind data,*
30  *either modelled or measured. Even though this assumption is usually not valid, the geometry of the Gulf of Finland and the location of the study area is such that for most of wind directions the waves that are generated in the Baltic Proper at latitudes to the south of the Gulf of Finland simply do not reach the study area. In this sense the results of calculations of the underlying maps are invariant with respect to the source of wind data.*

Are you using WAM 4.5.1 or a later version? Cycle 4.5.1 had a bug in the depth-induced wave breaking source
35  term that lead to unphysical high values at certain points. It has been fixed in cycle 4.5.4. This must be ruled out as an explanation for your surprising results.
*The maps were calculated using an early version of WAM cycle 4. As wave heights are not very large and both the wave periods and the propagation distance over shallow areas with possible depth-induced breaking are fairly small (usually no more than 1-2 grid cells), it is unlikely that bugs of this kind have any identifiable impact*
40  *on the results.*

Using Kalbådagrund data to force nearshore areas will overestimate the wind speed and probably result in an inaccurate wind direction. The authors cite some of their previous work, but I think some more details are needed also in this paper.

*We agree with all this; however, it is still the case that atmospheric models have difficulties with reproduction of strong wind directions for the Gulf of Finland. We provided more detailed arguments in the published response to the Referee and still think that the measured data more adequately represent the wind direction than the modelled data near the study area, at least during strong wind events that govern the wave heights larger than*
5 *the mean wave height.*

The validity of the modelling approach comes into question exactly because the authors present surprising results. Running a high-resolution model for 35 years is clearly a huge task. Creating the maps with a proper forcing is not. I think the Estonian Weather Service runs HIRLAM, which has been tried and tested in the GoF. A minimum requirement would be that the authors should identify the times that are responsible for creating the "surprising"
10 high wave set-ups and validate the instances by running the "full model". This will satisfy that, to the degree we can trust wave models, the surprising results are real.

*Thank you for this. We put considerable effort into retrieving locally produced high-resolution HIRLAM wind fields but this turned out to be a complicated problem. As the time foreseen for the revision approached rapidly, we chose another approach. To demonstrate the quality of the modelling approach for rapid reconstruction of*
15 *local wave properties, we provide in the revised version a fairly large (under the circumstances) comparison of measured and reconstructed wave data at the end of Section 2.2. This is also done to address the comment by Referee #2.*

*The comparison is performed for the time interval of 2012–August 2016 for which the buoy data are available in the vicinity of the study area. The basic properties of wave heights are represented reasonably. The bias is 0.05 m*
20 *(which we consider as almost perfect, given the distance of the wind measurement site from the study area and the simplicity of the calculation scheme). The rms difference is 0.5 m that is about twice as large as for the most recent direct simulations. A large part of this difference is because of mismatch of timing of modelled and measured wave events. This value is on the level of this quantity for some other sub-basins of the Baltic Sea (Björkqvist et al., 2018). We think that it is reasonable for our purposes as we do not need the exact sequence or*
25 *timing of wave events. The scatter diagram (not shown in the manuscript) is almost perfectly symmetric. The appearance of the relevant empirical probability distributions of the occurrence of different wave heights is similar for both data sets.*

Main comment #3: Usually the distributions are viewed and fitted by looking at the cumulative function, not the probability density function. The cumulative function is more stable, and I don't see any reason not to use it here.
30 *We agree with this; however, the use of a cumulative density function on many occasions smooths out differences in the shapes of the empirical and theoretical distributions. This feature is particularly relevant in our case as for large arguments the Wald distribution becomes similar to a Gaussian distribution. Our intention is to demonstrate that a large number (or proportion) of the empirical distributions follow at least locally a clearly different law than the classic distributions. This difference is better noticeable in terms of probability density*
35 *functions.*

The comments regarding "gaps" in the distribution are also confusing. What is the physical meaning of these gaps? The "first gap" is controlled by the accuracy of the wave data (10 cm, 1 cm etc), so why is this a meaningful point? There is also no reason to expect that experimental/modelled data should be without these gaps, so the existence of one is not a defining point in the distribution.
40 *We meant gaps in the calculated empirical distributions, not in the time series of wind or wave properties. It is natural that some specific set-up heights in such empirical distributions (in our case with a resolution of 1 cm) simply do not occur. This happens for very large values of set-up heights that are populated by a few events. We interpret the presence of such a gap as an indication that the number of occasions for the relevant set-up height*

*(and for the higher values of set-up) is too small for use in the estimates of the shape of the entire distribution. We explain this aspect more clearly in the revised manuscript.*

I am also a bit worried about the highest points being correlated. Is this the case? Then one event that is overestimated might "destroy" the entire distribution. Are the highest values recorded during the time ice was present? This needs to be evaluated carefully.

*We generally try not to include the highest values of wave set-up into estimates of the shape of the probability distributions for the same reasons. [Even though a comparison of measured and modelled data indicates that the ignoring of ice has not contributed to the „surprising" results.] This is why we look at the data points until the first „gap" in the distribution or limit the set-up heights to 0.4 m. Another reason for systematically ignoring the largest set-up heights is that they may represent another population of realisations with a completely different distribution (Coles, 2004). We plan to perform the analysis of extremes of set-up heights in the future using the technique of extreme value distributions.*

If you have found that the distribution of the wave set-up is really completely different for two neighbouring grid points about 500 m apart, then also give an example of the detailed shoreline structure that leads to such a dramatic variation so that the reader can be convinced what is going on and find some physical basis for this change in behaviour. Providing the structure of the shoreline that leads to this inverse-Gaussian behaviour would make this study highly useful for other coastal areas.

*The relevant comments are added to the description of the results. While some segments with non-Poisson features correspond to areas of rapid change in the orientation of the coastline, some other become evident along almost featureless shore segments. In general, a few segments having Gaussian or Weibull statistics are indeed exceptional whereas in the vicinity of segments having statistics that resemble an inverse Gaussian distribution the changes in the governing parameter „a" are fairly smooth.*

It is unclear how the second order polynomial fitting is done. It seems like you are fitting a polynomial to the data in the log-linear plot (is the variable p the probability density?). Please write the formula for the exponential distribution and how the polynomial fits into that, and for what fit this generalized exponential function becomes the "normal" exponential function. This will make it so much easier to follow the discussion and agree with the conclusions.

*This is a good recommendation. We have added the relevant remarks into Section 3.2, removed a typo in the description of the variables in the polynomial (p=probability was wrong; it is replaced by z=set-up height).*

I will save minor comments for the revised manuscript.
*We have carefully revised the entire manuscript and hope that the number of smaller issues is minimised.*

[revised manuscript text omitted]

---

## Referee Report (RR1)

Review comments to the revised version of "Variability of distributions of wave set-up heights along a shoreline with complicated geometry" (os-2019-25) by Tarmo Soomere, Katri Pindsoo, and Maris Eelsalu

I reviewed the original version of this manuscript, and concluded that it was worth publishing the results after major revisions of the manuscript. I have now read the authors' response to my comments and the revised manuscript. While the authors have cleared up several issues raised by the reviewers, I still have a few reasons to feel less than confident with the results and conclusions to be able to recommend that this version of the manuscript is ready for publication. As some of my comments might change the results, I have to label the possible revisions as major. This is not to say that progress hasn't been made with respect to the original version. Please find my comments below:

Main comment #1 (to the original manuscript): The authors have fully cleared up why the results are now different than in the previous submission to Earth System Dynamics. This matter has been fully addressed, and I leave it up to the editor how this should be reflected (if at all) in the final published manuscript.

Main comment #2 (to the original manuscript): My second main concern was the accuracy of the data. I recommended that the model would be run for a shorter time with a "full set-up" to validate the results. The authors chose to use measurements to validate their data, and while they are only available at one point, I think that this is a valid alternative to making additional model runs with a wind forcing from an atmospheric model. I accept this general approach taken by the authors to address my concerns. I also agree with the authors that a bias of 0.05 m is about as good as it gets. The revised manuscript states that the Tallinnamadal wave data are buoy data (although I have been under the impression that it is based on a pressure sensor). Is there a reference for these data?

So while I applaud the authors for using measurements to validate their model, I'm left feeling a bit uneasy about the results of the validation. The authors claim that "The appearance of the relevant empirical probability distributions of the occurrence of different wave heights is similar for both data sets (Fig. 3)". Still, I can't really see those two distributions being similar. The most important distinction is that a fit to measured data over 1 m would be reasonable well represented with a straight line (i.e. exponential fit), while a fit to the modelled data would be concave upwards. Doing the analysis on the model data would probably lead to a positive quadratic exponential, while a similar analysis on the measured data would result in a quadratic exponential close to zero. I understand that this is wave height and not wave set-up, but the former is a "forcing" for the latter. How can we trust the results of a quadratic exponential in the modelled wave set-up, if the shape of the distribution is not estimated correctly in the incoming wave height data?

Main comment #3 (to the original manuscript): This comment was concerning the fitting procedures and how they were presented. First, the authors have expanded on the description of the function they are fitting and what they are doing. This is now perfectly clear.

I also now understand why the authors want to use the gaps as a cut of the fitting: you want to avoid fitting the distribution to the "tail" of the data, which is not that statistically stable. I, however, still disagree that the gaps would have some kind of relevant meaning here. I would suggest that the upper limit of the fit would be a certain multiple of the mean wave set-up, or a certain amount of standard deviations above the mean. This would be a more objective and robust measure for the upper point than the gaps, which depend on the resolution of the binning, and also have a larger statistical variability (especially if this analysis should ever be reproduced using measurement data).

I also still disagree with the use of probability distributions instead of cumulative distribution. The

authors said that the cumulative distribution can smooth out differences. Another way to see it is that using the probability distribution exaggerates differences. Reviewer #1 also suggested using cumulative distributions, and while he mentioned extreme value analysis, I want to point out that the fitting of cumulative distributions are in no way restricted to performing extreme value analysis (which I know that you are specifically not doing). Still, since I haven't seen this type of fitting before, I can't say how big of a deal this really is. I would strongly suggest fitting to the cumulative distribution (as this is standard practice), but if you are set on fitting to the probability distribution, then, ultimately, I won't stop you.

I also disagree with the authors claim that the highest values would only be a part of some "extreme value distribution" and therefore outside the basic distribution (page 13, lines 16-17). This is not the case. When calculating e.g. block maxima the points in the resulting extreme value distribution are still members of the original underlying distribution. For example, sufficiently long block maxima of an exponentially distributed variable should follow a Gumbel distribution, but all of the block maxima were still points (but vary far up the tail) in the original exponential distribution. If you want to exclude outliers, don't use the previously mentioned claim to do it, since it is fundamentally incorrect.

In summary, I do believe that the basic idea adopted by the authors is correct, namely that the last (highest) points are infamously unstable, and excluding them from the fit might be warranted. My disagreement is with the motivation (citing extreme value distributions) and the fitting technique (probability distributions and use of gaps) adopted to implement this idea.

Main comment #4 (to the revised manuscript): This comment is based on information that was not available in the original manuscript. You use two simplifications in calculating the refraction and shoaling. Assumption of shallow water and Snell's law. You mention that the water depth can be between 4 and 27 m. This would mean a (deep water) wavelength of about 260 m and 1700 m meter to satisfy the shallow water condition (wave period up to 33 s!). This is not reasonable, especially in the Gulf of Finland. Calculating the true phase and group speed using iteration from the full dispersion relation is trivial, and I see no reason not to use it.

The second problem is Snell's law, since it assumes that the isolines in the bathymetry are straight and parallel. This might often be a good approximation, but looking at Fig. 10, this is a questionable assumption in Tallinn Bay. The proper refraction can be calculated using the full equations (as done in e.g. WAM). As one of your contributions in this study is to investigate the complex geometry and it's effect on the wave set-up, I find it surprising that you have neglected effects that seems to be important in this geographical area.

Specific comment #1: To really see the effect of the fitting range, please continue the blue line as (for example) a dashed blue line outside the fitting range to illustrate how well it captures the data that were not used for fitting.

Specific comment #2: On page, lines  you write: "In other words, in these locations the leading term a of the quadratic polynomial $az + bz + c$ is positive at a 95% significance level."

This isn't really true, since you didn't do the analysis for only one point. If you do the analysis for all points then, on average, around 5% of the points should give a "false positive" when using a 95% significance (if the null hypothesis was true). For you 25% >> 5%, so the effect is real, but the confidence in a single point is no longer positive at a 95% confidence level, and treating it as such seems to be some unintentional version of p-hacking. (I understand that there was no malicious intent here even though I used the phrase p-hacking. It was used only to make the point clear. This

comment was of a pure technical nature.)

Specific comment #3: Page 14, lines 16-17 "For large values of z this function behaves similarly to the probability density function of a Gaussian distribution."

Did you mean for small values of z?

Specific comment #4: Figure 8. I find it hard to believe that it wouldn't be possible to get a better Weibull fit than that presented in the figure, since the exponential distribution is contained in the Weibull family. Have you calculated the empirical cumulative distribution and plotted that in Weibull coordinates? You are claiming that Weibull is insufficient, so we need to make sure that this is right. As some other reviewer commented, it takes a lot to discard a three parameter Weibull.

Specific comment #5: As a suggestion, perhaps amend the "convex upwards" and "concave upwards" with (light tailed) and (heavy tailed) remarks. This might be especially useful the first time they are mentioned in the text, and perhaps in the discussion and conclusions, which some might read without going through the entire text and all the figures.

So e.g. on page 15, lines 26-27 the text would then read "The appearance of the distribution of modelled wave heights in the offshore (Fig. 3) is convex upwards (thin tailed) in the range of relatively frequent wave heights of 0.5–1.7 m."

---

## Referee Report (RR2)

Title: Variability of distributions of wave set-up heights along a shoreline with complicated geometry
Author(s): Tarmo Soomere and Katri Pindsoo
MS No.: os-2019-25
MS Type: Research article

**General comments**

The Authors did a thorough revision of the text which is now much more clear and readable. In particular Section 2 (Methods and data) and 3 (Results) have been greatly improved. Still the main problem of the study is the result, which indicates that in complex morphology the wave set-up can follow several (at least two) different statistical distributions, with no way to predict which one will be occurring at any given coastal point. On the whole the manuscript still suffers from being a bit too qualitative and somewhat lacking in rigour concerning the analysis, see specific comments. The latter would be a serious issue in any situation, but here the results presented are rather hard to accept and, furthermore, no clear explanation is given. So the thesis must be proved ' beyond a reasonable doubt'.

**Specific comments**

The only measure of uncertainty considered in the study is the use of the 95% significance level in the evaluation of the coefficients of the quadratic polynomial, which is not the same as to evaluate the goodness of the fit to the theoretical distribution, See figure 8.

A major weak point is the choice of a particular range of data to be fitted: a fixed interval [0.01,0.4] or a variable interval [.01,$\alpha$], where $\alpha$ the first gap in the distribution. It especially bothers me that in all cases shown, the frequency does not change with the increase of set-up height above 50 cm. Bottom line, in my opinion it appears that both choices are rather arbitrary. The first because the entire domain of setup heights is in the range [0,1] (fig.5) and the second because the threshold depends on the bin size. If I'm not badly mistaken the gaps would disappear if the frequencies were evaluated using different (i.e. larger) bins. Figure 3e-f show clearly the problem. Besides, how many points have been excluded in the region $x > 50$ $cm$ in order to get a straight line in figure 3f ?

The procedure of building the empirical distribution is not described in sufficient detail and this should really be fixed. The choice of the bins size in particular should be discussed showing what happens if classes of heights

are merged. This is a critical point for the kind of analysis proposed in the manuscript.

Another point to discuss in section 2 is the bathymetry, as it is critical factor in the evaluation of the set-up. In particular it should be verified that the bathymetry used to run the model has a resolution greater than or at least comparable with the model grid resolution. If not, the possible effect on the results should be discussed.

**Technical corrections**

The significant wave height is a statistical measure of the sea state on a wide area and during a long time, compared to a wave height at a fixed point in space and time. In section 2 they are sometimes used improperly (for example in paragraph 20). My suggestion is to define significant wave height as $H_S$ or $H_{m0}$ and monocromatc wave height as H, but more important is to stick to it in all the manuscript. A symbol for the set-up height would also help readability of the manuscript.

When Weibull distributions are considered is important to mention the value of the shape factor used, because it causes the the shape of the curve to change drastically.

---

## Referee Report (RR3)

**Review: Variability of distributions of wave set-up heights along a shoreline with complicated geometry**

Referee #

**General comments**

The Authors made a considerable effort in strengthening the statistical analysis in the study by using several techniques, among which the Kolmogorov-Smirnov test, thereby reinforcing the core message of the study, which is: it is found that in the complicated coastline considered, 75% of the points have wave set-up whose statistical distribution can be described by an exponential function, and 25% of the points have have set-up elevations whose statistical distribution can be described by an inverse exponential function.

This is still, in my opinion, the principal problem of the study. All results are obtained by processing input data coming from a wave model which is not phase resolved, so not particularly useful in coastal applications. Acknowledging the high scientific moral stand of the Authors, who present the results as they are, it important to consider also that the physical knowledge gained reading the manuscript, which is 27 pages long, plus 6 pages of bibliography and 13 figures, is quite limited. There are coastal points in which the distribution of the wave set-ups is exponential and points in which is inverse Gaussian. The points are mixed together and there is no way to separate them. Moreover the two descriptions are irreducible. The Authors did not find any possible physical reason or way to classify the points, in order to highlight oceanographical or hydrographic processes which may be of any concern to an oceanographer. And this is also due to the fact that no direct measure of wave set-up is presented, not from field campaigns, not from physical models in the laboratory, so there is no way to properly investigate the dynamics of the process which lie at the heart of the problem. It is clear that, as it is, the study would be of fairly limited use to any coastal engineer. But then, what would be the scientific interest for readers coming from to the more universal, and physically oriented, field of oceanography?

---

## Referee Report (RR4)

Review comments to "Variability of distributions of wave set-up heights along a shoreline with complicated geometry" (os-2019-25) by Tarmo Soomere, Katri Pindsoo, Nadezhda Kudryavtseva, Maris Eelsalu

I have reviewed a, once again, revised manuscript that presents how wave set-up is distributed in the southern shore of the Gulf of Finland. I want to thank the authors for putting in the effort to making substantial improvements. Especially the statistical tests are welcome and add rigor to the work. I feel my main concerns have been answered, and I can therefore recommend that this work should be accepted after minor revisions.

There are still quite large uncertainties in how reliable some of the results are, but think that the improvements made by the authors have made them more transparent in that the reader now has a better possibility to put the results in context. I think that few definite conclusions can be done based on this study, but it documents interesting findings and ideas that can possibly be studied further in other areas that e.g. have more validation data available.

My comments:

**Comment #1**
*P18, L11: "Selected segments"*

How are these selected? Or are you simply trying to say "some segments"?

**Comment #2**
*P18, L13-14: "it is likely that they mainly stretch the resulting distributions of set-up heights towards larger values but do not modify their basic shapes."*

Do you have a citation for this, or is this just your speculation? If the latter, I think it should be conveyed more clearly to the reader.

**Comment #3**
*P6, L26 We employ time series of wave properties (significant wave height, wave period and propagation direction)*

Perhaps write that it's peak period and mean direction (I know you mention that later)

**Comment #4**
*page 7 line 18 "The nearshore of the study area is divided into 174 coastal segments"*

Do you mean "The shoreline of the study area"?

**Comment #5**
*P7, L19-21 "Ignoring the presence of sea ice may lead to a certain overestimation of the overall wave energy in the region but apparently does not significantly distort the shape of the probability distribution of different wave heights (Fig. 3)"*

There are so many factors that impact the differences here, that drawing conclusions about what the role of ice is from this is a bit questionable. Perhaps just state "*Ignoring the presence of sea ice may*

*lead to a certain overestimation of the overall wave energy in the region."*, as one of the uncertainties that exist?

**Comment #6**
*P7,L32-P8,L9*

Here you explain why you didn't use numerical atmospheric models. I find this "model bashing" a bit uncalled for and you don't actually need it. The model of Keevallik and Soomer (2010) used was a 11 km HIRLAM run for 2007-2008. Today the current Harmonie-model has a resolution of 2.5 km (I think). Some studies using atmospheric models as forcing has also showed an accurate wave direction entering the coast (in the GoF, at least Björkqvist et al. 2017, Fig 2). Can't you just mention that Kalbådagrund provides an accurate direction, which is something that you need for this study? (PS. Didn't have access to Nikolkina et al. 2014)

Björkqvist et al. (2017): Improved estimates of nearshore wave conditions in the Gulf of Finland. https://doi.org/10.1016/j.jmarsys.2016.07.005

**Comment #7**
*P9, L30 and P10, L1*

Why do you introduce additional symbols for the significant wave height?

**Comment #8**
*P12, L1-3 "The processes that are not resolved by phase-averaged wave models such as reflection and diffraction may add even more wave energy to seemingly sheltered coastal segments."*

This is not entirely true. From the SWAN manual (bolded by me):

The following wave propagationprocesses are represented in SWAN:
• propagation through geographic space,
• refraction due to spatial variations in bottom and current,
• **diffraction,**
• shoaling due to spatial variations in bottom and current,
• blocking and reflections by opposing currents and
• transmission through, blockage by or **reflection against obstacles.**

(SWAN Scientific and Technical documentation, http://swanmodel.sourceforge.net/download/zip/swantech.pdf)

Perhaps just mention that they are not typically simulated by phase-averages models? You are, however, correct that diffraction is not "resolved" but parameterized in spectral models, but so is almost everything else. Reflection, however is probably resolved (haven't checked to code).

**Comment #9**
*P12, L20 "These aspects will be addressed in more detail elsewhere."*

Where? Also the entire section (lines 10-20) seems more like it belongs to the discussion.

**Comment #10**
*P12, L22-25*

I think the point was that the bins might be larger, not smaller, than 1 cm. Also, doesn't this belong to the part starting on page 14 line 7?

**Comment #11**
*P13, L9 "(grid point 106, Fig. 6d)"*
*P13, L15 "(e.g., grid point 129, Fig. 6e)"*

Should be 7d and 7e?

**Comment #12**
*P13, L17 "(e.g., grid point 1, Fig. 7)"*

Should this point to another figure?

**Comment #13**
Fig 8: Suggestion, put the value for the parameter "a" here to guide the reader to which way positive and negative values of a makes it bend?

**Comment #14**
Several references have incorrect formation in the doi: eg. Folkjs & Chhikara "https://doi.org./10.2307/2984691". Please go over the details of the references once more.

**Comment #15**
Make Figure 11 a four panel figure with one panel for each distribution? The panel with five distribution on top of each other is hard to read.

**Comment #16**
The title *"3.3 Fitting the empirical distribution with theoretical distributions"* doesn't really work, because you have been fitting distributions already in section 3.2. Perhaps one section with just the title "Testing goodness-of-fit", and the try to organize the other stuff logically before that?

**Comment #17**
I think this paper needs a separate section for discussion and conclusions. There is so much stuff and so many uncertainties, that you need to end with a section just for conclusions without muddling it up with discussion to answer "what can we actually take home from this study?"
* * *
I also have some general remarks that are more suggestions in that it is ultimately up to the authors to decide when they are satisfied with the structure of the manuscript. I do, however, feel that it would strongly be in the authors best interest to give it one more critical thought. The writing is sometimes quite verbose, and adding discussion in the results, along with some technical calculations, makes the reader lose the bearing easily. This isn't helped by the quite long, and sometimes weirdly names, sections.

**Suggestion #1**

Get rid of the general results section 3, and split that up into a couple of sections with clear titles.

**Suggestion #2**
P14, starting from line 7. This is mostly a sensitivity test. Would it be possible to put this into an separate appendix?

**Suggestion #3**
Subsection *2.2 "Wave time series in the nearshore of the study area"* is long, contains different subjects, and doesn't have a descriptive title. Perhaps split it up to one section for the "Study area", one for "Model implementation", one for "Measurements", and one for "Model validation" (I don't know if this would work exactly, but you get the point).

**Suggestion #4**
I think a lot of the text and stuff from the results section actually would belong to the discussion. Streamlining results, concentrating discussion to its own section, and ending by a well formulated conclusion section would make this paper a lot more approachable.

**Suggestion #5**
Have you though about having two subsections in the discussion? One discussion the stuff regarding the calculation of the set-up and the other about the distributions. Again, this would add a ton of structure to the latter part of the manuscript.

---

## Author Response (AR2)

Dear Sir/Madam:

We once more thank the Referees for their comments to the revised version of our submission. They again helped us to improve the manuscript. As usual, we have given careful consideration to all remarks; in particular, we have added a detailed discussion of goodness of fits of different theoretical distribution using the Kolmogorov-Smirnov test.

5 The relevant analysis was performed by Nadezhda Kudryavtseva. As this part is an essential constituent of the manuscript now, we think that she should be a co-author, and have included her into the authors' list.

Also, we have removed a few typos and checked once more formatting of references.

The comments of Referees are represented using normal font below and our response and a description of corrections using italics. All references to the changes in the manuscript are to the version that contains all corrections and is presented below.

10 Sincerely

Tarmo Soomere, Katri Pindsoo, Nadezda Kudryavtseva and Maris Eelsalu
07 April 2020

Referee #1

15 /---/ As some of my comments might change the results, I have to label the possible revisions as major.

*Thank you; we understand that.*

Main comment #1 (to the original manuscript): The authors have fully cleared up why the results are now different than in the previous submission to Earth System Dynamics. This matter has been fully addressed, and I leave it up to the editor how this should be reflected (if at all) in the final published manuscript.

20 *We shall be happy with any decision of the Editor on this issue.*

Main comment #2 (to the original manuscript): My second main concern was the accuracy of the data. I recommended that the model would be run for a shorter time with a "full set-up" to validate the results. The authors chose to use measurements to validate their data, and while they are only available at one point, I think that this is a valid alternative to making additional model runs with a wind forcing from an atmospheric model. I accept this general approach taken by the authors to address my concerns. I also agree with the authors that a bias of 0.05 m is about as good as it gets. The revised manuscript

25 states that the Tallinnamadal wave data are buoy data (although I have been under the impression that it is based on a pressure sensor). Is there a reference for these data?

*Thank you for noticing this. We re-checked this. The data indeed come from a pressure sensor (and this is why wave direction is not present). The accuracy of wave measurements is ±0.1 m. The description of the location and parameters of*

30 *the sensor: http://efficiensea.org/files/mainoutputs/wp4/efficiensea_wp4_27.pdf. There are some attempts to evaluate wave heights using standard navigation buoys but the relevant data are not yet reliable.*

/---/ I'm left feeling a bit uneasy about the results of the validation. The authors claim that "The appearance of the relevant empirical probability distributions of the occurrence of different wave heights is similar for both data sets (Fig. 3)". Still, I can't really see those two distributions being similar. The most important distinction is that a fit to measured data over 1 m

35 would be reasonable well represented with a straight line (i.e. exponential fit), while a fit to the modelled data would be concave upwards.

Doing the analysis on the model data would probably lead to a positive quadratic exponential, while a similar analysis on the measured data would result in a quadratic exponential close to zero. I understand that this is wave height and not wave set-up, but the former is a "forcing" for the latter. How can we trust the results of a quadratic exponential in the modelled wave

40 set-up, if the shape of the distribution is not estimated correctly in the incoming wave height data?

*We do not want to claim that our hindcast data is perfect. Most probably not. Our point is: Let's think that wave model distorts the statistics so that the replicated wave data show a shape of this distribution concave upwards (like a Weibull*

*distribution in log-linear coordinates that is actually expected here) instead of measured basically exponential distribution. It is then likely that a similar distortion (too large frequency of "intermediate"-height set-up events) should be evident in the distribution of set-up heights. Still the distribution of set-up heights is in quite many occasions concave downwards. This aspect is discussed on page 25, lines 11–19 below (equivalently, page 17, two last lines and page 18, lines 1–8 in the clean version of the manuscript.*

*We understand that this is not a particularly strong argument; however, it still says that concave downwards distributions of set-up heights probably reflect some specific features of the "conversion" of wave properties in to set-up heights.*

*To shed more light to the situation, we have considerably extended the analysis of distributions of set-up heights towards systematic comparison of their shape with several frequently occurring distributions. We created a new Section 3.3 to accommodate this description.*

Main comment #3 (to the original manuscript): This comment was concerning the fitting procedures and how they were presented. First, the authors have expanded on the description of the function they are fitting and what they are doing. This is now perfectly clear.

I also now understand why the authors want to use the gaps as a cut of the fitting: you want to avoid fitting the distribution to the "tail" of the data, which is not that statistically stable. I, however, still disagree that the gaps would have some kind of relevant meaning here. I would suggest that the upper limit of the fit would be a certain multiple of the mean wave set-up, or a certain amount of standard deviations above the mean. This would be a more objective and robust measure for the upper point than the gaps, which depend on the resolution of the binning, and also have a larger statistical variability (especiall y if this analysis should ever be reproduced using measurement data).

*Thank you for these suggestions. We have done this kind of analysis (Fig. 13) but using a somewhat „stronger" criteria for the match of the empirical and theoretical distributions. As discussed elsewhere, the gaps do not carry any specific meaning. For us, their presence signals that uncertainties of probabilities of even larger set-up heights may contain large uncertainties and it is not really justified to rely on these values.*

I also still disagree with the use of probability distributions instead of cumulative distribution. The authors said that the cumulative distribution can smooth out differences. Another way to see it is that using the probability distribution exaggerates differences. Reviewer #1 also suggested using cumulative distributions, and while he mentioned extreme value analysis, I want to point out that the fitting of cumulative distributions are in no way restricted to performing extreme value analysis (which I know that you are specifically not doing). Still, since I haven't seen this type of fitting before, I can't say how big of a deal this really is. I would strongly suggest fitting to the cumulative distribution (as this is standard practice), but if you are set on fitting to the probability distribution, then, ultimately, I won't stop you.

*Thank you for understanding. We are not totally against the use of cumulative distributions. It is just our understanding that they do not shed light to our problem and may even provide deceptive confidence. To make our point clear, we have added Figure 12 where this difference is clearly demonstrates. To avoid such possibly deceptive signal, we have chosen another way to compare empirical distributions with possible theoretical ones via D-values of Kolmogorov-Smirnov test*

I also disagree with the authors claim that the highest values would only be a part of some "extreme value distribution" and therefore outside the basic distribution (page 13, lines 16-17). This is not the case. When calculating e.g. block maxima the points in the resulting extreme value distribution are still members of the original underlying distribution. For example, sufficiently long block maxima of an exponentially distributed variable should follow a Gumbel distribution, but all of the block maxima were still points (but vary far up the tail) in the original exponential distribution. If you want to exclude outliers, don't use the previously mentioned claim to do it, since it is fundamentally incorrect.

*Thank you; our formulation was inappropriate indeed. Our wish was to stress that the largest values may belong to some other population of events similarly to the idea of (Suursaar and Sooäär, 2007). The sentence is deleted.*

In summary, I do believe that the basic idea adopted by the authors is correct, namely that the last (highest) points are infamously unstable, and excluding them from the fit might be warranted. My disagreement is with the motivation (citing

extreme value distributions) and the fitting technique (probability distributions and use of gaps) adopted to implement this idea.

Main comment #4 (to the revised manuscript): This comment is based on information that was not available in the original manuscript. You use two simplifications in calculating the refraction and shoaling. Assumption of shallow water and Snell's law. You mention that the water depth can be between 4 and 27 m. This would mean a (deep water) wavelength of about 260 m and 1700 m meter to satisfy the shallow water condition (wave period up to 33 s!). This is not reasonable, especially in the Gulf of Finland. Calculating the true phase and group speed using iteration from the full dispersion relation is trivial, and I see no reason not to use it.

*This must be misunderstanding. We do use the full dispersion relation for tracing the waves from the last wave model grid cell to the breaker line and only apply shallow water approximation at the breaker line. This is explained a few lines after Eq. (9). We added the word "full" to make this clear. We only use shallow water approximation at the breaker line. We are aware that this approximation and the evaluation of set-up height based on wave properties at this line are not perfect as dispersion still works to some extent in the surf zone. However, we think that the uncertainty or error generated via this approximation is still acceptable.*

The second problem is Snell's law, since it assumes that the isolines in the bathymetry are straight and parallel. This might often be a good approximation, but looking at Fig. 10, this is a questionable assumption in Tallinn Bay. The proper refraction can be calculated using the full equations (as done in e.g. WAM)

*The WAM model with the resolution we use provides wave data at a distance of a few hundred of metres from the breaker line. It is of course possible to evaluate refraction and shoaling using more elaborate methods; however, this would need phase-resolving models and bathymetry at much higher resolution.*

As one of your contributions in this study is to investigate the complex geometry and it's effect on the wave set-up, I find it surprising that you have neglected effects that seems to be important in this geographical area.

*We agree with this point. In essence, we study the problem by assuming that all coastal segments are (i) favourable for the formation of high set-up and (ii) are approximately homogeneous alongshore. These assumptions are only valid for selected segments of the study area. We added the relevant comment to Discussion (page 25, lines 22–24 below; equivalently, page 18, lines 10–12 in the clean version). However, we are of opinion that our study is still of some value as it establishes the typical shape of the distribution of wave set-up and its possible variation in ideal conditions. Another (quite weak) argument is that a sandy coastal area (favourable for set-up) with a similar complexity simply cannot exist.*

Specific comment #1: To really see the effect of the fitting range, please continue the blue line as (for example) a dashed blue line outside the fitting range to illustrate how well it captures the data that were not used for fitting.

*It would be nice to explore this item for single points; however, we are of opinion that doing so would defocus the presentation. Therefore, instead, we chose to present the magnitude of this effect in terms of the variation in the Kolmogorov-Smirnov statistic for the entire study area for three examples of the fitting range (Figure 13).*

Specific comment #2: On page, lines you write: "In other words, in these locations the leading term a of the quadratic polynomial $az^2 + bz + c$ is positive at a 95% significance level."

This isn't really true, since you didn't do the analysis for only one point. If you do the analysis for all points then, on average, around 5% of the points should give a "false positive" when using a 95% significance (if the null hypothesis was true). For you 25% >> 5%, so the effect is real, but the confidence in a single point is no longer positive at a 95% confidence level, and treating it as such seems to be some unintentional version of p-hacking. (I understand that there was no malicious intent here even though I used the phrase p-hacking. It was used only to make the point clear. This comment was of a pure technical nature.)

*Thank you; this is a good point, and we adjusted our formulations to make clear what we mean: this estimate of significance is valid for single points but not for the set of such points (page 22, lines 19–22 below; equivalently, page 15, lines 6–9).*

Specific comment #3: Page 14, lines 16-17 "For large values of z this function behaves similarly to the probability density function of a Gaussian distribution."

Did you mean for small values of z?

*Thank you for this question. It's actually a stupid typo (corrected now) as the inverse Gaussian goes as an exponential function ~exp(-az) for very large arguments (when z^(-3/2) is small compared to exp) while Gaussian goes as ~exp(-az^2).*

Specific comment #4: Figure 8. I find it hard to believe that it wouldn't be possible to get a better Weibull fit than that presented in the figure, since the exponential distribution is contained in the Weibull family. Have you calculated the empirical cumulative distribution and plotted that in Weibull coordinates? You are claiming that Weibull is insufficient, so we need to make sure that this is right. As some other reviewer commented, it takes a lot to discard a three parameter Weibull.

*We performed the relevant analysis (expanding substantially Section 3) in terms of D-values of Kolmogorov-Smirnov test and added a couple of images to make these aspects very clear. Just a small note: we use 2-parameter Weibull distribution as we do not consider extreme values of set-up. And of course, the more parameters a fitting function has, the better fit can be produced.*

Specific comment #5: As a suggestion, perhaps amend the "convex upwards" and "concave upwards" with (light tailed) and (heavy tailed) remarks. This might be especially useful the first time they are mentioned in the text, and perhaps in the discussion and conclusions, which some might read without going through the entire text and all the figures.

*Thank you for this idea. It has been implemented in a slightly different context as the Gaussian and generally Weibull distributions are light-tailed (~exp(-az^2)) and exponential and inverse Gaussian are heavy-tailed (~exp(-az)). This is now explained on page 21, lines 10–14 (equivalently, at the beginning of page 14 of the clean version). Btw, highlighting this difference partially explains why inverse Gaussian provides a good fit.*

So e.g. on page 15, lines 26-27 the text would then read "The appearance of the distribution of modelled wave heights in the offshore (Fig. 3) is convex upwards (thin tailed) in the range of relatively frequent wave heights of $0.5-1.7$ m."

*As pointed above, the property of being convex and having a heavy/light tail are not so directly related.*

Referee #2

The Authors did a thorough revision of the text which is now much more clear and readable. In particular Section 2 (Methods and data) and 3 (Results) have been greatly improved. Still the main problem of the study is the result, which indicates that in complex morphology the wave set-up can follow several (at least two) different statistical distributions, with no way to predict which one will be occurring at any given coastal point. On the whole the manuscript still suffers from being a bit too qualitative and somewhat lacking in rigour concerning the analysis, see specific comments. The latter would be a serious issue in any situation, but here the results presented are rather hard to accept and, furthermore, no clear explanation is given. So the thesis must be proved `beyond a reasonable doubt'.

*We generally agree with this remark and provide more detailed analysis of the match of obtained empirical distributions with commonly used theoretical distributions. However, we do not fully agree with the comment that the results are rather hard to accept. In essence, we show that in some 3/4 of cases the distribution of set-up heights follows (at least qualitatively) an exponential distribution. Thus, we understand that the problematic point is that another, not very often used, distribution pops up in some locations, and we do not have a good explanation for that. Such things are an intrinsic part of research, and we are of opinion that hiding such occasions would be totally wrong. In this sense we hope that even if we do not have a rigorous proof, our communication still has a value.*

Specific comments

The only measure of uncertainty considered in the study is the use of the 95% significance level in the evaluation of the coefficients of the quadratic polynomial, which is not the same as to evaluate the goodness of the fit to the theoretical distribution, See Figure 8.

*We expanded substantially Section 3 towards filling this gap.*

5 A major weak point is the choice of a particular range of data to be fitted: a fixed interval [0.01,0.4] or a variable interval [.01,α], where α is the first gap in the distribution.

*We actually did all the analysis several times for different cut-off schemes, one of these involving all data points. This has been explained in all revisions; now on page 21, 18–27 (equivalently, page 14, lines 7–14 in the clean version). The results were very consistent, so we decided to provide only one version in Figs. 9 and 10. This consistency is now thoroughly*
10 *illustrated Figure 13.*

It especially bothers me that in all cases shown, the frequency does not change with the increase of set-up height above 50 cm.

*This is simply because large set-up heights have occurred exactly once during the study period. This is also why we use several variations of cut-off.*

15 Bottom line, in my opinion it appears that both choices are rather arbitrary. The first because the entire domain of setup heights is in the range [0,1] (fig. 5) and the second because the threshold depends on the bin size. If I'm not badly mistaken the gaps would disappear if the frequencies were evaluated using different (i.e. larger) bins. Figure 3e-f show clearly the problem.

*This is indeed true, and this is another reason why we have used several cut-off schemes in Fig. 13 to be sure that our results*
20 *are consistent.*

Besides, how many points have been excluded in the region $x > 50$ cm in order to get a straight line in figure 3f? [*7f in the revised manuscript*]

*No points have been excluded in calculations of the red dashed line and 13 entries have been excluded in the calculation of the blue solid line.*

25 The procedure of building the empirical distribution is not described in sufficient detail and this should really be fixed. The choice of the bins size in particular should be discussed showing what happens if classes of heights are merged. This is a critical point for the kind of analysis proposed in the manuscript.

*In this respect we partially disagree. The procedure of building empirical distribution from time series is a standard counting exercise that, to our understanding, needs not to be described in a research paper.*

30 *As the modelled wave heights are evaluated with an accuracy of 1 cm, using even finer resolution for the construction of empirical probability distributions is not justified. As the total number of entries in time series of positive set-up heights is 40 000–70 000, using the resolution of 1 cm ensures that at least a few examples of set-up heights will belong to the relevant "size classes" of set-up heights down to frequencies about $10^{-2}$%. This range is apparently large enough to estimate the main properties of the distributions in question. We added this explanation into the beginning of Section 3.2.*

35 *Even though some details of the appearance of empirical distributions will change owing to a change in the bins' size, no large changes to the properties of approximating functions are expected (until the number of bins decreases below a certain limit). This is because such fitting operations are basically linear or rely on the main statistical parameters (e.g., the method of moments only takes into account the mean and standard deviation of the data set for 2-parameter distributions).*

Another point to discuss in section 2 is the bathymetry, as it is critical factor in the evaluation of the set-up. In particular it
40 should be verified that the bathymetry used to run the model has a resolution greater than or at least comparable with the model grid resolution. If not, the possible effect on the results should be discussed.

*To meet suggestions of another referee, we added a more detailed description of the wave model and its bathymetry into Section 2.2. In particular, we explain there that the bathymetry used in the innermost grid was constructed using high-resolution sailing maps by Estonian Maritime Board, with an efficient resolution ~200 m. We think this is enough for wave modelling at a resolution of about 470 m.*

5  Technical corrections

The significant wave height is a statistical measure of the sea state on a wide area and during a long time, compared to a wave height at a fixed point in space and time. In section 2 they are sometimes used improperly (for example in paragraph 20). My suggestion is to define significant wave height as HS or Hm0 and monochromatic wave height as H, but more important is to stick to it in all the manuscript.

10  *We distinguish the height of monochromatic waves (and denote its value at the breaker line by H_b) from significant wave height. The latter is, as recommended, defined in terms of m_0. The output of wave measurements is also given in terms of significant wave height. There is of course certain difference between how this quantity is defined in wave models and measurements; however, its explanation is beyond the scope of our study. The seeming inconsistency may arise at the point where we link the modelled significant wave height with the wave height at the breaker line. We have adjusted the text in a*
15  *few locations so that this logic is clearer.*

A symbol for the set-up height would also help readability of the manuscript.

*We introduced the symbol \eta_{max} in Eq. (9) but further on it seemed to us that the use of words "set-up height" is more traditional.*

When Weibull distributions are considered is important to mention the value of the shape factor used, because it causes the
20  shape of the curve to change drastically.

*Thank you; we indicate now the shape parameter, e.g., for Fig. 11 and 12.*

Referee #3

The manuscript discusses wave setup, which in some coastal areas and high wave events can be a significant proportion of
25  the total sea level at coast. The introduction gives a nice overview about the topic.

*Thank you for this.*

Maybe the recent studies in the Baltic Sea related e.g. to the analyse joint distributions of sea level and surface waves (Leijala et al., 2018. https://doi.org/10.5194/nhess-18-2785-2018) could be added.

*Thank you; it is important indeed to remind the reader that the baseline level of wave run-up also includes the local*
30  *elevation of water level owing to set-up. We added this reference to the relevant remark on page 3, line 3.*

My main concern is the reliability of the model results in coastal areas and their suitability for the study in question. Especially when the distributions show some exceptional behaviour at some of the coastal grid points.

*We fully agree with this point.*

Although the triple nested model setup has been used in earlier studies (Soomere 2005 and Soomere et al. 2013 (missing
35  from the references)) a brief description should be given here as well as it is important in order to interpret the results. I assume that the areas covered area the Baltic Sea, the Gulf of Finland (nest1) and the Estonian coastal area (nest2). What is used as forcing for the Baltic Sea and the Gulf of Finland grids? As they provide the boundary conditions, they play an important role in the accuracy of the results.

*We apologise for the missing reference (Soomere et al., 2013) in the original submission; it has been added. We also added*
40  *a description of the model setup into Section 2.2.*

*All models were used with Kalbådagrund wind data. Even though wind properties in Baltic proper may be different, waves generated in the sea area to the south of the latitudes of the Gulf of Finland almost never reach Tallinn Bay and thus are irrelevant in the context of our study.*

The authors state that the wind reanalysis and NWP systems available are not sufficiently accurate to model waves in coastal areas. This might have been the case ten years ago, but presently there are several sufficiently good quality reanalysis available and they have been successfully used in wave model studies in the Baltic Sea. And even if they are not perfect, they are able to better represent the spatial variability in the wind field than a single point measurement. Although Kalbådagrund is representative station for open sea wind conditions in the Gulf of Finland, the values are most likely too high to be used in coastal wave modelling and might results into too high wave energies.

*We agree that the quality of numerically replicated wind information is continuously increasing. To show that our results are adequate, we added a comparison with measured wave data at the northern border of the second nesting area into the end of Section 2.2. Our model forced with Kalbådagrund data lead to a bias of 5 cm and to a reasonably replicated the empirical distribution of occurrence of different wave heights. As the further propagation time of waves from the border of the second nesting until the shore is usually few dozens of minutes, variations in the wind properties in the nearshore apparently play a very minor role in the formation of the wave properties at the breaker line.*

It is not also clear to me how the use of 'precomputed maps of wave properties' affects the statistics compared to a full reanalysis or hindcast.

*This is probably a misunderstanding that has been already resolved in the first revision of the manuscript. We additionally adjusted the relevant parts of the text to make our point clear. In essence, our point is that this simplified scheme of replication of wave time series reasonably replicates also wave statistics. As just discussed, the numerically evaluated wave heights have bias of 5 cm compared to the measured ones and the model reasonably replicates the empirical distribution of occurrence of different wave heights. Thus, we are of opinion that the model, albeit it is greatly simplified, produces a realistic hindcast of time series of wave properties and reasonably replicates wave statistics as shown in Fig. 3. The process of using precomputed maps is quite complicated and is described in detail in (Soomere, 2005).*

The Authors refer to Baltic Sea wave model studies, that have shown wave model WAM as a good tool to model Baltic Sea wave fields. But the end results is not only based on the model, it is a combination of model, the selections (physics, numerics) you make within the model, accuracy of bathymetric data, the description of coastline with the given resolution, and the forcing used. Therefore some validation of the model setup used should be performed. If there is no data available from the coastal area in question, then the results from Gulf of Finland grid could be used for validation against the Gulf of Finland wave buoys. That would at least give some estimates on the reliability of the results.

*Thank you; we have added into first revision of Section 2.2 quite detailed comparison of hindcast and measured wave properties at the border of the second nested area (Fig. 2, Fig. 3). This is the only location in the vicinity of the study area where reliable wave data is available. It is thus likely that the wave model produces acceptable-quality wave time series in the open sea. Most of the innermost grid area has water depth >20 m and thus waves with typical periods for this area (3–4 s) practically do not "feel" the presence of bottom. As water depth increases rapidly offshore from the nearshore grid cells used for our calculations, WAM model propagates waves through the areas where the waves may experience substantial interaction with seabed only over one, maximum two grid cells. The bathymetry is built using high-resolution marine maps and thus is realistic. The propagation of waves from the closest to land grid cell to the breaker line is resolved within linear theory but its limitations are well known.*

Also one could check that the highest waves in the coastal areas are plausible given the accuracy of the offshore value, the bathymetry and the propagation angle of waves.

*The analysis of highest waves in the nearshore is provided in (Soomere et al., 2013 and Figure 3 in this source). We added the relevant remark to page 16, lines 25–28 below (equivalently, page 9, lines 7–10 of the clean version). This check is also included into the first revision by means of comparison of empirical probability distributions of simulated and measured wave heights (page 7–8, Fig. 3). This figure shows that during the comparison period three measured and two simulated significant wave heights exceeded 5 m whereas the maximum measured and simulated wave heights differ by 20 cm. This*

*sort of behaviour of extremes is realistic as significant wave height of 5.2 m has been recorded twice about 50 km from the study area since 2001. As for the further propagation of waves from the border of the innermost grid to the nearshore, we assume that the WAM model adequately does its job.*

Also the fact that the ice conditions are not included might distort the statistics. Have you done calculations by including only the ice-free periods?

*We agree that taking into account the presence of ice may change the results to some extent quantitatively. However, as the number of ice-free winters is increasing, we think that it is interesting to present ice-free statistics as a starting point.*

I suggest that the quality of the model results is addressed before considering this paper for publication in OS.

*As discussed above, we have performed a careful comparison of the model output against available measured wave data at the border of the study area.*

Also, I suggest considering to upgrade the model setup to newer WAM versions and using spatially varying wind forcing datasets.

*As pointed by other Referees, doing so would basically mean running the entire wave model for many decades and thus is not realistic.*

[revised manuscript text omitted]

---

## Author Response (AR3)

Dear Sir/Madam:

We carefully revised the manuscript in the light of comments of two referees and the Editor; first of all from the viewpoint of better structuring, removing of repetitions and redundant aspects.

About two pages of text have been moved into an Appendix. We would be happy to convert this part into supplementary material. A partial repetition of the list of different theoretical distributions is removed from the beginning of page 4 (originally lines 4–7). Whenever possible, the text has been made more compact. One panel of Fig. 13 is removed as it provides fairly limited information compared to the two remaining panels. Also, we have removed a few typos and checked once more formatting of references.

The comments of Referees are represented using normal font below and our response and a description of corrections using italics.

Sincerely

Tarmo Soomere, Katri Pindsoo, Nadezda Kudryavtseva and Maris Eelsalu

24 June 2020

Referee #1

General comments

The Authors made a considerable effort in strengthening the statistical analysis in the study by using several techniques, among which the Kolmogorov-Smirnov test, thereby reinforcing the core message of the study, which is: it is found that in the complicated coastline considered, 75% of the points have wave set-up whose statistical distribution can be described by an exponential function, and 25% of the points have set-up elevations whose statistical distribution can be described by an inverse exponential function.

This is still, in my opinion, the principal problem of the study. All results are obtained by processing input data coming from a wave model which is not phase resolved, so not particularly useful in coastal applications.

*We appreciate the recognition of our efforts by the Referee but still do not agree with the point that phase-averaged models are not particularly useful in coastal applications. We agree that phase-resolving models are definitely necessary for calculations of dynamic wave set-up and run-up of single waves but would like to emphasize that running such models for climatic purposes along longer coastline sections (as we do) is an extremely resource-consuming task. Also, it is not clear how to produce wave phases from the output of phase-averaged models or from wave generation models. Therefore, we are of opinion that phase-averaged models and associated approaches are still valid and contemporary tools for scientific research.*

Acknowledging the high scientific moral stand of the Authors, who present the results as they are, it important to consider also that the physical knowledge gained reading the manuscript, which is 27 pages long, plus 6 pages of bibliography and 13 figures, is quite limited.

*We partially agree with these critics in terms of the length of bibliography and figures (but just mention that the manuscript itself was 18 pages long). Also, we would like to emphasize that quite substantial amount of material was added to the manuscript to meet (repeated) suggestions of referees. (In fact, we started from a 12-page manuscript and 7 figures.) Following the recommendation of the Editor, we have moved some technical parts of the manuscript into an Appendix.*

5    *However, we disagree in terms of advancement of our knowledge presented in the manuscript. As publications of the relevant earlier research (Hsu et al., 2006; Shi and Kirby, 2008) contained typos, it was necessary to bring the correct sequence of expressions for the calculations of set-up height (even if this information was not new). Based on this, we established the basic shape of the empirical distribution of set-up heights. This is definitely a new knowledge, generally worth of publishing alone. Then we saw that this distribution has an unexpected shape in some coastal segments. To*

10    *understand what's going on, we (i) performed systematic approximation of this shape, (ii) established where it is a sort of reflection of a Poisson process (with certain statistical significance) and (iii) evaluated how the parameters of this approximation vary along the shore. To our eyes, it is an absolutely necessary step on the way towards finding a reasonably fitting distribution. Even though this sort of mathematics is not in everyday use in oceanography, it is a feasible way to understand things. Then we took the challenge of using an inverse Gaussian distribution. This caused quite some reaction*

15    *from referees and the necessity to include some more statistical testing. Finally, we have proved that inverse Gaussian distribution gives systematically better fit than other classic (frequently used) distributions.*

There are coastal points in which the distribution of the wave set-ups is exponential and points in which is inverse Gaussian. The points are mixed together and there is no way to separate them. Moreover the two descriptions are irreducible.

*Not really. For very large values of set-up heights the inverse Gaussian behaves like an exponential distribution. This is*

20    *explained at the top of p. 13 of the clean revised version.*

The Authors did not find any possible physical reason or way to classify the points, in order to highlight oceanographical or hydrographic processes which may be of any concern to an oceanographer.

*We agree with this. Even though we tried several hypotheses, we did not find a decent explanation. However, we do not think that this is a failure or a severe weakness of our study. In contrary, we hope that this situation motivates other, possibly*

25    *brighter, scientists to understand the reasons for such behaviour of set-up heights (which possibly do not become evident on largely straight open ocean shores and may remain unnoticed there). Also, we hope that the amount of new understanding (see above) deserves publication.*

And this is also due to the fact that no direct measure of wave set-up is presented, not from field campaigns, not from physical models in the laboratory, so there is no way to properly investigate the dynamics of the process which lie at the

30    heart of the problem.

*There exists quite a large pool of literature about field and laboratory measurements of wave set-up. We added one more reference to (Dean and Walton, 2010) who gave a nice overview of the relevant efforts.*

It is clear that, as it is, the study would be of fairly limited use to any coastal engineer.

*We strongly disagree with this point. The identification of a decent and suitable theoretical probability distribution is one of the core steps in understanding and forecast of different threats, vulnerability and openness indices etc. What we say is simply: be aware because common distributions may fail to represent the danger by wave set-up.*

But then, what would be the scientific interest for readers coming from to the more universal, and physically oriented, field of oceanography?

*This is exactly where (we think) our contribution is. For example: water level in basins like the Baltic Sea is generally driven by three mechanisms: water volume of the entire sea, local storm surge and wave set-up. The first one follows well a Gaussian distribution, the second one satisfactorily an exponential distribution – and now we see that the third one may follow a third distribution. This is both a warning sign (as it is not easy to estimate the joint probability) and a challenge (to create methods that allow handling of three different distributions).*

Referee #2

/--/ I want to thank the authors for putting in the effort to making substantial improvements. Especially the statistical tests are welcome and add rigor to the work. I feel my main concerns have been answered, and I can therefore recommend that this work should be accepted after minor revisions.

*Thank you for this opinion. We have adjusted the manuscript according to all suggestions. Following the recommendation of the Editor, we have also converted some technical parts of the manuscript into supplementary material.*

There are still quite large uncertainties in how reliable some of the results are, but think that the improvements made by the authors have made them more transparent in that the reader now has a better possibility to put the results in context. I think that few definite conclusions can be done based on this study, but it documents interesting findings and ideas that can possibly be studied further in other areas that e.g. have more validation data available.

*We have reshaped the discussion so it considers the possible uncertainties in blocks and makes sure which ones may be considered as possible reasons for the development of a specific shape of distributions of set-up heights.*

Comment #1: *P18, L11: "Selected segments"* – How are these selected? Or are you simply trying to say "some segments"?

*Yes, indeed, we meant that high set-up events may only occur in some segments of the study area.*

Comment #2: *P18, L13-14: "it is likely that they mainly stretch the resulting distributions of set-up heights towards larger values but do not modify their basic shapes."* – Do you have a citation for this, or is this just your speculation? If the latter, I think it should be conveyed more clearly to the reader.

*This conjecture is based on the result of (Dean and Bender, 2006). They show that the presence of vegetation (or the impact of bottom shear stress) basically means that, to a first approximation, all set-up heights are multiplied by a constant compared to the ideal situation. In such cases the basic shape (convex/concave/following a straight line) of the relevant empirical distribution should not change much. (Minor changes may occur if size classes are not redefined). We have added the relevant comment.*

Comment #3: *P6, L26 We employ time series of wave properties (significant wave height, wave period and propagation direction)* – Perhaps write that it's peak period and mean direction (I know you mention that later).

*Thank you; doing so makes reading easier indeed.*

Comment #4: *page 7 line 18 "The nearshore of the study area is divided into 174 coastal segments"* – Do you mean "The shoreline of the study area"?

*Yes, of course.*

Comment #5: *P7, L19-21 "Ignoring the presence of sea ice may lead to a certain overestimation of the overall wave energy in the region but apparently does not significantly distort the shape of the probability distribution of different wave heights (Fig. 3)"* – There are so many factors that impact the differences here, that drawing conclusions about what the role of ice is from this is a bit questionable. Perhaps just state "*Ignoring the presence of sea ice may lead to a certain overestimation of the overall wave energy in the region.*", as one of the uncertainties that exist?

*We fully agree, and (following the recommendation of the editor) deleted this conjecture (which, we agree, was more a hope than a conclusion).*

Comment #6: *P7,L32-P8,L9* – Here you explain why you didn't use numerical atmospheric models. I find this "model bashing" a bit uncalled for and you don't actually need it. The model of Keevallik and Soomere (2010) used was a 11 km HIRLAM run for 2007-2008. Today the current Harmonie-model has a resolution of 2.5 km (I think). Some studies using atmospheric models as forcing has also showed an accurate wave direction entering the coast (in the GoF, at least Björkqvist et al. 2017, Fig 2). Can't you just mention that Kalbådagrund provides an accurate direction, which is something that you need for this study? (PS. Didn't have access to Nikolkina et al. 2014).

Björkqvist et al. (2017): Improved estimates of nearshore wave conditions in the Gulf of Finland. https://doi.org/10.1016/j.jmarsys.2016.07.005.

*We agree. This is much better and somewhat shorter argument. We have also deleted the references (Soomere and Keevallik, 2010) and (Nikolkina et al. 2014) that are not necessary in the new formulation.*

Comment #7: *P9, L30 and P10, L1* – Why do you introduce additional symbols for the significant wave height?

*We think it is easier for the reader to follow the line of thoughts. The relevant symbol for wave height without subscript is just generic. One of the earlier referees requested that the version of significant wave height we use should be denoted by $H_S$ and defined in terms of zeroth-order moment. We changed the text accordingly. Then we need a symbol for the approach angle at the wave model grid cell. Even though the wave height at this location could be denoted without subscript, we believe that using "0" as subscript also there makes formulas easier to comprehend. Thus, we would like to leave this as it is.*

Comment #8 *P12, L1-3 "The processes that are not resolved by phase-averaged wave models such as reflection and diffraction may add even more wave energy to seemingly sheltered coastal segments."* – This is not entirely true. From the SWAN manual (bolded by me):

The following wave propagation processes are represented in SWAN:

•propagation through geographic space,

•refraction due to spatial variations in bottom and current,

•diffraction,

•shoaling due to spatial variations in bottom and current,

•blocking and reflections by opposing currents and

•transmission through, blockage by or reflection against obstacles.

(SWAN Scientific and Technical documentation, http://swanmodel.sourceforge.net/download/zip/

swantech.pdf)

Perhaps just mention that they are not typically simulated by phase-averages models? You are, however, correct that diffraction is not "resolved" but parameterized in spectral models, but so is almost everything else. Reflection, however is probably resolved (haven't checked to code).

*Thank you for this. The claim is correct for the WAM model, and we adjusted it accordingly.*

Comment #9: *P12, L20 "These aspects will be addressed in more detail elsewhere."* – Where? Also the entire section (lines 10-20) seems more like it belongs to the discussion.

*We deleted this sentence as we had in mind a manuscript in progress. The point of this section is to highlight and explain possible differences between the earlier and current research in the light of information displayed in Figure 6. Even though it might be not in the focus of our message, we think that it is important to make sure how our results compare with the earlier research in S2013. As this is not really relevant to the core message, we prefer to leave these remarks in the Results section.*

Comment #10: *P12, L22-25* I think the point was that the bins might be larger, not smaller, than 1 cm. Also, doesn't this belong to the part starting on page 14 line 7?

*Yes, indeed; however, then we could smooth out important features of the shape of the resulting distribution. For this reason we decided to make this aspect clear at the beginning of the subsection.*

Comment #11: *P13, L9 "(grid point 106, Fig. 6d)"; P13, L15 "(e.g., grid point 129, Fig. 6e)"* –Should be 7d and 7e?

*Yes, thank you. We added a figure into the previous revision and evidently did not spot all necessary corrections.*

Comment #12: *P13, L17 "(e.g., grid point 1, Fig. 7)"* – Should this point to another figure?

*Yes, it must be Fig. 8 for the same reason.*

Comment #13: Fig 8: Suggestion, put the value for the parameter "a" here to guide the reader to which way positive and negative values of a makes it bend?

*Thank you; we did so.*

Comment #14: Several references have incorrect formation in the doi: e.g. Folks & Chhikara "https://doi.org./10.2307/2984691". Please go over the details of the references once more.

*Thank you, this was wrong by some reason indeed. We found and corrected several other minor typos of the doi indices.*

Comment #15: Make Figure 11 a four panel figure with one panel for each distribution? The panel with five distributions on top of each other is hard to read.

*We intentionally split Fig. 11 into two panels. They should be next (or close) to each other for better view. Then Fig. 11a shows that an inverse Gaussian distribution provides a nice match, and Fig. 11b demonstrates that the three others have much worse match. If this will not work, we will be happy to reshape the panels.*

Comment #16 The title *"3.3 Fitting the empirical distribution with theoretical distributions"* doesn't really work, because you have been fitting distributions already in section 3.2. Perhaps one section with just the title "Testing goodness-of-fit", and the try to organize the other stuff logically before that?

*We have adjusted the title of Section 3.3 and reorganised some of its material.*

Comment #17

I think this paper needs a separate section for discussion and conclusions. There is so much stuff and so many uncertainties, that you need to end with a section just for conclusions without muddling it up with discussion to answer "what can we actually take home from this study?"

*We definitely agree with this observation and have included a short section of conclusions. However, we would still like to mention that – according to Fig. 3 – the uncertainties associated with reconstruction of wave properties have jointly produced a (more) convex upwards distribution of modelled wave heights than show the measured wave heights. Also, the shape of the distributions of set-up heights is invariant with respect to some other unknown variables such as bottom stress or presence of vegetation (Dean and Bender, 2006). We hope that these aspects are now made clearer in the text.*
* * *
I also have some general remarks that are more suggestions in that it is ultimately up to the authors to decide when they are satisfied with the structure of the manuscript. I do, however, feel that it would strongly be in the authors best interest to give it one more critical thought. The writing is sometimes quite verbose, and adding discussion in the results, along with some technical calculations, makes the reader lose the bearing easily. This isn't helped by the quite long, and sometimes weirdly names, sections.

Suggestion #1

Get rid of the general results section 3, and split that up into a couple of sections with clear titles.

*We have used this kind of attitude in some other manuscripts and usually got a strict advice from the editor or technical editor to use the classic IMRAD scheme. However, we adjusted the subtitles so that they better reflect the content and moved some technical parts into Appendix.*

Suggestion #2

P14, starting from line 7. This is mostly a sensitivity test. Would it be possible to put this into a separate appendix?

*We have made this part of the text more compact. To our understanding, it is too short for another appendix.*

Suggestion #3

Subsection *2.2 "Wave time series in the nearshore of the study area"* is long, contains different subjects, and doesn't have a descriptive title. Perhaps split it up to one section for the "Study area", one for "Model implementation", one for "Measurements", and one for "Model validation" (I don't know if this would work exactly, but you get the point).

*Thank you for this observation. This section grew steadily after every round of revision and became indeed too long in the previous version. As the implementation of the wave model has been already described in several publications (including open access research papers) and is neither new nor instructive, and the Editor also recommended to move some parts of the manuscript into supplementary material, we decided to move the description of model implementation and the use of wind data into an Appendix, and to shorten the text to some extent as described above. We also added into the Appendix a remark (by some reason missing from the previous version) on how the Kalbadagrund wind data were reduced to the standard height of 10 m used in the WAM model.*

Suggestion #4

I think a lot of the text and stuff from the results section actually would belong to the discussion. Streamlining results, concentrating discussion to its own section, and ending by a well formulated conclusion section would make this paper a lot more approachable.

*This is definitely true for some parts of the text, and we reshaped these parts accordingly. However, some other parts (e.g., discussion of differences between the earlier research and our simulations in light of Fig. 6) are, to our understanding, part of justification of the method and should not be brought to the attention of the readers as conclusions.*

Suggestion #5

Have you thought about having two subsections in the discussion? One discussion the stuff regarding the calculation of the set-up and the other about the distributions. Again, this would add a ton of structure to the latter part of the manuscript.

*Thank you; we have done so, with one deviation from this suggestion: we have started from the distributions and then discussed how calculations of wave properties and set-up may affect the distributions. In this light, we also decided to change one sentence in Abstract.*

=========================================================

[revised manuscript text omitted]